# DeepBlip: Estimating Conditional Average Treatment Effects Over Time

**Haorui Ma** [1 2]  **Dennis Frauen** [1 2]  **Stefan Feuerriegel** [1 2]

## Abstract

Structural nested mean models (SNMMs) are a principled approach to estimate the treatment effects over time. A particular strength of SNMMs is to break the joint effect of treatment sequences over time into localized, time-specific "blip effects". This decomposition promotes interpretability through the incremental effects and enables the efficient offline evaluation of optimal treatment policies without re-computation. However, neural frameworks for SNMMs are lacking, as their inherently sequential g-estimation scheme prevents end-to-end, gradient-based training. Here, we propose DeepBlip, the first neural framework for SNMMs, which overcomes this limitation with a novel double optimization trick to enable simultaneous learning of all blip functions. Our DeepBlip seamlessly integrates sequential neural networks like LSTMs or transformers to capture complex temporal dependencies. By design, our method correctly adjusts for time-varying confounding to produce unbiased estimates, and its Neyman-orthogonal loss function ensures robustness to nuisance model misspecification. Finally, we evaluate our DeepBlip across various clinical datasets, where it achieves state-of-the-art performance.

## 1. Introduction

Predicting the effects of treatment sequences is crucial for personalized medicine to choose the best therapeutic strategy for a patient based on their history (Feuerriegel et al., 2024). Methodologically, the **conditional average treatment effect (CATE)** *over time* captures the combined effect of multiple treatments in the next $\tau$ time steps (see Fig. 1). In clinical practice, CATEs over time can be frequently estimated from observational data with patient histories, such as the electronic health records (Allam et al., 2021; Bica et al., 2021). Yet, a key challenge is *time-varying confounding*, which arises when past treatments affect future covariates that, in turn, influence both subsequent treatment assignment and the outcome of interest.

Several statistical strategies exist to address time-varying confounding. While these strategies all target the same estimand, they differ in their practical strengths, especially in finite-sample settings. ① One approach is *G-computation* (Robins, 1986; Robins et al., 1999), with neural instantiations such as **G-Net** (Li et al., 2021) and **IGC-Net** (Hess et al., 2026a). This approach sequentially models the conditional outcome given treatment and covariates, and then averages (integrates) over the distribution of time-varying covariates under specific treatment regimes. ② Another approach is *marginal structural models (MSMs)* (Robins, 1994; 2004), with neural instantiations such as **R-MSNs** (Lim, 2018). However, MSMs rely on inverse propensity weighting and can be unstable as extreme weight often occurs in long prediction windows. ③ A third approach is ***structural nested mean models (SNMMs)*** (Robins, 1994; 2004). A particular strength of SNMMs is their ability to break the joint effect of treatment sequences over time into localized, time-specific ***"blip effects"***. This yields a simpler estimand that does not require learning full counterfactual outcome trajectories, which can be especially beneficial over long horizons. In practice, this can have several benefits: the incremental effects are interpretable, and, the blip effects can be reused to predict treatment effects for new treatment sequences without re-computation, which allows for identifying optimal treatment sequences through offline evaluation. $\Rightarrow$ *Our work is located in the third stream and contributes to research on SNMMs.*

However, neural implementations of SNMMs are missing. The closest effort in this direction is **Dynamic DML** by (Lewis & Syrgkanis, 2021); however, this approach has several *limitations*: *(i)* The original implementation is based on linear models. While, in principle, other machine learning models *could* be used, this is impractical because essentially *a different machine model must be estimated for each time step*, so the complexity *grows linearly with the prediction horizon*. *(ii)* Dynamic DML uses a sequential minimization scheme that does not scale to large datasets. Hence, their training follows an *iterative* procedure, while our second

[1]AI in Management, LMU, Munich, Germany [2]Munich Center for Machine learning, Germany. Correspondence to: Haorui Ma <H.Ma@lmu.de>, Stefan Feuerriegel <feuerriegel@lmu.de>.

*Proceedings of the $43^{rd}$ International Conference on Machine Learning*, Seoul, South Korea. PMLR 306, 2026. Copyright 2026 by the author(s).

*Table 1.* **Methods for learning CATE over time**.

| Stream | Method | Methodological advantage | | Benefits for medical practice | |
|---|---|---|---|---|---|
| | | Unbiased | Neural | Orthogonal | Offline efficiency |
| W/o adjustment | CRN (Bica et al., 2020) | ✗ | ✓ | ✗ | ✓ |
| | CT (Melnychuk et al., 2022) | ✗ | ✓ | ✗ | ✓ |
| G-computation | G-Net (Li et al., 2021) | ✓ | ✓ | ✗ | ✗ |
| | IGC-Net (Hess et al., 2026a) | ✓ | ✓ | ✗ | ✗ |
| MSMs | R-MSNs (Lim, 2018) | ✓ | ✓ | ✗ | ✗ |
| | DR-learner (Frauen et al., 2025) | ✓ | ✓ | ✓ | ✗ |
| SNMMs | Dynamic DML (Lewis & Syrgkanis, 2021) | ✓ | ✗ | ✓ | ✓ |
| | **DeepBlip (ours)** | ✓ | ✓ | ✓ | ✓ |

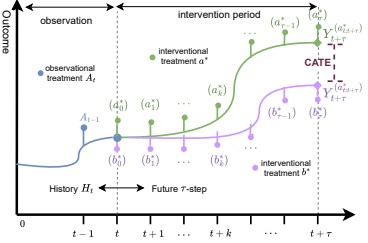

*Figure 1.* **CATE over time.**[1]

stage later is *end-to-end*. *(iii)* Dynamic DML is unable to process the full, accumulating history of a patient. Its architecture requires a fixed-size input, preventing it from conditioning on patient data (clinical records) that grows over time. Yet, building a neural framework for SNMMs is non-trivial. One major obstacle is that the SNMM formulation inherently prevents end-to-end learning due to its sequential scheme. For the same reason, Dynamic DML cannot be directly adapted. As a remedy, we propose a new *double optimization trick* to break the sequential dependence and enable gradient-based training. This approach allows us to address *(i)*–*(iii)*.

Here, we propose **DeepBlip**, the first *neural* framework to estimate CATE over time by leveraging the blip function from SNMMs. DeepBlip decomposes the joint effect of treatment sequences over time into localized, time-specific blip effects, which enables more tractable and stable learning. As a result, our DeepBlip adjusts for time-varying confounding and is thus *unbiased*. To do so, a key novelty is that we propose a new *double optimization trick* to break the sequential dependence in SNMMs and allow for gradient-based training. Our DeepBlip is built on top of sequential neural networks (e.g., LSTMs, transformers) to capture complex temporal dependencies. For this, it employs a two-stage architecture: Stage 1 models the probability of time-varying treatments and mean outcomes conditioned on a patient's history, while Stage 2 reformulates $g$-estimation (Robins, 1994; 2004) as a risk minimization task to directly learn the blip functions.

Our DeepBlip has several further strengths for medical practice: **(i)** The estimated blip effects are *interpretable*, so that clinicians can understand the incremental gain from treatment. **(ii)** The loss function in DeepBlip is *Neyman-orthogonal*, which makes the estimates robust against model misspecification. **(iii)** The learned blip effects are reusable, which allows clinicians to predict outcomes under novel treatment sequences *without re-computation*. This offers an efficient approach for offline evaluation of different therapeutic strategies and thus to search for optimal treatment sequences. Formally, at inference time, DeepBlip can identify the optimal treatment sequence within just one forward pass. This is unlike other methods, which typically require

either re-training (Li et al., 2021; Hess et al., 2026a) or multiple forward passes due to exhaustive search (Lim, 2018; Bica et al., 2020; Melnychuk et al., 2022).

Our **contributions** are three-fold:[2] (**1**) We introduce the first neural framework to predict CATE over time via the SNMM framework. (**2**) Our framework is carefully tailored to medical practice by benefiting from robustness due to Neyman-orthogonality and efficient offline evaluation. (**3**) We conduct extensive experiments across multiple medical datasets to demonstrate that our DeepBlip is effective and also robust across long time horizons.

**Conflict of interest disclosure:** The authors declare no financial conflicts of interest related to this work.

## 2. Related Work[3]

**Estimating CATE in the static setting:** There has been extensive research in estimating CATE with neural networks in the static setting (Wager & Athey, 2018; Alaa & van der Schaar, 2017; Shalit et al., 2017; Yoon et al., 2018; Künzel et al., 2019; Curth & van der Schaar, 2021). However, these methods are aimed at static settings and thus struggle with medical datasets such as electronic health records, where patient histories are recorded *over time* and give rise to *time-varying confounding*.

**Estimating ATE over time:** One line of work has developed methods for the ATE over time (Frauen et al., 2023; Shirakawa et al., 2024). However, the ATE captures only population-level effects and thus overlooks differences in treatment effectiveness across patients. In contrast, we focus on the CATE, which provides a more granular, individualized estimate of treatment outcomes, which is highly relevant for personalized medicine (Feuerriegel et al., 2024).

**Estimating CATE over time:** There are several *neural* methods for this task[4], which can be broadly categorized

---

[1]Trajectories of potential outcomes under two interventional sequences $a^*_{t:t+\tau}, b^*_{t:t+\tau}$ given the shared observed history $H_t$. The difference between the two curves is the CATE over time.

[2]Code is available at GitHub.

[3]We provide an extended related work in Appendix B

[4]There have been some attempts to use non-parametric models

into different streams (see Table 1):

• *No proper adjustment for confounding and thus bias:* Some methods for CATE over time fail to properly adjust for time-varying confounding, which leads to estimates that are *biased*. Here, prominent examples are counterfactual recurrent network (**CRN**) (Bica et al., 2020) and the causal transformer (**CT**) (Melnychuk et al., 2022). These methods attempt to alleviate time-varying confounding via balanced representations. However, balancing was originally designed for reducing finite-sample estimation variance and *not* for mitigating confounding bias (Shalit et al., 2017). Hence, such methods act as heuristics without a theoretical justification. The difficulty of enforcing balanced representations may even introduce further confounding bias (Melnychuk et al., 2024). Unlike these methods, our DeepBlip allows for proper adjustments and is thus unbiased.

• *Non-SNMM-based adjustment strategies:* Other neural methods build on frameworks from statistics such as marginal structural models (MSMs) (Robins et al., 2000; Orellana et al., 2010) and G-computation (Robins, 1986; Robins et al., 1999). Examples are: ① **G-Net** (Li et al., 2021) and **IGC-Net** (Hess et al., 2026a), which are both based on G-computation and thus compute nested conditional expectations over time. Hence, these methods require the entire counterfactual outcome trajectories and thus model the full data-generating process of covariates and outcomes, which becomes exponentially more complex as time horizons grow. ② MSMs-based methods like **R-MSNs** (Lim, 2018) and the **DR-learner** (for time-varying settings) (Frauen et al., 2025) use inverse propensity weighting (IPW) to re-weight outcomes as in a randomized control trial. In time-varying settings, the propensity is a multiplication of a sequence of probabilities, which has known to become unstable when the horizon is large. In sum, methods from both streams can often become *unstable* over long time horizons (different from SNMMs that break the CATE into localized effects at each time step).

• *SNMM-based adjustment strategies:* SNMMs (Robins, 1994; 2004) offer a principled framework for estimating CATE over time by directly estimating incremental treatment effects via so-called blip functions. In principle, SNMMs could address both of the above limitations; however, SNMMs are only an abstract theoretical foundation, ***not*** an off-the-shelf model that can be directly applied. Early implementations (Robins, 1994; 2004) relied on strong parametric assumptions (e.g., linearity) and were limited to short time horizons.

(Xu et al., 2016; Schulam & Saria, 2017; Soleimani et al., 2017), yet these approaches impose strong assumptions on the outcomes, and their scalability is limited. For these reasons, we focus on neural methods, which offer better flexibility and scalability for complex, high-dimensional medical data.

Despite the theoretical appeal of SNMMs, a neural implementation is missing. Closest to our work is Dynamic DML (Lewis & Syrgkanis, 2021), which builds upon SNMMs but still has key *limitations*. The original implementation is provided for *linear* models, which cannot capture complex, non-linear relationships. In principle, the framework in (Lewis & Syrgkanis, 2021) is model-agnostic and *could* be used along with other deep learning models. Yet, this is impractical because essentially *a different network must be estimated for each time step*, so the complexity *grows linearly with the prediction horizon*. Because of this, extending Dynamic DML into a scalable neural framework is *non-trivial:* the sequential g-estimation of SNMM (as used by Dynamic DML) is iterative and essentially prevents efficient gradient-based and thus end-to-end optimization. As a remedy, we later propose a new *double optimization trick*.

**Research gap:** To the best of our knowledge, there is no neural implementation of SNMMs. For this, we introduce a new neural architecture together with a new *double optimization trick* to leverage the blip function from SNMMs for CATE estimation over time.

## 3. Problem Formulation

**Setup:** We follow the standard setup (Bica et al., 2020; Melnychuk et al., 2022; Hess et al., 2026a; Frauen et al., 2025) for estimating CATE over time $t \in \{1, 2, \ldots, T\} \subset \mathbb{N}$ given by: (1) the target outcome $Y_t \in \mathbb{R}$; (2) time-varying covariates $X_t \in \mathcal{X} \subset \mathbb{R}^{d_x}$; and (3) treatment $A_t \in \mathcal{A} \subset \mathbb{R}^{d_a}$ that can be either discrete or continuous. We assume w.l.o.g. that the static features (e.g., age, sex) are included in the covariate.

**Notation:** To simplify notation, we use overlines to denote the full sequence of a variable (e.g., $\overline{X}_t = (X_1, \ldots, X_t)$). We refer to a sequence of variables that starts at $t$ and ends at $t + \tau$ via $A_{t:t+\tau} = (A_t, A_{t+1}, \ldots, A_{t+\tau})$. We use the lowercase letter to denote a realization of a random variable (e.g., $X_t = x_t$). We use an asterisk $*$ to indicate a constant quantity (e.g., a fixed treatment $a_t^*$). We denote the patient history by $H_t = (\overline{X}_t, \overline{A}_{t-1}, \overline{Y}_{t-1})$.

**CATE estimation over time:** We build upon the potential outcome framework (Rubin, 1972) for the time-varying setting (Robins et al., 1999). We aim to estimate the CATE over time between two treatment sequences for a given patient history, i.e.,

$$\mathbb{E}\left[Y_{t+\tau}^{(a_{t:t+\tau}^*)} - Y_{t+\tau}^{(b_{t:t+\tau}^*)} \mid H_t = h_t\right], \quad 0 \leq t \leq T - \tau, \tag{1}$$

where $Y_{t+\tau}^{(a_{t:t+\tau}^*)}$ and $Y_{t+\tau}^{(b_{t:t+\tau}^*)}$ represent the $\tau$-step-ahead potential outcomes under interventions $\mathrm{do}(A_{t:t+\tau} = a_{t:t+\tau}^*)$ and $\mathrm{do}(A_{t:t+\tau} = b_{t:t+\tau}^*)$, respectively (see Appendix A for

a formal definition of potential outcomes and interventions).

**Identifiability:** We make the following identifiability assumptions (Robins et al., 2000; Petersen & van der Laan, 2014) that are standard in the time-varying setting (Lim, 2018; Bica et al., 2020; Melnychuk et al., 2022; Li et al., 2021): (1) *Consistency*: The potential outcome under the intervention by the observed treatment equals the observed outcome, namely, $Y_t^{(A_t)} = Y$. (2) *Overlap*: Given an observed history $H_t = h_t$, if $\Pr(H_t = h_t) > 0$, then any possible treatment has a positive probability of being received: $\forall a \in \mathcal{A}, \Pr(A_t = a \mid H_t = h_t) > 0$. (3) *Sequential ignorability*: The potential outcome under an arbitrary intervention is independent of the treatment assignment conditioned on the history, i.e., $Y_t^{(a_t^*)} \perp A_t \mid H_t = h_t$.

However, estimating the CATE over time is non-trivial due to *time-varying confounding* (Coston et al., 2020; Hess et al., 2026a). In the time-varying setting, covariates act as confounders because they are influenced by earlier treatments and then affect later treatments and outcomes. This means they cannot be adjusted for naïvely as in the static setting without introducing bias (see Appendix A.2). In this paper, we address this issue via SNMMs.

**Blip function:** SNMMs model the incremental effect of treatments (which are called "blips") at time $t + k$ on the mean outcome at $t + \tau$, given observed patient history $H_t$ (Vansteelandt & Joffe, 2014). These "blips" accumulate over time into the total treatment effect and thus allow to rigorously adjust for time-varying confounding (Robins, 2004) (see Thm 3.1 below). Formally, let $d$ be a treatment policy. The blips are defined via a **blip function** (Robins, 1994; 2004):

$$
\begin{aligned}
\gamma_{t,k}\big(\bar{x}_{t+1:t+k}, \bar{a}_{t:t+k}\,;h_t\big) &= \mathbb{E}\Big[Y_{t+\tau}^{(a_{t:t+k},\, d_{t+k+1:t+\tau})} \\
&\quad - Y_{t+\tau}^{(a_{t:t+k-1},\, 0,\, d_{t+k+1:t+\tau})}\Big|\, a_{t:t+k}, x_{t+1:t+k}, h_t\Big].
\end{aligned}
\quad (2)
$$

Intuitively, the blip function $\gamma_{t,k}$ isolates the causal effect of each treatment decision *locally*. By isolating the local effect of each treatment, this formulation avoids modeling the full outcome trajectory, making SNMMs often stable over long horizons.

**Theorem 3.1.** *Adjustment via blip functions (Theorem 3.1 from Robins (2004))* *Given a policy $d$ between time $t$ and $t + \tau$, the following identification holds under the sequential ignorability assumption (Appendix C):*

$$
\mathbb{E}\Big[Y^{(d)}\Big| H_t = h_t\Big] \quad (3)
$$
$$
= \mathbb{E}\Big[Y + \sum_{k=0}^{\tau}\big(\gamma_{t,k}\big(X_{t+1:t+k}, (A_{t:t+k-1}, d_{t+k})\,;h_t\big) \\
- \gamma_{t,k}\big(X_{t+1:t+k}, A_{t:t+k}\,;h_t\big)\big)\Big| H_t = h_t\Big]
$$

The theorem demonstrates that SNMMs adjust for time-varying confounding by making local adjustments at each step. Furthermore, if we can correctly estimate the blip functions, the CATE over time estimation will converge to the unbiased causal effect given sufficiently large data.

## 4. Our DeepBlip Framework

In this section, we present DeepBlip. First, we introduce how we learn the CATE via blip functions using a neural parameterization (Sec. 4.1), then introduce our $L^2$-moment loss (Sec. 4.2), our model architecture (Sec. 4.3), and the training and inference procedure (Sec. 4.4).

### 4.1. Learning the CATE via blip functions

**Overview:** Our DeepBlip leverages Eq. (4) to adjust for time-varying confounding. Our task thus reduces to estimating the blip functions – in particular, so-called *blip coefficients* that parametrize the blip functions. However, we do **not** attempt to estimate the coefficients via the iterative g-estimation. Instead, we optimize a $L^2$-moment loss that directly predicts the blip coefficients and which allows us to estimate Eq. (4) more efficiently.

**Parameterization trick:** We first explain how we estimate the CATE via the blip function. For this, we adopt a similar parametrization for the blip function as in (Lewis & Syrgkanis, 2021), namely, $\gamma_{t,k}\big(\bar{x}_{t+1:t+k}, \bar{a}_{t:t+k}\,;h_t\big) = \psi_{t,k}\big(h_t\big)' a_{t+k}$, but where $\psi_{t,k}$ is a **neural network**. [5] Under identifiability assumptions and the parametrization for $\gamma_{t,k}$ defined above, the CATE of $a^*$ against $b^*$ for any two treatment sequences $a^*, b^* \in \mathbb{R}^{(\tau+1)\cdot d_a}$ is (see (Lewis & Syrgkanis, 2021) for a formal derivation):

$$
\mathbb{E}\big[Y_{t+\tau}^{(a^*)} - Y_{t+\tau}^{(b^*)} \mid H_t = h_t\big] = \sum_{k=0}^{\tau} \psi_{t,k}(h_t)'\big(a_{t+k}^* - b_{t+k}^*\big).
\quad (4)
$$

We refer to $\psi_{t,k}(h_t)$ as the conditional *blip coefficients* of the blip function $\gamma_{t,k}$.

*Why do we need a tailored architecture and learning algorithm?* A key component of our framework is that the function $\psi_{t,k}(h_t)$ is parameterized by a sequential neural network (e.g., LSTM or transformer). This is a crucial difference from traditional SNMMs, which were developed for estimating the ATE over a fixed number of time steps (i.e. $\mathcal{H}_t = \emptyset \wedge t \equiv 0 \wedge T \equiv \tau$) and where, as a result, blip coefficients are constants. These constants are typically estimated through *iteratively* solving a set of moment equations

---

[5]This formulation is chosen mainly for clarity. In fact, it is readily extended to capture nonlinear treatment interactions by incorporating a high-dimensional, nonlinear feature map $\phi(\cdot)$. The blip function can then be parametrized as $\gamma_{t,k}(h_t) = \psi_{t,k}'(h_t)\phi(a_{t+k})$.

via g-estimation (Robins, 1994; 2004; Vansteelandt & Joffe, 2014) (see Appendix C.1). However, such an approach is not compatible with neural network-based learning. In contrast, DeepBlip introduces a tailored neural architecture (Sec. 4.3) that we can train via gradient-based optimization (Sec. 4.2).

## 4.2. $L^2$-moment loss

We adapt the moment-based iterative minimization from (Lewis & Syrgkanis, 2021) to our history-conditioned setting. For $k = \tau, \ldots, k = 0$, at each time step $k$, we aim to find the minimizer $\psi_{t,k}^*(\cdot)$, which is a function that maps the history $h_t$ to the blip coefficients, via

$$\psi_{t,k}^* = \operatorname*{arg\,min}_{\widehat{\psi}_{t,k}(\cdot) \in \Phi_{t,k}} \mathbb{E}\Big[\Big(\widetilde{Y}_{t,k} - \sum_{j=k+1}^{\tau} \psi_{t,j}^*(h_t)' \widetilde{A}_{t,j,k} - \widehat{\psi}_{t,k}(h_t)' \widetilde{A}_{t,k,k}\Big)^2\Big], \quad (5)$$

where $\Phi_{t,k}$ is the function space for the blip coefficient predictors and where

$$\widetilde{Y}_{t,k} = Y_{t+\tau} - \mathbb{E}\big[Y_{t+\tau} \mid H_{t+k} = h_{t+k}\big], \quad (6)$$

$$\widetilde{A}_{t,j,k} = A_{t+j} - \mathbb{E}\big[A_{t+j} \mid H_{t+k} = h_{t+k}\big]. \quad (7)$$

are residuals of the conditional means for $Y$ and $A$, respectively. We name the target as the $L^2$-**moment loss** for step $k$ and denote it as $\mathcal{L}_k$. Notice that only one $\widehat{\psi}_{t,k}(\cdot)$ is solved by minimizing $\mathcal{L}_k$; hence, in order to solve the entire task (i.e., all the $\widehat{\psi}_{t,k}, (0 \leq k \leq \tau)$), one would normally need to conduct a series of minimizations **sequentially** from $k = \tau$ down to $k = 0$ such as in Dynamic DML from (Lewis & Syrgkanis, 2021).

**Double optimization trick for our $L^2$-moment loss:** In order to find $\psi_{t,k}^*$, all the previous blip predictors $\psi_{t,j}^*, j \geq k$ are required. To avoid solving $\psi_{t,k}^*$ sequentially, we propose a *double optimization trick* that allows *simultaneous* training of all the blip predictors: During each iteration, first, the blip predictor $\widehat{\psi}_t$ makes two forward passes to generate two sets of the blip coefficients $\widehat{\psi}_t^1(h_t)$ and $\widehat{\psi}_t^2(h_t)$. Then, $\widehat{\psi}_{t,j}^2(h_t)$ serves as a fixed, "pseudo-ground-truth" target for the future blip effects in Eq. (5) and is detached from the computation graph. For $k = 0, \ldots, \tau$, the adapted $L^2$-moment loss at step $k$ is then given empirically by

$$\mathcal{L}_{\text{blip}}^k = \frac{1}{n(T-\tau)} \sum_{t=1}^{T-\tau} \sum_{i=1}^{n} \Big(\widetilde{Y}_{t+k}^i - \sum_{j=k+1}^{\tau} \widehat{\psi}_j^2(H_t^i)' \widetilde{A}_{t,j,k}^i$$

$$\quad (8)$$

$$- \widehat{\psi}_k^1(H_t^i)' \widetilde{A}_{t,k,k}^i\Big)^2.$$

**A gradient-level explanation of double optimization:** Consider the case $\tau = 2$. Without detachment, all blip predictors are updated simultaneously from the combined loss

$\mathcal{L}_{\text{blip}} = \sum_{k=0}^{2} \mathcal{L}_{\text{blip}}^k$, so the gradient for the final predictor contains both the correct term and spurious terms from earlier losses, i.e., $\nabla_{\psi_2} \mathcal{L}_{\text{blip}} = \nabla_{\psi_2} \mathcal{L}_{\text{blip}}^2 + \nabla_{\psi_2} \mathcal{L}_{\text{blip}}^1 + \nabla_{\psi_2} \mathcal{L}_{\text{blip}}^0$. The latter two terms push $\psi_2$ to compensate for the current, noisy estimates of $\psi_0$ and $\psi_1$, which creates biased training signals. In our double optimization trick, the future blip predictions that enter earlier losses are detached; hence, $\nabla_{\psi_2} \mathcal{L}_{\text{blip}} = \nabla_{\psi_2} \mathcal{L}_{\text{blip}}^2$, and each blip predictor receives gradients only from its own moment loss. Appendix D provides the formal convergence argument.

**Why is our double optimization trick necessary for scalability?** Existing SNMM methods, like Dynamic DML (Lewis & Syrgkanis, 2021), rely on a sequential scheme that solves for blip coefficients one at a time. For neural networks, this would require *training a separate model for each of the $\tau + 1$ time steps, which is computationally prohibitive*. In contrast, our double optimization trick *bypasses this sequential dependency*, allowing us to instead optimize a single, unified objective: $\mathcal{L}_{\text{blip}} = \sum_{k=0}^{\tau} \mathcal{L}_{\text{blip}}^k$. This makes our framework significantly more scalable, as a neural version of the sequential approach would multiply the total training time by a factor of at least $\tau + 1$ (see Figure 4).

We show in Appendix D that the double optimization trick is a single-step gradient descent realization of an iterative scheme for the complete blip predictor $\widehat{\psi}$. Under this scheme, all the intermediate blip predictors $\{\widehat{\psi}_j \mid (j < \tau)\}$ converge to the true value as long as the final predictor $\widehat{\psi}_\tau$ converges to the true value. Further, the convergence of the final predictor $\widehat{\psi}_\tau$ is guaranteed by the $L^2$-moment loss being the correct objective, since $\mathcal{L}_{\text{blip}}^\tau = \mathcal{L}_\tau$. Therefore, all the blip predictors will converge under the double optimization trick. We further perform ablation studies on the double optimization trick in Appendix E.1 and E.2 to empirically show its effectiveness.

**Theoretical properties:** Below, we state that our loss satisfies *Universal Neyman-orthogonality* (see (Foster & Syrgkanis, 2019) for a formal definition), which ensures double robustness. This means that the target loss is *robust* against perturbations of the nuisance functions (Chernozhukov et al., 2018; Kennedy, 2023). The remark below follows from the theory in Lewis & Syrgkanis (2021), which can be extended to our double optimization trick (see Appendix C.2 for a formal derivation, respectively).

*Remark* 4.1. *Our moment loss is Neyman-orthogonal. This makes second-stage blip estimation insensitive to first-order errors in the nuisance predictions, so moderate nuisance misspecification affects the loss gradient only at second order.*

## 4.3. Model architecture

DeepBlip works in two stages (see Fig. 2): • **Stage** ① (*nuisance network*): models the nuisance functions to estimate the residuals in Eq. (6). • **Stage** ② (*blip prediction network*): estimates the blip coefficients given the observed history $h_t$. The neural networks in both stages have a similar structure: (i) a *sequential encoder* that encodes the observed history $H_t$, and (ii) multiple *prediction heads* that take the encoded history as input to predict the targets.

**Why we need a two-stage design:** To construct the $L^2$-moment loss defined Eq. (5), we need the variables $\widetilde{Y}_{t,k}$, $\widetilde{A}_{t,j,k}$, defined in Eq. (6) (see Appendix C.1 for details). Based on the definitions, we must compute the *residuals* between the outcome variable and the regressed means *before* we optimize $\mathcal{L}_{\text{blip}}^k$. We thus follow previous literature (Lewis & Syrgkanis, 2021; Kennedy, 2023; Frauen et al., 2025) and treat the conditional expectations $\mathbb{E}\big[Y_{t+\tau} \mid H_{t+k} = h_{t+k}\big]$ and $\mathbb{E}\big[A_{t+j} \mid H_{t+k} = h_{t+k}\big]$ as *nuisance functions*, so that we first estimate the nuisance function (Stage ①) and then train the blip prediction network to minimize the $L^2$-moment loss (Stage ②).

**Neural backbone:** Our DeepBlip is flexible and allows for different neural backbones (e.g., LSTM or transformer). These are necessary to capture the patient history: We notice that the networks at both Stage ① and ② take the history variable $H_t = (\overline{X}_t, \overline{A}_{t-1}, \overline{Y}_{t-1}) \in \mathcal{H}_t$ as input. Hence, both stages can be written as a function $f : \cup_{t=1}^{T} \mathcal{H}_t \to \mathbb{R}^c$, where $c$ is the number of outputs. However, $\dim(H_t)$ varies over time, which makes $H_t$ not suitable as a direct input to a neural network. A standard way to handle this is by using a sequential model to iteratively take the inputs $(X_t, A_{t-1}, Y_{t-1})$ and then maintain a vector $Z_t \in \mathbb{R}^{d_z}$ with fixed dimension that encodes all the necessary information (Lim, 2018; Li et al., 2021; Melnychuk et al., 2022; Hess et al., 2026a). Here, we thus use LSTMs and transformers (see details of the architectures in Appendix I). Finally, we stress that each stage uses a *separate* encoder: $\mathcal{E}_{\theta_N}^N$ for Stage ①, and $\mathcal{E}_{\theta_B}^B$ for Stage ② ($N$ for **N**uisance and $B$ for **B**lip) with different model weights $\theta_N$ and $\theta_B$.

**Stage ①: Nuisance network.** The nuisance network $(\mathcal{E}_{\theta_N}^N, \{\text{gp}_{\theta_N}^k\}_{k=0}^{\tau}, \{\text{gq}_{\theta_N}^{j,k}\}_{0\le k\le j\le\tau})$ consists of a sequential encoder $\mathcal{E}_{\theta_N}^N$ and a collection of prediction heads $\{\text{gp}_{\theta_N}^k\}_{k=0}^{\tau}, \{\text{gq}_{\theta_N}^{j,k}\}_{0\le k\le j\le\tau}$. The nuisance networks are responsible for computing the following nuisance functions:

$$p_{t,k}(h_{t+k}) := \mathbb{E}\big[Y_{t+\tau} \mid H_{t+k} = h_{t+k}\big], 0 \le k \le \tau \quad (9)$$

$$q_{t,j,k}(h_{t+k}) := \mathbb{E}\big[A_{t,j} \mid H_{t+k} = h_{t+k}\big], 0 \le k \le j \le \tau \quad (10)$$

For a patient with history $H_t$ and subsequent covariates $X_{t+1:t+k}, A_{t:t+k-1}$, we proceed as follows: First, the encoder $\mathcal{E}_{\theta_N}^N$ processes history $H_{t+k}$ to produce a representa-

tion $Z_{t+k}^N = \mathcal{E}_\theta^N(H_{t+k})$. Then, the prediction heads receive $Z_{t+k}^N$ to compute the regressed outcomes for the nuisance functions via:

$$\text{gp}_{\theta_N}^k(Z_{t+k}^N) = \widehat{p}_{t,k}(H_{t+k}), \quad \text{gq}_{\theta_N}^{j,k}(Z_{t+k}^N) = \widehat{q}_{t,j,k}(H_{t+k})$$

where $Z_{t+k}^N = \mathcal{E}_{\theta_N}^N(H_{t+k})$. Third, the residuals are computed via

$$\widetilde{Y}_{t,k} \approx Y_{t+\tau} - \text{gp}_{\theta_N}^k(Z_{t+k}^N), \quad \widetilde{A}_{t,j,k} \approx A_{t+j} - \text{gq}_{\theta_N}^{j,k}(Z_{t+k}^N)$$

**Stage ②: Blip prediction network.** The blip prediction network $(\mathcal{E}_{\theta_B}^B, \{\text{gb}_{\theta_B}^k\}_{k=0}^{\tau})$ is responsible for predicting the blip coefficients $\boldsymbol{\psi_t}(h_t) = (\psi_{t,0}, \dots, \psi_{t,\tau}) \in \mathbb{R}^{r(\tau+1)}$ as described in Eq. ((5)). Here, we proceed as follows. First, the sequential encoder $\mathcal{E}_\theta^B$ (B for **B**lip) processes the patient's history $H_t$ into a representation $Z_t^B = \mathcal{E}_\theta^B(h_t)$. Then, for each horizon $k \in \{0, 1, \dots, \tau\}$, the prediction head $\text{gb}_{\theta_B}^k$ maps $Z_t^B$ onto the corresponding blip coefficient:

$$\widehat{\psi}_{t,k}(H_t) = \text{gb}_{\theta_B}^k(Z_t^B) \sim \psi_{t,k}(H_t) \in \mathbb{R}^r. \quad (11)$$

## 4.4. Training and Inference

Taken together, the training procedure of DeepBlip now follows two steps (see Fig. 2): (1) train the nuisance networks with $W$-fold cross-fitting and compute held-out residuals, and (2) train the blip prediction network on these residuals. In contrast, inference with DeepBlip is highly efficient as it involves *only* the second-stage blip prediction network. Details are below. We provide the pseudocode in Alg. 1 and Alg. 2 in the appendix.

**Step ①: Train nuisance network.** The nuisance network is trained to predict nuisance functions $p_{t,k}(h_{t+k})$ and $q_{t,j,k}(h_{t+k})$ simultaneously. Since $p_{t,k}(h_{t+k})$ is the conditional expectation of real outcome $Y_{t+\tau} \in \mathbb{R}$, we use the squared error loss $\mathcal{L}_p = \frac{1}{(T-\tau)(\tau+1)} \sum_{t=1}^{T-\tau} \sum_{k=0}^{\tau} (\text{gp}_{\theta_N}^k(Z_{t+k}^N) - Y_{t+\tau})^2$. For $q_{t,j,k}(h_{t+k})$, which denotes the treatment response, we proceed for the $i$-th treatment in $A_{t+j} \in \mathbb{R}^{d_a}$ as follows. If $(A_{t+j})_i$ is a continuous variable, then we apply the squared loss: $\mathcal{L}_{q,i} = \frac{2}{(T-\tau)(\tau+1)(\tau+2)} \sum_{t=1}^{T-\tau} \sum_{0\le k\le j\le\tau} \big(\text{gq}_{\theta_N}^{k,j}(Z_{t+k}^N)_i - (A_{t+j})_i\big)^2$. If $(A_{t+j})_i$ is a binary variable, then we apply the binary cross entropy loss $\mathcal{L}_{q,i} = \frac{2}{(T-\tau)(\tau+1)(\tau+2)} \sum_{t,k} \text{BCE}\big((A_{t+j})_i, \text{gq}_{\theta_N}^{k,j}(Z_{t+k}^N)_i\big)$. For categorical variables with more than 2 classes, we preprocess the variable into a one-hot vector of binary variables. Since the network predicts these targets simultaneously, we update the parameter $\theta_N$ by backpropagating the sum of all the losses discussed above, i.e., $\mathcal{L}_N = \mathcal{L}_p + \frac{1}{d_a} \sum_{i=1}^{d_a} \mathcal{L}_{q,i}$

**Step ②: Train blip prediction network.** After having trained the nuisance network, we freeze its parameters and

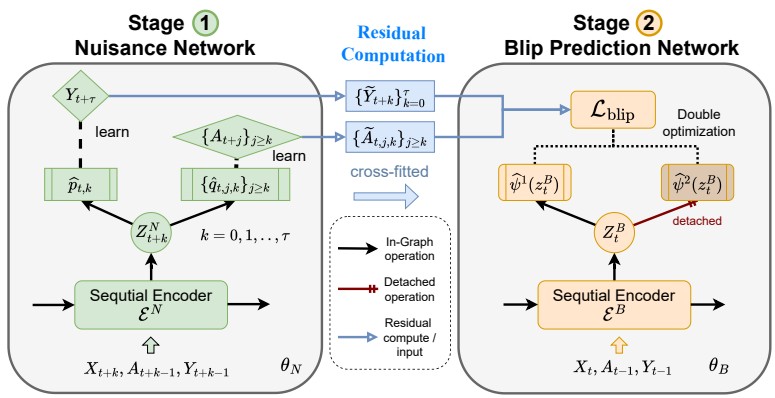

*Figure 2.* **Neural architecture of the two-stage DeepBlip framework.**

*Figure 3.* **Results for tumor growth dataset.** Normalized RMSE (averaged over 5 runs) of CATE predictions against ground-truth over growing confounding. Here: $\tau = 2$

then compute the residuals of each sample as in Eq. (6) using held-out predictions from the cross-fitted nuisance models. These residuals are then used for the $L^2$-moment loss. To accelerate the training process, we adopt the double optimization trick from above: For $k = 0, \ldots, \tau$, we perform two forward passes that create two predictions $\widehat{\psi}^1_{t,k}(H_t)$ and $\widehat{\psi}^2_{t,k}(H_t)$. The latter is then *detached* from the computation graph before feeding into the adapted $L^2$-moment loss at step $k$. The final loss target is then given by: $\mathcal{L}_{\text{blip}} = \sum_{k=0}^{\tau} \mathcal{L}_{\text{blip}}^k$.

*Remark* 4.2. *Under standard assumptions, the output of our DeepBlip has a mean squared error guarantee:*

$$\max_t \max_{k \leq \tau} \mathbb{E}\left[\left\|\widehat{\psi}_{t,k} - \psi_{t,k}\right\|_{2,2}^2\right] = O\left(\left(\frac{\log\log(n)}{n}\right)^2\right) \tag{12}$$

*The insight is that, once the nuisance functions and neural blip class satisfy the regularity conditions, the blip coefficient error vanishes at a near-parametric rate. Details are in Appendix C.3).*

**Inference at runtime:** Once trained, our DeepBlip predicts the CATE over time (i.e., $\mathbb{E}[Y_{t+\tau}^{(a^*)} - Y_{t+\tau}^{(b^*)} \mid H_t = h_t]$) through only the blip prediction network:

$$\sum_{k=0}^{\tau} \text{gb}_{\theta_B}^k(z_t^B)'(a_k^* - b_k^*), \quad \text{where} \quad z_t^B = \mathcal{E}_{\theta_B}^B(h_t) \tag{13}$$

**Efficient offline evaluation:** Once we have estimated the blip coefficients, then we can instantly identify the treatment sequence $a^*$ with the best effect compared to the baseline $b^*$ (e.g., a treatment sequence with no interventions). The reason is that the blip coefficients do *not* depend on treatments. Hence, our DeepBlip is much more efficient for evaluating the personalized effects of different treatment sequences compared to existing methods that require re-computation (Lim, 2018; Bica et al., 2020; Melnychuk et al., 2022) or

even re-training (Li et al., 2021; Hess et al., 2026a). This is highly relevant in personalized medicine where clinicians and patients jointly reason about different treatment strategies (Feuerriegel et al., 2024).

**Implementation details.** We instantiate our DeepBlip with a transformer architecture (see Appendix I). We also provide a variant based on an LSTM, which, despite the simpler architecture, is still highly competitive (see Appendix E.1).

# 5. Experiments

Our experiments primarily serve two purposes to demonstrate DeepBlip is an effective neural framework for SN-MMs: (i) we aim to show that our DeepBlip performs adequate adjustment for time-varying confounding, which is a key property of SNMMs; and (ii) we aim to show that our DeepBlip is especially robust for longer time horizons, which is a natural strength of SNMMs.

**Baselines:** We demonstrate the performance of our Deep-Blip against key baselines from the literature (see Table 1) for the task of estimating CATE (or conditional average potential outcomes) on medical datasets. Descriptions of the baseline methods are available in Appendix G. We select the HA-PI-learner from Frauen et al. (2025) instantiated by transformer (named **HA-TRM**) as a naïve baseline. We provide additional implementation details – including architecture choices, training procedures, and hyperparameter tuning – in Appendix I. To ensure a fair comparison, all methods – including baselines – use the **same** neural backbone architecture, so any performance differences can be attributed to the respective learning frameworks. All results are averaged over 5 runs.

**Ablations:** We include ablation studies of both the architecture and the double optimization trick in Appendix E.1. We also validate our component-wise blip coefficient estimates in Appendix E.2.

*Table 2.* **MIMIC-III with longer time horizons $\tau$.** Normalized RMSE (mean $\pm$ std. dev. over 5 runs) for $\tau$-step-ahead CATE estimation on the MIMIC-III dataset. We highlight the relative improvement over the best-performing baseline.

| | $\tau = 3$ | $\tau = 4$ | $\tau = 5$ | $\tau = 6$ | $\tau = 7$ | $\tau = 8$ | $\tau = 9$ | $\tau = 10$ |
|---|---|---|---|---|---|---|---|---|
| HA-TRM (naïve) | $0.89 \pm 0.03$ | $0.97 \pm 0.04$ | $1.02 \pm 0.10$ | $1.42 \pm 0.20$ | $1.92 \pm 0.40$ | $2.57 \pm 0.44$ | $2.58 \pm 0.56$ | $3.11 \pm 0.72$ |
| R-MSNs | $0.98 \pm 0.17$ | $1.12 \pm 0.21$ | $1.25 \pm 0.28$ | $1.65 \pm 0.57$ | $2.25 \pm 1.02$ | $2.85 \pm 1.18$ | $3.20 \pm 1.42$ | $3.55 \pm 1.50$ |
| CRN | $0.66 \pm 0.11$ | $0.82 \pm 0.12$ | $1.05 \pm 0.22$ | $1.22 \pm 0.35$ | $1.43 \pm 0.33$ | $1.62 \pm 0.42$ | $1.83 \pm 0.43$ | $2.04 \pm 0.54$ |
| CT | $0.64 \pm 0.12$ | $0.79 \pm 0.11$ | $1.01 \pm 0.18$ | $1.18 \pm 0.33$ | $1.77 \pm 0.52$ | $1.85 \pm 0.49$ | $1.99 \pm 0.63$ | $1.98 \pm 0.60$ |
| G-Net | $0.58 \pm 0.08$ | $0.73 \pm 0.12$ | $1.05 \pm 0.25$ | $1.38 \pm 0.40$ | $1.75 \pm 0.60$ | $2.15 \pm 0.80$ | $2.55 \pm 0.90$ | $3.12 \pm 1.05$ |
| IGC-Net | $0.52 \pm 0.02$ | $0.63 \pm 0.08$ | $0.75 \pm 0.17$ | $0.85 \pm 0.13$ | $0.95 \pm 0.26$ | $1.10 \pm 0.34$ | $1.25 \pm 0.37$ | $1.50 \pm 0.45$ |
| Dynamic DML | $0.72 \pm 0.15$ | $0.88 \pm 0.23$ | $1.12 \pm 0.27$ | $1.34 \pm 0.68$ | $1.67 \pm 0.43$ | $1.98 \pm 0.55$ | $2.33 \pm 0.57$ | $2.46 \pm 0.72$ |
| DeepBlip (**ours**) | $\mathbf{0.50 \pm 0.12}$ | $\mathbf{0.58 \pm 0.16}$ | $\mathbf{0.64 \pm 0.19}$ | $\mathbf{0.75 \pm 0.21}$ | $\mathbf{0.86 \pm 0.24}$ | $\mathbf{0.89 \pm 0.27}$ | $\mathbf{0.95 \pm 0.28}$ | $\mathbf{0.98 \pm 0.32}$ |
| Improvement | 3.8% | 7.9% | 14.7% | 11.7% | 9.5% | 19.1% | 24.0% | 34.6% |

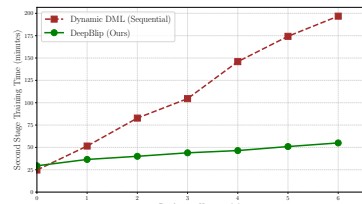

*Figure 4.* **Scalability of Dynamic DML vs. DeepBlip.** Shown is the training time.

### 5.1. Tumor growth dataset

**Setting:** We use the pharmacokinetic-pharmacodynamic tumor growth dataset (Geng et al., 2017), which is commonly used for benchmarking CATE methods over time (Lim, 2018; Bica et al., 2020; Li et al., 2021; Melnychuk et al., 2022; Hess et al., 2026a). The dataset describes the time-varying effects of chemotherapies and radiotherapies on cancer volumes. The treatment assignments depend on previous outcomes, subject to time-varying confounding. The amount of confounding is controlled by the simulation parameter $\gamma_{\text{conf}}$. Details are in Appendix F.1.

**Results:** Figure 3 shows the average RMSE of CATE against increasing confounding $\gamma_{\text{conf}}$ and under $\tau = 2$. Our **DeepBlip** outperforms all baselines for $\gamma_{\text{conf}} \geq 2$. This matches the purpose of our method to deal with time-varying confounding. More importantly, our DeepBlip achieves large performance gains under strong confounding levels ($\gamma_{\text{conf}} > 6$). This highlights that our DeepBlip is robust against time-varying confounding by providing adequate adjustment for time-varying confounding.

We could further make the following observations: ① The MSM-based **R-MSN** performs poorly across all confounding levels and even has a higher RMSE than HA-LSTM for $\gamma_{\text{conf}} \leq 6$. This aligns with the inherent instability of inverse propensity weighting in MSMs, which was the motivation for our method. ② Baselines like **CT** and **CRN** that use balanced representation (in orange) are ineffective since this technique does not adjust for confounding. ③ G-computation-based methods like **G-Net** and **IGC-Net** show slightly lower RMSE for $\gamma_{\text{conf}} \leq 1$ but still perform significantly worse than our DeepBlip for $\gamma_{\text{conf}} \geq 6$. We attribute this to the fact that the learning is unstable, which we empirically verify in the following by varying the prediction horizon $\tau$. ④ Dynamic DML is competitive, thereby showing that SNMMs are a suitable modeling framework for this task, while our DeepBlip is better by a clear margin.

### 5.2. MIMIC-III semi-synthetic dataset

**Setting:** Next, we evaluate the performance for longer prediction horizons $\tau$ on semi-synthetic data. We build upon

MIMIC-III (Johnson et al., 2016), a widely used benchmark for evaluating CATE over time (Bica et al., 2020; Melnychuk et al., 2022; Hess et al., 2026a). Following previous literature (Melnychuk et al., 2022; Hess et al., 2026a; Schulam & Saria, 2017), we extract patient vitals from MIMIC-III and then simulate the patient outcome over time with the mixed dynamics of exogenous dependency, endogenous dependency, and treatment effects combined. Treatments are assigned based on previous outcomes and patient vitals, which again, introduces time-varying confounding (see Appendix F.2).

**Results:** Table 2 shows the average RMSE (with std. dev.) over five different runs with $\gamma_{\text{conf}} = 1$ and varying prediction horizon $\tau$. Our **DeepBlip** consistently achieves lower RMSE than all baselines for $\tau \geq 3$, and the performance gap grows with the horizon: for $\tau = 10$, DeepBlip achieves $\sim 34\%$ improvement over the second-best model (IGC-Net), highlighting its temporal stability.

We further make the observations that all baselines either struggle with high-dimensional covariates or become unstable as $\tau$ increases. ① The MSM-based method (**R-MSNs**) exhibits the highest variance across all $\tau$ because inverse propensity weighting produces unstable weights. ② Methods like **CRN** and **CT** perform better due to the way they handle high-dimensional covariates. However, both **CRN** and **CT** are known to be biased and thus inferior to IGC-Net and our DeepBlip. ③ G-computation-based methods (i.e., **G-Net** and **IGC-Net**) achieve a lower RMSE than the other baselines due to proper adjustments, but still are not as stable as our method. This is because G-computation accumulates error over time due to modeling nested expectations. ④ The performance gain of DeepBlip over Dynamic DML is large, which can be attributed to the complexity of the dataset, especially in light of the long time windows. IGC-Net can outperform Dynamic DML because IGC-Net uses a transformer to encode the full patient history, whereas Dynamic DML is constrained to fixed-vector histories and sequential regressions for the blip coefficients.

**Scalability of DeepBlip:** To demonstrate the computational advantage of our framework, we tested the second-stage training time of both **Dynamic DML** and our **Deep-**

**Blip** over increasing $\tau$ on the MIMIC-III semi-synthetic dataset. The results are in Figure 4: we observed a large difference in scalability. The training time for Dynamic DML grows linearly with $\tau$, due to its sequential scheme, which requires training a separate model at *each* time step. In contrast, the runtime for DeepBlip remains *nearly constant*, which confirms the efficiency of our double optimization trick and thus demonstrates that our DeepBlip overcomes the scalability bottleneck of SNMMs.

## 5.3. MIMIC-III real-world dataset

**Setting:** We apply DeepBlip directly to the real MIMIC-III dataset to assess practical applicability. Using the same 25-dimensional feature space (Appendix F.2), we take mean arterial pressure (MAP) as the target outcome and estimate the time-varying causal effect of sequential vasopressor treatments. Since ground-truth CATEs are unavailable in observational data, we validate the estimates by checking whether the blip coefficients and cumulative CATE align with established clinical knowledge.

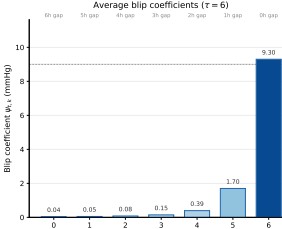
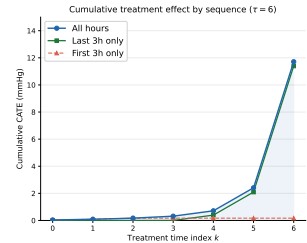

*(a)* Average blip coefficients $\psi_{t,k}$ (in mmHg) quantifying the stepwise causal effect of binary vasopressor treatment at hour $t + k$ on blood pressure at the final endpoint $t + \tau$ ($\tau = 6$).

*(b)* Cumulative conditional average treatment effect ($\tau = 6$) on mean MAP under three vasopressor regimes: continuous administration, last three hours only, and first three hours only.

*Figure 5.* **Real-world MIMIC-III sanity check.** Estimated vasopressor effects on blood pressure through local blip coefficients and cumulative CATE.

**Results:** From Figure 5, we observe that the average blip coefficient is monotonically increasing in $k$; the concurrent treatment ($k = \tau$) contributes 9.3 mmHg; and all coefficients at gaps $\geq$ 3h fall below 0.15 mmHg, which is consistent with average MAP increases reported in existing medical studies (Nishikimi et al., 2026). Treating only the last three hours captures $> 95\%$ of the total effect, whereas treating only the first three hours contributes $< 3\%$, showing that the CATE is concentrated almost entirely in the hours proximal to outcome measurement, consistent with the short pharmacokinetic window of direct vasopressor effects (Kattan et al., 2024).

## 5.4. Ablation study

To further verify our framework's robustness, we replace the transformer architecture by an LSTM (Hochreiter & Schmidhuber, 1997) for our DeepBlip and other baselines in the ablation study (see Appendix E.1). Even with a simpler architecture, our model is highly competitive and outperforms some of the transformer-based baselines (see Appendix E.1), underscoring the effectiveness of our framework.

## 6. Conclusion

We introduced DeepBlip, the first neural framework for SNMMs that overcomes the scalability issues of prior works via a double optimization trick. By using blip functions to decompose total effects into localized increments, our method provides unbiased estimates that are stable over long horizons. Experiments across clinical datasets confirm that DeepBlip achieves state-of-the-art performance, offering a powerful and scalable tool for personalized medicine.

## Impact Statement

We expect our contribution to have a significant impact on *reliable* decision-making in personalized medicine. DeepBlip provides a *stable* learning framework for *efficient* offline evaluation of personalized treatment strategies over long time horizons. Like any other clinical machine learning models, our method might inherit bias within the observational dataset. Careful assessment of the modeling assumptions, dataset quality and robustness checks are required before actual deployment. In addition, the estimated effects are intended for decision support, not as a replacement to clinicians. Overall, we expect our work to enable more robust and scalable causal modeling for time-dependent interventions.

## Acknowledgements

This work has been supported by the German Federal Ministry of Education and Research (Grant: 01IS24082).

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

# A. Preliminaries

In this section we briefly introduce the preliminaries of causal inference.

## A.1. Causal modeling and treatment effect

**Structural causal model** A structural causal model is a tuple $\mathcal{M} = (\mathcal{U}, \mathcal{V}, \mathcal{F}, P)$ where:

1. $\mathcal{U}$ is a set of exogenous variables, which are not caused by any other variables, namely the unknown background factors or simply noises. Each $U_i \in \mathcal{U}$ has a value domain denoted $\mathrm{dom}(U)$. $\mathcal{U}$ is defined on a probability space $(\Omega, \mathcal{B}, P)$.

2. $\mathcal{V}$ is a set of endogenous variables that are determined within the model from other variables. The domain of $V \in \mathcal{V}$ is $\mathrm{dom}(V)$.

3. $\mathcal{F}$ is a set of structural functions: $\mathcal{F} = \{f_i : \mathrm{dom}(\mathrm{Pa}_i) \times \mathrm{dom}(U_i) \to \mathrm{dom}(V_i) \mid V_i \in \mathcal{V}\}$, where $\mathrm{Pa}_i$ is the set of endogenous variable that directly influences $V_i$, and $U_i \in \mathcal{U}$ is the exogenous variable that directly influences $V_i$. Here, $f_i$ is a deterministic function that describes the structural causal mechanism for generating $V_i$: $V_i = f_i(\mathrm{Pa}_i, U_i)$.

Furthermore, the causal relationships specified by $\mathcal{F}$'s $\{\mathrm{Pa}_i \mid V_i \in \mathcal{V}\}$ induce a directed graph $\mathcal{G} = (\mathcal{V}, E)$, for which nodes correspond to the endogenous variables and $(V_i, V_j) \in E \iff V_i \in Pa_j$. It is important to note that the causal graph $\mathcal{G}$ must be a **directed acyclic graph (DAG)**, meaning that there should not be a causal cycle, which is contradictory to both real world and mathematics.

For SCM, the joint probability distribution $P$ induces the probability on the whole variable sets. Once the set $\mathcal{U}$'s variables takes a fixed value, the whole endogenous set's values are determined through $\mathcal{F}$. Let $P_M$ be the joint probability distribution over $\mathcal{V}$, then for any value $v \in \prod_{V_i \in \mathcal{V}} \mathrm{dom}(V_i)$:

$$P_M(V = v) = \Pr(\{u \in \prod_{U_i \in \mathcal{U}} \mathrm{dom}(U_i) \mid \mathcal{F}(u) = v\}) \tag{14}$$

**Intervention:** Intervention could be defined under the context of SCM as forcing a subset of variables (here we assume endogenous) $X \subset \mathcal{V}$ to take a specific value $x \in \prod_{V_i \in X} \mathrm{dom}(V_i)$ [6], independent of the original induced probability. This action simulates the process of controlled experiment or a hypothetical scenario.

An intervention, denoted as $\mathrm{do}(X = x)$, creates a **new** SCM $\mathcal{M}_x = (\mathcal{U}, \mathcal{V}, \mathcal{F}_x, P)$, which is a mutation from the original SCM $\mathcal{M}$, where:

1. The set of endogenous $\mathcal{V}$ and exogenous $\mathcal{U}$ remain unchanged with the same probability space $(\Omega, \mathcal{B}, P)$.

2. For each $V_i \in X$, the old structural equation $V_i = f_i(\mathrm{Pa}_i, U_i)$ is replaced by a degenerated constant equation $V_i \equiv x_i$.

3. For all the other endogenous variables, the mapping $f_i$ stays unchanged, but their distributions may adjust.

On the whole, the intervention $\mathrm{do}(X = x)$ induces a new probability distribution $P_{M_x}$. The interventional distribution of a variable, say $Y \in \mathcal{V}$, is denoted as $\Pr(Y \mid \mathrm{do}(X = x))$ or $\Pr(Y^{\mathrm{do}(X=x)})$. Under certain context for simplicity, we could directly use $\Pr(Y^{(x)})$ without ambiguity.

**Potential Outcome and Treatment effect:** From an intuition perspective, intervention is like cutting the causal links to $X$ from the original diagram, setting the value to $x$ and then observe how the system responds. We emphasize that $\Pr(Y \mid \mathrm{do}(X = x)) \neq \Pr(Y \mid X = x)$, which is just a conditional distribution under the original SCM.

A **potential outcome (PO)** of a variable $Y \in \mathcal{V}$ is the value under a hypothetical scenario, where we make an intervention $\mathrm{do}(X = x)$, given a specific realization of $U = u$. We denote this as $Y_x(u)$. When $U \sim P$. This random variable $Y_x(U)$ is just $Y^{(\mathrm{do}(X=x))}$, which follows the distribution $\Pr(Y \mid \mathrm{do}(X = x))$. The average potential outcome (APO) is the expectation of PO: $\mathbb{E}[Y^{\mathrm{do}(X=x)}]$. A **conditional average potential outcome (CAPO)** is nothing but the expectation of the

---

[6]General intervention could be defined as setting some variables to follow a specific **distribution**(Peters et al., 2017), however we follow a common practice to assume the intervention means forcing a fixed value

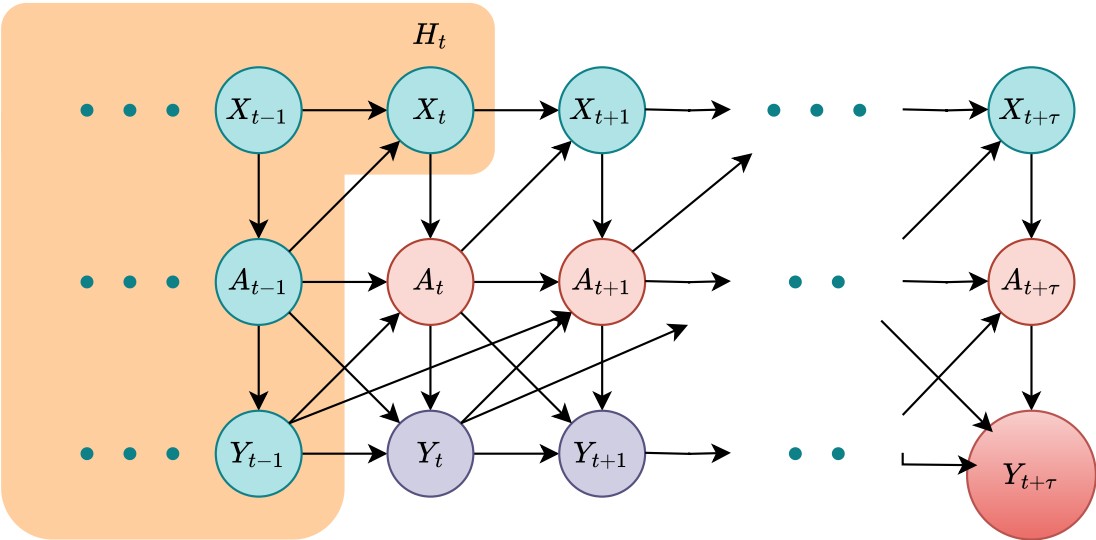

*Figure 6.* An example of causal diagram for the SCM in the setting of estimating CATE over time. The variable sequences are truncated at $t + \tau$ (maximal length is $T$). Exogenous variables $\mathcal{U}$ are hidden.

random variable $Y_x(U)$ conditioned on some subgroups $W = w$, where $W \subset \mathcal{V}$. This conditioning on $W = w$ could be traced back to a posterior distribution on $U$, namely $\Pr(U|W = w) := P_w$, then the CAPO is just $\mathbb{E}_{U_w \sim P_w}[Y_x(U_w)]$ or equivalently $\mathbb{E}[Y^{(x)} \mid W = w]$.

An **average treatment effect (ATE)** is the expectation difference between two POs under intervention $x_1$ and $x_2$ respectively: $\mathbb{E}[Y^{(x_1)} - Y^{(x_2)}]$. A **conditional average treatment effect (CATE)** is then the difference between two CAPOs: $\mathbb{E}[Y^{(x_1)} - Y^{(x_2)} \mid W = w]$.

So far we have strictly defined these foundational concepts. Overall, the potential outcomes are counterfactual variables in intervened SCM while treatment effects are the contrast between these potential outcomes. Within this rigorous framework, we are able to clearly define the goal for our work on estimating CATE over time conditioned on patient history $H_t$ as in Sec. 3.

### A.2. SCM for CATE estimation over time

We formalize the setting for estimating CATE over time with the following structural causal model $\mathcal{M}$:

1. Exogenous variables: $\mathcal{U} = \{U_{X_t}, U_{A_t}, U_{Y_t}\}_{t=1}^{T}$, where $U_{X_t}, U_{A_t}, U_{Y_t}$ are mutually independent noise.

2. Endogenous variables are $\mathcal{V} = \{X_t, A_t, Y_t\}_{t=1}^{T}$. Denote $H_t = \{X_s\}_{s=1}^{t} \cup \{A_j\}_{j=1}^{t-1} \cup \{Y_i\}_{i=1}^{t-1}$ as the history up until time $t$.

3. Structural functions $\mathcal{F}$: for $t \in \{1, \ldots, T\}$

$$X_t = f_{X_t}(H_{t-1} \cup \{A_{t-1}\}, U_{X_t})$$
$$A_t = f_{A_t}(H_t, U_{A_t})$$
$$Y_t = f_{Y_t}(H_t, A_t, U_{Y_t})$$

These equations generate the full observed patient trajectory in the following temporal order: $(X_1 \to A_1 \to Y_1 \to \ldots \to X_T \to A_T \to Y_T)$.

A possible instantiation of temporal graph of this given structural causal model is presented in Figure 6 When talking about SCMs, it is important to point out that SCM represents perfect knowledge of the data generating process. In real-world settings, there are possibly omitted variables (e.g. unmeasured confounders).

**Time-varying treatment and potential outcomes:** A **time-varying intervention** is intervention on a continuous sequence of treatment variables $\text{do}(A_{t:t+\tau} = a_{t:t+\tau})$, which modifies the structural equations for $A_t, \ldots, A_{t+\tau}$:

$$\forall k \in \{t, \ldots, t+\tau\} : \quad A_k \equiv a_k.$$

This induces a new SCM $\mathcal{M}^{(a_{t:t+\tau})}$, generating the potential outcome:

$$Y_{t+\tau}^{(a_{t:t+\tau})} = f_{Y_{t+\tau}} \left( H_{t+\tau}^{(a_{t:t+\tau})}, X_{t+\tau}^{(a_{t:t+\tau})}, a_{t+\tau}, U_{Y_{t+\tau}} \right),$$

where $H_{t+\tau}^{(a_{t:t+\tau})}, X_{t+\tau}^{(a_{t:t+\tau})}$ are recursively computed under the intervention using structural equations defined in 3.

**Identification assumptions:** The structural functions, although might be unspecified, already contains rich information about the data generating process. It is easy to verify that under the specified data generating process and the mutual independence between the exogenous variables, the following two conditions are satisfied:

1. **Consistency**: If $A_{t:t+\tau} = a_{t:t+\tau}$ is the observed treatment sequence for a given patient $u$, then $Y_{t+\tau}^{(a_{t:t+\tau})}(u) = Y(u)$. Or, equivalently $Y_{t+\tau}^{(A_{t:t+\tau})} = Y_{t+\tau}$. This means when we intervene with the observed treatment, the potential outcome equals the observed outcome.

2. **Sequential ignorability**: The potential outcome under a fixed intervention is conditionally independent with the treatment assignment:
$$Y_t^{(a_t)} \perp A_t \mid H_t, \quad \forall a_t, 1 \le t \le T$$
This ignorability assumption implies that there are no hidden confounders that affect both outcome and the treatment, which is already decided with the SCM.

3. **Sequential overlap**: If $\Pr(H_t = h_t) > 0$, then:
$$\Pr(A_t = a_t \mid H_t = h_t) > 0 \quad \forall a_t \in \mathcal{A}_t, 1 \le t \le T.$$

These three assumptions together allow the use of observational data to compute causal quantities, such as potential outcome over time and treatment effect over time (Robins et al., 2000). Without these assumptions, exact identification would be impossible. While these assumptions themselves are untestable, they could be justified under appropriate study design and domain knowledge.

**Time-varying confounding:** Estimating treatment effect over time poses a significant challenge due the existence of time-varying confounding (Coston et al., 2020), which differentiate the task from the static setting ($\tau = 0$). The underlying reason is that, the future time-varying confounders, such as $X_{t+1:t+tau}$ or $Y_{t:t+\tau-1}$, are *unobserved* during the inference phase. As a result, the naïve adjustment with the history variable $H_t$ is not sufficient for the backdoor criteria (Peters et al., 2017) and, thus, leads to biased estimation:

$$\text{When } \tau > 0 : \quad \mathbb{E}\left[ Y_{t+\tau}^{(a_{t:t+\tau})} \mid H_t = h_t \right] \ne \mathbb{E}\left[ Y_{t+\tau} \mid A_{t:t+\tau} = a_{t:t+\tau}, H_t = h_t \right]. \tag{15}$$

# B. Extended Related Works

## B.1. Non-parametric models

There have been some attempts to use non-parametric models (Schulam & Saria, 2017; Soleimani et al., 2017; Xu et al., 2016) to estimate CATE or CAPO over time, yet these approaches impose strong assumptions on the outcomes, and their scalability is limited. For these reasons, we focus on neural methods, which offer better flexibility and scalability for complex, high-dimensional medical data.

## B.2. Treatment effect Estimation over continuous time

A streamline in the literature, orthogonal to ours, is estimating HTEs or potential outcomes over *continuous* time. Prominent examples like Hess et al. (2024); Seedat et al. (2022) leverage Neural Controlled Differential Equations (NCDEs), a method well-suited for modeling dynamic systems, to address the problem. However, these continuous-time models introduce significant complexity and computational overhead, limiting their practical applicability. Hence, we only focus on discrete-time treatment effect estimation.

## B.3. Balanced representation based approaches

Learning the balanced representation to predict treatment effect or potential outcome is very popular. Similar to the previously introduced Causal Transformer (Melnychuk et al., 2022), Wang et al. (2024) adopts an adversarial generative approach to learn the balanced representations, which reduces the confounding bias. Wang et al. (2025a) introduces a de-correlation strategy that removes cross-covariance between current treatments and representation of past trajectory, based on a data selective state-space model architecture called Mamba. However, balanced representation-based methods are considered *biased* because they are only heuristics that lack formal theoretical guarantee for adjusting confounding. In fact, these methods were originally designed to reduce finite-sample variance, not to mitigate confounding bias. Moreover, the process of enforcing such balance would even introduce more bias (Melnychuk et al., 2024).

## B.4. Other model-based approaches

Wang et al. (2025b) proposed Variational Counterfactual Intervention Planning (VCIP), a framework that uses variational inference to directly model the likelihood of the potential outcome over sequential interventions. The method relies on the assumption that the ELBO derived from observational data truly approximates the interventional likelihood. However, this is hard to verify. Furthermore, the approach uses simple distributions such as Gaussian (state transition) and Beta (outcome distribution), which may fail to capture the complex relationship, leading to biased estimation.

Meng et al. (2023) introduces the COSTAR framework that first pre-trains a transformer-based encoder using a self-supervised loss and then fine-tunes the encoder to predict the factual outcome. The work lacks a causal adjustment mechanism, hoping that the pre-trained transformer can incidentally transfer to the setting of estimating counterfactual outcomes. Hence, the approach is a heuristic and does not properly adjust for time-varying confounding.

## B.5. Other meta-learners

Recent meta-learners further develop various orthogonal learning methods for sequential treatment. Javurek et al. (2026) propose the DRQ-learner for individualized potential outcomes in Markov decision processes, establishing double robustness, Neyman-orthogonality, and quasi-oracle efficiency for Q-function estimation. However, the method relies on inverse propensity weighting, similar to MSMs, and thus is still vulnerable to instability from extreme weights. Hess et al. (2026b) propose an overlap-weighted orthogonal meta-learner that explicitly addresses exploding variance from poor treatment-sequence overlap by focusing on high-overlap regions. This improves stability, but changes the target toward an overlap-weighted estimand, which may bias estimates under misspecified models.

## C. Theory of structural nested mean models

In this section, we show by strict causal reasoning how to use the structural nested mean models to estimate CATE over time. The theory consists of the foundations of SNMM (Robins, 2004) and a specific realization (Lewis & Syrgkanis, 2021), which we adopt for our method. We show the mathematical derivations for completeness. Next, in sec. C.2 we show that our loss is Neyman orthogonal. Finally, we briefly explain the mean squared error rate for the estimated blip coefficients which is proved by previous works in SNMM (Lewis & Syrgkanis, 2021).

To simplify the notation, we denote $Y = Y_{t+\tau}$. And we use $\underline{W}_{t+k}$ to denote $\underline{W}_{t+k:t+\tau}$ for any variable $W \in \mathcal{V}$.

### C.1. SNMM

**Identification:** The blip functions can be accumulated to identify the potential outcomes, as summarized by the theorem (Robins, 2004) below. We show the proof for completeness.

**Theorem C.1.** *Adjustment via blip functions (Theorem 3.1 from Robins (2004)) Given a policy $d$ between time $t$ and $t+\tau$, the following identification holds under the sequential ignorability assumption (Appendix C):*

$$\mathbb{E}\Big[Y^{(d)} \mid H_t = h_t\Big] = \mathbb{E}\Big[Y + \sum_{k=0}^{\tau} \rho_{t,k}\big(X_{t+1:t+k}, A_{t:t+k}; h_t\big) \mid H_t = h_t\Big] \tag{16}$$

*where* $\rho_{t,k}\big(X_{t+1:t+k}, A_{t:t+k}; h_t\big) = \gamma_{t,k}\Big(X_{t+1:t+k}, \big(A_{t:t+k-1}, d_{t+k}\big); h_t\Big) - \gamma_{t,k}\big(X_{t+1:t+k}, A_{t:t+k}; h_t\big).$

*Proof.* By the definition of $\gamma_{t,k}$, we could write $\gamma_{t,k}$ into the following form:

$$\gamma_{t,k}\Big(x_{t+1:t+k}, \big(a_{t:t+k-1}, d_{t+k}\big); h_t\Big)$$
$$\stackrel{\text{Def.}}{=} \mathbb{E}\Big[Y^{\big(a_{t:t+k-1}, \underline{d}_{t+k}\big)} - Y^{\big(a_{t:t+k-1}, 0, \underline{d}_{t+k+1}\big)} \mid X_{t+1:t+k} = x_{t+1:t+k}, A_{t:t+k-1} = a_{t:t+k-1}, A_{t+k} = d_{t+k}, H_t = h_t\Big]$$
$$\stackrel{\text{Ignor.}}{=} \mathbb{E}\Big[Y^{\big(a_{t:t+k-1}, \underline{d}_{t+k}\big)} - Y^{\big(a_{t:t+k-1}, 0, \underline{d}_{t+k+1}\big)} \mid X_{t+1:t+k} = x_{t+1:t+k}, A_{t:t+k} = a_{t:t+k}, H_t = h_t\Big] \tag{17}$$

Similarly:

$$\gamma_{t,k}\Big(x_{t+1:t+k}, a_{t:t+k}\big); h_t\Big)$$
$$\stackrel{\text{Def.}}{=} \mathbb{E}\Big[Y^{\big(a_{t:t+k}, \underline{d}_{t+k+1}\big)} - Y^{\big(a_{t:t+k-1}, 0, \underline{d}_{t+k+1}\big)} \mid X_{t+1:t+k} = x_{t+1:t+k}, A_{t:t+k} = a_{t:t+k}, H_t = h_t\Big] \tag{18}$$

This implies that:

$$\rho_{t,k}\big(X_{t+1:t+k}, A_{t:t+k}; h_t\big)$$
$$= \gamma_{t,k}\Big(X_{t+1:t+k}, \big(A_{t:t+k-1}, d_{t+k}\big); h_t\Big) - \gamma_{t,k}\big(X_{t+1:t+k}, A_{t:t+k}; h_t\big)$$
$$\stackrel{\text{By 17,18}}{=} \mathbb{E}\Big[Y^{\big(A_{t:t+k-1}, \underline{d}_{t+k}\big)} - Y^{\big(A_{t:t+k}, \underline{d}_{t+k+1}\big)} \mid X_{t+1:t+k}, A_{t:t+k}, H_t = h_t\Big] \tag{19}$$

Figure 7 visualizes how the blip function isolates the localized treatment effect at each time step.

By consistency 1, $Y^{(A_{t:t+\tau})} = Y$, and that by towering law of the conditional expectation:

$$\mathbb{E}\Big[\rho_{t,k}\big(X_{t+1:t+k}, A_{t:t+k}; h_t\big) \mid H_t = h_t\Big]$$
$$= \mathbb{E}\Big[\mathbb{E}\big[Y^{\big(A_{t:t+k-1}, \underline{d}_{t+k}\big)} - Y^{\big(A_{t:t+k}, \underline{d}_{t+k+1}\big)} \mid X_{t+1:t+k}, A_{t:t+k}, H_t = h_t\big] \mid H_t = h_t\Big]$$
$$= \mathbb{E}\Big[Y^{\big(A_{t:t+k-1}, \underline{d}_{t+k}\big)} - Y^{\big(A_{t:t+k}, \underline{d}_{t+k+1}\big)} \mid H_t = h_t\Big] \tag{20}$$

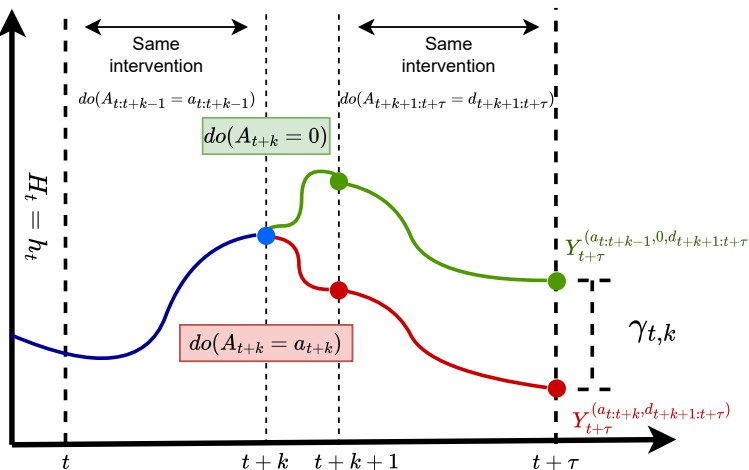

*Figure 7.* Visualization of the blip function $\gamma_{t,k}$. The blip function quantifies the localized treatment effect at each time step.

Summing up (20) for $k = 0, 1, .., \tau$, we observe that all the middle terms cancel off:

$$\mathbb{E}\left[\sum_{k=0}^{\tau} \rho_{t,k}\big(X_{t+1:t+k}, A_{t:t+k}; h_t\big) \mid H_t = h_t\right]$$

$$= \mathbb{E}\left[\sum_{k=0}^{\tau} Y^{(A_{t:t+k-1}, \underline{d}_{t+k})} - Y^{(A_{t:t+k}, \underline{d}_{t+k+1})} \mid H_t = h_t\right]$$

$$= \mathbb{E}\left[Y^{(d)} - Y \mid H_t = h_t\right] \tag{21}$$

This is equivalent with the Eq. (3). □

Lemma C.1 shows that, in order to estimate the potential outcomes, it suffices to estimate the blip functions. Now the identification problem is reduced into correctly estimating these blip functions. Suppose $\gamma_t^\dagger(\cdot, \cdot; h_t) = (\gamma_{t,0}^\dagger, \gamma_{t,1}^\dagger, \ldots, \gamma_{t,\tau}^\dagger)$ is an **arbitrary** candidate function that takes $x_{t+1:t+k}, a_{t:t+k}$ as input and is subject to the following conditions:

$$\forall k \in [0, \tau], \quad \gamma_{t,k}^\dagger(x_{t+1:t+k}, (a_{t:t+k-1}, 0); h_t) = 0. \tag{22}$$

Let:

$$H_{t,k}(\gamma_t^\dagger) = Y + \sum_{j=k}^{T} \left(\gamma_{t,k}^\dagger(X_{t+1:t+j}, (A_{t:t+j-1}, d_{t+j})) - \gamma_{t,k}^\dagger(X_{t+1:t+j}, A_{t:t+j})\right) \tag{23}$$

Then, with Lemma C.1 and the sequential conditional ignorability, Theorem 3.2 from Robins (2004) states that the moment restriction below is satisfied. Here we present it as a crucial lemma and show the proof for completeness.

**Lemma C.2.** *Moment restriction for blip functions* *When (22) is satisfied, then for arbitrary function $S_k$, the following two conditions are equivalent:*

1. *$\gamma_{t,k}^\dagger(\cdot, \cdot; h_t)$ is the true $\gamma_{t,k}(\cdot, \cdot; h_t)$.*

2. *The following moment restriction is satisfied:*

$$\mathbb{E}\left[H_{t,k}(\gamma_t^\dagger)(S_k(X_{t+1:t+k}, A_{t:t+k}) - \mathbb{E}[S_k(X_{t+1:t+k}, A_{t:t+k}) \mid X_{t+1:t+k}, A_{t:t+k-1}]) \mid H_t = h_t\right] = 0 \tag{24}$$

*Proof.* We only prove the direction of $1 \to 2$, which already conveys the core thought of the moment restriction.

We denote:

$$R_k(X_{t+1:t+k}, A_{t:t+k}) = S_k(X_{t+1:t+k}, A_{t:t+k}) - \mathbb{E}[S_k(X_{t+1:t+k}, A_{t:t+k}) \mid X_{t+1:t+k}, A_{t:t+k-1}, H_t = h_t]$$

Observing the LHS of (24):

$$
\begin{aligned}
\text{L.H.S.} =& \mathbb{E}\left[H_{t,k}(\boldsymbol{\gamma}_t^{\dagger})R_k(X_{t+1:t+k}, A_{t:t+k}) \mid H_t = h_t\right] \\
\stackrel{\text{Cond.}}{=}& \mathbb{E}\left[\mathbb{E}\left[H_{t,k}(\boldsymbol{\gamma}_t)R_k(X_{t+1:t+k}, A_{t:t+k}) \mid X_{t+1:t+k}, A_{t:t+k}\right] \mid H_t = h_t\right] \\
=& \mathbb{E}\left[\mathbb{E}\left[H_{t,k}(\boldsymbol{\gamma}_t) \mid X_{t+1:t+k}, A_{t:t+k}\right] \cdot R_k(X_{t+1:t+k}, A_{t:t+k}) \mid H_t = h_t\right] \\
\stackrel{\text{Lem. } C.1}{=}& \mathbb{E}\left[\mathbb{E}\left[Y^{(A_{t:t+k-1}, \underline{d}_{t+k})} \mid X_{t+1:t+k}, A_{t:t+k}\right] \cdot R_k(X_{t+1:t+k}, A_{t:t+k}) \mid H_t = h_t\right] \\
\stackrel{\text{Ignor.}}{=}& \mathbb{E}\left[\mathbb{E}\left[Y^{(A_{t:t+k-1}, \underline{d}_{t+k})} \mid X_{t+1:t+k}, A_{t:t+k-1}\right] \cdot R_k(X_{t+1:t+k}, A_{t:t+k}) \mid H_t = h_t\right] \\
\stackrel{\text{Tower}}{=}& \mathbb{E}\left[\mathbb{E}\left[Y^{(A_{t:t+k-1}, \underline{d}_{t+k})} \mid X_{t+1:t+k}, A_{t:t+k-1}\right] \cdot \mathbb{E}\left[R_k(X_{t+1:t+k}, A_{t:t+k}) \mid X_{t+1:t+k}, A_{t:t+k-1}\right] \mid H_t = h_t\right] \\
\stackrel{\text{Def.}}{=}& \mathbb{E}\left[\mathbb{E}\left[Y^{(A_{t:t+k-1}, \underline{d}_{t+k})} \mid X_{t+1:t+k}, A_{t:t+k-1}\right] \cdot 0 \mid H_t = h_t\right] = 0
\end{aligned}
\tag{25}
$$

$\square$

To further reduce the complexity of the moment restriction to achieve locally efficient doubly robust estimator (see Section 3.3 from Robins (2004)), we could subtract $H_{t,k}(\gamma^*)$(24) by its conditional expectation $\mathbb{E}[H_{t,k}(\gamma_t^{\dagger}) \mid X_{t+1:t+k}, A_{t:t+k-1}]$:

$$
\mathbb{E}\left[\left(H_{t,k}(\gamma_t^{\dagger}) - \mathbb{E}[H_{t,k}(\gamma_t^{\dagger}) \mid X_{t+1:t+k}, A_{t:t+k-1}]\right)R_k(X_{t+1:t+k}, A_{t:t+k}) \mid H_t = h_t\right] = 0
\tag{26}
$$

**Parametrization of the blip function:** To achieve parametric rates of the estimation, we use the common semi-parametrization of the blip function[7] (Lewis & Syrgkanis, 2021; Robins, 2004):

$$
\gamma_{t,k}\left(x_{t+1:t+k}, a_{t:t+k} \,; h_t\right) = \psi_{t,k}(h_t)' a_{t+k}
\tag{27}
$$

Here $\psi_{t,k}$ is a function that maps a given $H_t = h_t$ to $\psi_{t,k}(h_t)$, which is the parameter of the blip function $\gamma_{t,k}(\cdot, \cdot \,; h_t)$. Intuitively, these $\psi_{t,k}(h_t)$ works as the coefficients in a linear parametrization. Hence, we call them in our method *blip coefficients* and the mapping $\psi_{t,k}(\cdot)$ *blip coefficient predictor* or *blip coefficient estimator*.

**Deriving the moment restriction of SNMM:** We abuse the notation of $H_{t,k}(\gamma_t^*)$ with using the parameter $\psi_t$ to represent the $H_{t,k}(\gamma_t^{\psi_t})$ as $H_{t,k}(\psi_t)$. For convenience, we introduce several notation for residuals:

$$
\widetilde{A}_{t,j,k} = A_{t+j} - \mathbb{E}\left[A_{t+j} \mid X_{t+1:t+k}, A_{t:t+k-1}, H_t = h_t\right], \quad 0 \leq k \leq j \leq \tau
\tag{28}
$$

$$
\widetilde{Y}_{t,k} = Y - \mathbb{E}\left[Y \mid X_{t+1:t+k}, A_{t:t+k}, H_t = h_t\right], \qquad 0 \leq k \leq \tau
\tag{29}
$$

Then we could derive the following (with definition of $H_{t,k}(\cdot)$ in (23)), which is an adaptation from Lewis & Syrgkanis (2021):

$$
\mathbb{E}\left[H_{t,k}(\psi_t) - \mathbb{E}[H_{t,k}(\psi_t) \mid X_{t+1:t+k}, A_{t:t+k-1}] \mid H_t = h_t\right] = \widetilde{Y}_{t,k} - \sum_{j=k}^{\tau} \psi_{t,j}' \widetilde{A}_{t,j,k}
\tag{30}
$$

Moreover, we specify $S_k(x_{t+1:t+k}, a_{t:t+k-1}) = a_{t+k}$ from Lemma C.2, then:

$$
\begin{aligned}
& R_k(X_{t+1:t+k}, A_{t:t+k-1}) \\
=& S_k(X_{t+1:t+k}, A_{t:t+k}) - \mathbb{E}\left[S_k(X_{t+1:t+k}, A_{t:t+k}) \mid X_{t+1:t+k}, A_{t:t+k-1}, H_t = h_t\right] = \widetilde{A}_{t,j,k}
\end{aligned}
\tag{31}
$$

Therefore, the moment restriction in (26) could be simplified as:

$$
\forall k \in \{0, \ldots, \tau\}: \quad \mathbb{E}\left[\left(\widetilde{Y}_{t,k} - \sum_{j=k+1}^{\tau} \psi_{t,j}(h_t)' \widetilde{A}_{t,j,k} - \psi_{t,k}(h_t)' \widetilde{A}_{t,k,k}\right) \widetilde{A}_{t,k,k} \mid H_t = h_t\right] = 0
\tag{32}
$$

which is the primary moment restriction for our setting, adapted from Lewis & Syrgkanis (2021). To estimate $\psi_t(h_t)$ we need to solve the set of equations defined in (32) in an iterative way (See Algorithm 3 from Lewis & Syrgkanis (2021)):

---

[7]In our paper, we choose the simplest parametrization for clarity. In fact, the blip functions could have more sophisticated parametrizations, such as $\gamma_{t,k}\left(x_{t+1:t+k}, a_{t:t+k} \,; h_t\right) = \psi_{t,k}(h_t)' \phi_{t,k}(x_{t+1:t+k}, a_{t:t+k} \,; h_t)$, where $\phi$ is a nonlinear, high-dimensional feature map (Lewis & Syrgkanis, 2021; Robins, 2004).

1. Step 0: for k = $\tau$, solve $\widehat{\psi}_{t,\tau}(h_t) \in \mathbb{R}^r$

2. Step $\tau - k, (k \geq 1)$, With the given $\widehat{\psi}_{t,j}(h_t), j > k$, solve $\widehat{\psi}_{t,k}(h_t)$.

Each step we need to solve a linear equation, which has a unique solution under standard assumptions (see Theorem 8 from Lewis & Syrgkanis (2021)). And since the solution is unique, the solution equals the true blip coefficients. However, we argue that this classical method has limitations due to its sequential nature which prevents us from batch training. Noticing that solving (32) is equivalent to solving this minimization problem, we successfully establish the estimation of $\psi_t$ under risk minimization paradigm, suitable for training with neural networks:

$$\psi_{t,k}^* = \underset{\widehat{\psi}_{t,k}(\cdot) \in \Phi_{t,k}}{\arg\min} \ \mathbb{E}\left[\left(\widetilde{Y}_{t,k} - \sum_{j=k+1}^{\tau} \psi_{t,j}^*(h_t)' \widetilde{A}_{t,j,k} - \widehat{\psi}_{t,k}(h_t)' \widetilde{A}_{t,k,k}\right)^2\right] \tag{33}$$

**Estimating CATE over time:** Once the parameters $\psi_t(h_t) = (\psi_{t,0}(h_t), .., \psi_{t,\tau}(h_t))$ are estimated, we could proceed to estimate the potential outcome from Eq. (3):

$$\mathbb{E}\Big[Y^{(d)} \ \Big| \ H_t = h_t\Big]$$

$$= \mathbb{E}\Big[Y \ + \ \sum_{k=0}^{\tau} \rho_{t,k}\big(X_{t+1:t+k}, A_{t:t+k}; h_t\big) \ \Big| \ H_t = h_t\Big]$$

$$= \mathbb{E}\Big[Y + \sum_{k=0}^{\tau} \widehat{\psi}_{t,k}(h_t)'(d_{t+k} - A_{t+k}) \ | \ H_t = h_t\Big]. \tag{34}$$

Then the CATE over time under treatment sequences $a^*$ and $b^*$ is computed by taking difference between the potential outcomes respectively Lewis & Syrgkanis (2021); Robins (1994; 2004):

$$\mathbb{E}\Big[Y^{(a^*)} - Y^{(b^*)} \ \Big| \ H_t = h_t\Big]$$

$$= \mathbb{E}\Big[\sum_{k=0}^{\tau} \psi_{t,k}(h_t)'\big[(a_{t+k}^* - A_{t+k}) - (b_{t+k}^* - A_{t+k})\big] \ \Big| \ H_t = h_t\Big]$$

$$= \mathbb{E}\Big[\sum_{k=0}^{\tau} \psi_{t,k}(h_t)'(a_{t+k}^* - b_{t+k}^*) \ | \ H_t = h_t\Big]$$

$$= \sum_{k=0}^{\tau} \psi_{t,k}(h_t)'(a_{t+k}^* - b_{t+k}^*) \tag{35}$$

The expectation term vanishes since all the terms left are constant in the conditional expectation, enabling us to directly estimate CATE over time.

### C.2. Neyman orthogonality of the $L^2$-moment loss

**Frechet Derivative:** The Fréchet derivative of any functional $\mathcal{L}(f)$ is defined as:

$$\forall v, \quad D_f \mathcal{L}(f)[v] = \frac{\partial}{\partial t} \mathcal{L}(f + tv)\Big|_{t=0}. \tag{36}$$

where $h = (p, q)$ is the nuisance parameter, $\theta$ is the trainable weight of the blip prediction model from Stage 2. Our objective is to show that:

$$D_h(\partial_\theta \mathcal{L}_{D,k}(\theta; h)) = 0 \tag{37}$$

where $D_h$ is the Fréchet derivative w.r.t. the nuisance function $h$. This property is called the **universal orthogonality** (Foster & Syrgkanis, 2019), which mandates that the gradient of the loss $\partial_\theta \mathcal{L}_{D,k}(\theta; h)$ has a vanishing derivative over the nuisance parameters $h$. A functional loss that satisfies this property is more robust to the perturbation of the nuisance

parameter, since the first-order error of $h$ would only have a second-order effect on the gradient. Secondly, with the universal Neyman orthogonality, it is possible to achieve lower prediction error (Foster & Syrgkanis, 2019), as stated in the theorem in Section C.3.

Recall that our double-optimization-adapted $L^2$-moment loss at step $k$ is:

$$\mathcal{L}_{\text{blip}}^k := \mathcal{L}_{D,k}(\theta; h) = \mathbb{E}\Big[\Big(Y - p_{t,k}(H_{t+k}) - \sum_{j=k+1}^{\tau} {}'\psi_{t,j}^{(\text{ps})}(H_t)'\big(A_{t+j} - q_{t,j,k}(H_{t+k})\big) \tag{38}$$
$$-\psi_{t,k}(H_t; \theta)'\big(A_{t+k} - q_{t,k,k}(H_{t+k})\big)\Big)^2\Big],$$

where $h = (\mathbf{p}, \mathbf{q})$ is the nuisance parameter, where $\mathbf{p} = \{p_{t,k} \mid 0 \le k \le \tau\}$ and $\mathbf{q} = \{q_{t,j,k} \mid 0 \le k \le j \le \tau\}$ and $\theta$ is the trainable weight of the blip prediction model from Stage 2. $\psi_{t,j}^{(\text{ps})}(H_t)$ is the pseudo target resulting from gradient detachment in the adapted loss.

First, we denote:

$$\phi_k(\theta, h) := Y - p_{t,k}(H_{t+k}) - \sum_{j=k+1}^{\tau} \psi_{t,j}^{(\text{ps})}(H_t)'\big(A_{t+j} - q_{t,j,k}(H_{t+k})\big)$$
$$-\psi_{t,k}(H_t; \theta)'\big(A_{t+k} - q_{t,k,k}(H_{t+k})\big), \tag{39}$$

then, we derive the gradient of $\mathcal{L}_{D,k}$ over $\theta$. Note that the nuisance functions $p_{t,k}, q_{t,j,k}$ does not depend on the model weight $\theta$, the gradient is then:

$$\partial_\theta \mathcal{L}_{D,k}(\theta, h) = \mathbb{E}\Big[2 \cdot \phi_k(\theta, h) \cdot -\partial_\theta \psi_{t,k}(h_t; \theta)'\big(A_{t+k} - q_{t,k,k}(H_{t+k})\big)\Big]. \tag{40}$$

Let

$$\delta_k(\theta, h) = -\partial_\theta \psi_{t,k}(H_t; \theta)'\big(A_{t+k} - q_{t,k,k}(H_{t+k})\big), \tag{41}$$

Then we have

$$\partial_\theta \mathcal{L}_{D,k}(\theta, h) = 2 \cdot \mathbb{E}\Big[\phi_k(\theta, h) \cdot \delta_k(\theta, h)\Big]. \tag{42}$$

Then we proceed to show that the Fréchet derivative of $\partial_\theta \mathcal{L}_{D,k}(\theta, h)$ over $p_{t,k}, q_{t,j,k}$ (where $j > k$, and $q_{t,k,k}$ are all zero.

**(1) Orthogonality over $p_{t,k}$:** Note that $D_{p_{t,k}}\delta_k(\theta, h) = 0$ and $D_{p_{t,k}}\phi_k(\theta, h)[v] = v(H_{t+k})$. then

$$D_{p_{t,k}}\partial_\theta \mathcal{L}_{D,k}(\theta, h)[v] = 2 \cdot \mathbb{E}\Big[v(H_{t+k}) \cdot \delta_k(\theta, h)\Big]$$
$$= 2 \cdot \mathbb{E}\Big[\mathbb{E}[v(H_{t+k}) \cdot \delta_k(\theta, h) \mid H_{t+k}]\Big]$$
$$= 2 \cdot \mathbb{E}\Big[v(H_{t+k}) \cdot \mathbb{E}[\delta_k(\theta, h) \mid H_{t+k}]\Big]$$
$$= 2 \cdot \mathbb{E}\Big[v(H_{t+k}) \cdot 0\Big] = 0. \tag{43}$$

**(2) Orthogonality over $q_{t,j,k}$:** Note that $D_{q_{t,j,k}}\phi_k(\theta, h)[u] = \psi_{t,j}^{(\text{ps})}(H_t)'u(H_{t+k})$ and $D_{q_{t,j,k}}\delta_k(\theta, h) = 0$, then

$$D_{q_{t,j,k}}\partial_\theta \mathcal{L}_{D,k}(\theta, h)[u] = 2 \cdot \mathbb{E}\Big[\psi_{t,j}^{(\text{ps})}(H_t)'u(H_{t+k}) \cdot \delta_k(\theta, h)\Big]$$
$$= 2 \cdot \mathbb{E}\Big[\mathbb{E}[\psi_{t,j}^{(\text{ps})}(H_t)'u(H_{t+k}) \cdot \delta_k(\theta, h) \mid H_{t+k}]\Big]$$
$$= 0. \text{ (similar to the proof of } p_{t,k}) \tag{44}$$

**(3)Orthogonality over** $q_{t,j,k}$**:** Note that $D_{q_{t,k,k}}\phi_k(\theta, h)[u] = \psi_{t,k}(H_t, \theta)'u(H_{t+k})$ and $D_{q_{t,k,k}}\delta_k(\theta, h) = \partial_\theta\psi_{t,k}(H_t;\theta)'u(H_{t+k})$, then

$$D_{q_{t,j,k}}\partial_\theta\mathcal{L}_{D,k}(\theta, h)[u] = 2 \cdot \mathbb{E}\Big[\psi_{t,k}(H_t, \theta)'u(H_{t+k}) \cdot \delta_k(\theta, h) + \tag{45}$$

$$\phi_k(\theta, h) \cdot \partial_\theta\psi_{t,k}(H_t;\theta)'u(H_{t+k})\Big]$$

$$= 2 \cdot \mathbb{E}\Big[\mathbb{E}\big[\psi_{t,k}(H_t, \theta)'u(H_{t+k}) \cdot \delta_k(\theta, h) + \tag{46}$$

$$\phi_k(\theta, h) \cdot \partial_\theta\psi_{t,k}(H_t;\theta)'u(H_{t+k}) \mid H_{t+k}\big]\Big]$$

$$= 2 \cdot \mathbb{E}\Big[\psi_{t,k}(H_t, \theta)'u(H_{t+k}) \cdot \mathbb{E}\big[\delta_k(\theta, h) \mid H_{t+k}\big] + \tag{47}$$

$$\mathbb{E}\big[\phi_k(\theta, h) \mid H_{t+k}\big] \cdot \partial_\theta\psi_{t,k}(H_t;\theta)'u(H_{t+k})\Big]$$

$$= 2 \cdot \mathbb{E}\Big[\psi_{t,k}(H_t, \theta)'u(H_{t+k}) \cdot 0 + 0 \cdot \partial_\theta\psi_{t,k}(H_t;\theta)'u(H_{t+k})\Big]$$

$$= 0. \tag{48}$$

Putting (1), (2), (3) together, we prove that the overall Fréchet derivative of $\partial_\theta\mathcal{L}_{D,k}(\theta, h)$ over nuisance parameter $h$ is zero. And hence, the adapted $L^2$-moment loss satisfies universal Neyman orthogonality.

### C.3. The MSE rate theorem

A mean squared error (MSE) guarantee under finite samples for the estimator $\hat{\psi}_t$ in a primary heterogeneous setting, where the conditioning variable is the static feature $X_0$ (see theorem 10 from Lewis & Syrgkanis (2021)). We point out that the conclusion of the theorem could be extended to our setting, where conditioning variable is the history $H_t$. Before we show the theorem, we clarify the prerequisite notations and regular assumptions:

**Mathematical notations:**

1. **Norm of a function in function space** Define the norm for a vector-valued function $\psi : \mathcal{X} \to \mathbb{R}^d$, then the $u, v$-norm is defined as:

$$\|\psi\|_{u,v} = \mathbb{E}_{X \sim P_X}\Big[\big(\sum_{i=1}^d \psi_i(X)^u\big)^{\frac{v}{u}}\Big]^{\frac{1}{v}}. \tag{49}$$

We point out that when $u = v$, the $u, v$-norm degenerate into the normal norm: $\|\cdot\|_{u,u} = \|\cdot\|_u$. Further, we could define the product norm between two functions $f, g$ as:

$$\|f \circ g\|_{u,v} = \mathbb{E}\Big[\|f(\hat{X})\|_u^v \cdot \|g(\hat{Y})\|_u^v\Big]^{1/v} = \mathbb{E}\Big[\big(\sum_{i=1}^r f_i(X)^u\big)^{\frac{v}{u}} \cdot \sum_{i=1}^r g_i(X)^u\big)^{\frac{v}{u}}\Big]^{\frac{1}{v}} \tag{50}$$

2. **Localized Rademacher complexity:** The localized Rademacher complexity is an extended concept of the standard Radmacher complexity. It measures the the fitting capacity of a function space $\mathcal{F}$ where functions have bounded second moment. Formally it is defined as:

$$R_{P_X,n}(\mathcal{F}; \delta) = \mathbb{E}_{\epsilon_{1:n}, X_{1:n} \sim P_X}\Big[\sup_{f \in \mathcal{F}:\|f\|_2 \leq \delta} \frac{1}{n}\sum_{i=1}^n \epsilon_i f(X_i)\Big] \tag{51}$$

**Assumptions:** We adapt the assumptions from the original heterogeneous setting (Lewis & Syrgkanis, 2021) to conditioning on the history varible $H_t$:

1. All the variables and functions in our problem setting are bounded.

2. A regularity assumption regarding the treatment variables: The following quantities are also bounded:

$$c_{k,j} := \sup_{t \leq T-\tau} \sup_{h_t \in \mathcal{H}_t} \mathbb{E}\left[\|C_{t,k,j}\|_{u,u}^v \mid H_t = h_t\right]^{2/v} < +\infty \tag{52}$$

where $C_{t,k,j} := \text{Cov}(A_{t+k}, A_{t+j} \mid H_{t+k})$, and:

$$\forall h_t \in \mathcal{H}_t, \quad \mathbb{E}\left[C_{t,k,k} \mid H_t = h_t\right] \succeq \lambda I \tag{53}$$

3. The $u, \infty$-norms of $\psi_{t,k}$ are bounded by:

$$M := \max_{t \leq T-\tau} \max_{0 \leq k \leq \tau, \psi_{t,k} \in \Psi_{t,k}} \|\psi_{t,k}\|_{u,\infty} < +\infty \tag{54}$$

**Theorem 1 (Adapted from Heterogeneous R-learner(Lewis & Syrgkanis, 2021)):** Suppose the assumptions above are satisfied. Let $\delta_n$ be an upper bound on the critical radius of the star hull of all the function spaces

$$\{\Psi_{t,k,i} - \psi_{t,k,i}\}_{t \in [SL-\tau], k \in [\tau], i \in [r]}$$

and that $\delta_n = \Omega\left(\sqrt{\frac{r \log\log(n)}{n}}\right)$. Suppose:

$$\forall 1 \leq t \leq T-\tau, \quad \max_{0 \leq k \leq j \leq \tau} \mathbb{E}_{\hat{q}_{t,k,k}, \hat{q}_{t,j,k}}\left[\|(\hat{q}_{t,k,k} - q_{t,k,k}) \circ (\hat{q}_{t,j,k} - q_{t,j,k})\|_{2,2}^2\right] = O(r^2 \delta_{n/2}^2) \tag{55}$$

$$\forall 1 \leq t \leq T-\tau, \quad \max_{0 \leq k \leq \tau} \mathbb{E}_{\hat{q}_{t,k,k}, \hat{q}_{t,k}}\left[\|(\hat{q}_{t,k,k} - q_{t,k,k}) \circ (\hat{p}_{t,k} - p_{t,k})\|_{2,2}^2\right] = O(r^2 \delta_{n/2}^2), \tag{56}$$

then the blip coefficient predictor $\widehat{\psi}_{t,k}$ trained under the $L^2$-moment loss satisfies:

$$\max_{t \leq T-\tau} \max_{k \in \{0,\ldots,\tau\}} \mathbb{E}\left[\left\|\widehat{\psi}_{t,k} - \psi_{t,k}\right\|_{2,2}^2\right] = O\left(r^2 \delta_n^2\right), \quad \delta_n \propto \frac{\log\log(n)}{n}. \tag{57}$$

The theorem uses the notion of critical radius which is often applied to describe the statistical complexity of functional spaces(Wainwright, 2019). Intuitively, the theorem states that when:

- our sequential neural architecture is correctly specified and trained with empirical risk minimization,

- the product norm between nuisance networks has a error small enough,

- the boundness assumption stated above are satisfied,

then our trained neural network for predicting the sequential blip effect will only have an $\Omega(\frac{\log\log(n)}{n})$ root mean squared error rate.

# D. Theoretical justification for the Double Optimization Trick

Our double optimization trick implements the blip estimation stage with a simultaneous gradient update. This section provides the theoretical rationale for this design. We show that our approach is a practical, one-step approximation of an iterative process that is guaranteed to converge to the correct solution.

1. We define an idealized, iterative solution operator and prove that it converges to the true blip coefficients.

2. We derive an error propagation bound for a single application of this operator.

3. We discuss the practical implications of this theory for our model's training and error propagation.

## D.1. Iterative operator for learning Blip estimators

The $L^2$-moment loss from Eq. (5) in the main chapter defines a system of interdependent optimizations. The solution for the blip coefficients at step $k$, i.e. $\psi_{t,k}$, depends on all the future steps $j > k$. This sequential solution scheme is, however, extremely slow. Hence, we replace it with a fixed point iteration scheme, which leads to the same exact solution.

Let $\boldsymbol{\Psi}$ be the space of all possible blip coefficient functions $\boldsymbol{\psi} = (\psi_0, \ldots, \psi_\tau)$. We define an operator $T : \boldsymbol{\Psi} \to \boldsymbol{\Psi}$ that maps an input set of functions $\boldsymbol{\psi}^{(\text{old})}$ to an output set $\boldsymbol{\psi}^{(\text{new})}$. The $k$-th component of the output is defined as the exact minimizer of the loss at step $k$, using the input functions as targets:

$$\psi_{t,k}^{(\text{new})}(h_t) = \underset{\widehat{\psi}_{t,k}(\cdot) \in \boldsymbol{\Psi}}{\arg\min} \mathbb{E}\left[\left(\widetilde{Y}_{t,k} - \sum_{j=k+1}^\tau \psi_{t,j}^{(\text{old})}(h_t)' \widetilde{A}_{t,j,k} - \widehat{\psi}_{t,k}(h_t)' \widetilde{A}_{t,k,k}\right)^2\right] \tag{58}$$

**Our double optimization trick is a practical, gradient-based implementation of a single iteration of this operator.** The input $\boldsymbol{\psi}^{(\text{old})}$, in practice, is just the output of the detached network $\widehat{\boldsymbol{\psi}^2}$ in Eq. (8). The loss function $\mathcal{L}_{\text{blip}} = \sum_k \mathcal{L}_{\text{blip}}^k$ is constructed to optimize the summed targets from Eq.58. The goal of the update step of the double optimization is to make the trained blip prediction network one step closer to the ideal target output $\psi_{t,k}^{(\text{new})}$. Actually, this kind of approach is not a new heuristic. It resembles the policy update in actor-critic methods in reinforcement learning: A single gradient step of the policy is taken to approximate the $\arg\max$ operation on the Q-value function.

**Convergence of the operator:** A fundamental property of this operator is that it looks backward. The output at step $k$, $\psi_{t,k}^{(\text{new})}$, depends only on the input functions from future steps, $j > k$. At the final time step $k = \tau$, the term $\sum_{j=\tau+1}^\tau (\ldots)$ is an empty sum that equals zero. The definition of $\psi_{t,\tau}^{(\text{new})}$ is therefore independent of the input $\boldsymbol{\psi}^{(\text{old})}$. This means for any two different input blip estimators $\boldsymbol{\psi}$ and $\boldsymbol{\phi}$, the operator produces the exact same output for the final step: $T(\boldsymbol{\psi})_\tau = T(\boldsymbol{\phi})_\tau$.

This provides a stable anchor at the end of the horizon. When the operator $T$ is applied again, the output for the second-to-last step, $\psi_{t,\tau-1}$, will now also be uniquely determined, as its only dependency is on the now-fixed final step. This logic propagates backward through all time steps. After $\tau + 1$ applications of the operator, any initial difference between two blip predictors will vanish completely. This means that, starting with an arbitrary $\boldsymbol{\psi}_t^{\text{initial}}$, the output of the operator converges to the true blip coefficient after $\tau + 1$ iterations.

## D.2. Error bound of the output of the operator

In previous section, we have shown the convergence property of the transformation $T$. In practice, our double optimization trick is only an approximation of $T$. Therefore, in this section, we analyze how estimation errors propagate through a single application of the operator T.

### D.2.1. FORMAL ASSUMPTIONS

We invoke two standard regularity assumptions:

1. **Strong Overlap**: The smallest eigenvalue of the conditional covariance matrix $C_k(h_t) = \mathbb{E}[\widetilde{A}_{t,k,k}\widetilde{A}'_{t,k,k}|H_t = h_t]$ is larger than some $\sigma_{\min}^2 > 0$.

2. **Bounded Moments**: The conditional cross-moments of treatment residuals are bounded. There exists a constant $M < \infty$ such that $\mathbb{E}\left[\|\widetilde{A}_{t,k,k}\widetilde{A}'_{t,j,k}\|_2^2|H_t = h_t\right] \leq M^2$.

### D.2.2. DERIVATION OF THE ERROR BOUND

Let $\mathbf{\Delta}^{(\text{in})} = \boldsymbol{\psi}^{(\text{in})} - \boldsymbol{\phi}^{(\text{in})}$ be the difference between two input estimators, and $\mathbf{\Delta}^{(\text{out})} = T(\boldsymbol{\psi}^{(\text{in})}) - T(\boldsymbol{\phi}^{(\text{in})})$ be the difference in their outputs. The closed-form solution to the least-squares problem in Eq. (58) gives the difference at step $k$:

$$\Delta_k^{(\text{out})}(h_t) = \mathbb{E}\left[-C_k(h_t)^{-1}\sum_{j=k+1}^{\tau}\widetilde{A}_{t,k,k}\widetilde{A}'_{t,j,k}\Delta_j^{(\text{in})}(h_t)\,\middle|\,H_t = h_t\right]$$

By taking the squared norm, applying Jensen's inequality, Cauchy-Schwarz, and our assumptions, we can bound the expected error (see derivation below):

$$\mathbb{E}[\|\Delta_k^{(\text{out})}\|^2] \leq C\sum_{j=k+1}^{\tau}\mathbb{E}[\|\Delta_j^{(\text{in})}\|^2] \tag{59}$$

where the constant $C = \frac{\tau M^2}{\sigma_{\min}^4}$ depends on the horizon and the data properties (overlap and moments).

*Proof.*

$$\|\Delta_k^{(\text{out})}(h_t)\|^2 = \left\|\mathbb{E}\left[-C_k(h_t)^{-1}\sum_{j=k+1}^{\tau}\widetilde{A}_{t,k,k}\widetilde{A}'_{t,j,k}\Delta_j^{(\text{in})}(h_t)\,\middle|\,H_t = h_t\right]\right\|^2$$

$$\leq \mathbb{E}\left[\left\|C_k(h_t)^{-1}\sum_{j=k+1}^{\tau}\widetilde{A}_{t,k,k}\widetilde{A}'_{t,j,k}\Delta_j^{(\text{in})}(h_t)\right\|^2\,\middle|\,H_t = h_t\right] \quad\text{(by Jensen's Inequality)}$$

$$\leq \mathbb{E}\left[\|C_k(h_t)^{-1}\|_2^2\left\|\sum_{j=k+1}^{\tau}\widetilde{A}_{t,k,k}\widetilde{A}'_{t,j,k}\Delta_j^{(\text{in})}(h_t)\right\|^2\,\middle|\,H_t = h_t\right]$$

$$\leq \frac{1}{\sigma_{\min}^4}\mathbb{E}\left[(\tau-k)\sum_{j=k+1}^{\tau}\left\|\widetilde{A}_{t,k,k}\widetilde{A}'_{t,j,k}\Delta_j^{(\text{in})}(h_t)\right\|^2\,\middle|\,H_t = h_t\right] \quad\text{(Assumption 1 \& Cauchy-Schwarz)}$$

$$\leq \frac{\tau-k}{\sigma_{\min}^4}\sum_{j=k+1}^{\tau}\mathbb{E}\left[\|\widetilde{A}_{t,k,k}\widetilde{A}'_{t,j,k}\|_2^2\cdot\|\Delta_j^{(\text{in})}(h_t)\|^2\,\middle|\,H_t = h_t\right]$$

$$\leq \frac{(\tau-k)M^2}{\sigma_{\min}^4}\sum_{j=k+1}^{\tau}\|\Delta_j^{(\text{in})}(h_t)\|^2 \quad\text{(Assumption 2)}$$

Taking the expectation over $H_t$ on both sides yields the final bound in Eq. (59). $\qquad\square$

### D.3. Practical Implications

This theory provides two crucial insights for our practical implementation.

First, the error bound confirms the importance of the stable anchor at $k = \tau$. Because the error at any step $k$ depends only on the errors from future steps, if we can learn an accurate estimator for the final step, $\widehat{\psi}_{t,\tau}$, its correctness will propagate backward to stabilize the learning of all preceding estimators. The loss for the final step, $\mathcal{L}_{\text{blip}}^{\tau}$, is exactly the original $L^2$-moment loss $\mathcal{L}_\tau$ in Eq. (5), guaranteeing that $\widehat{\psi}_{t,\tau}$ converges to its true target.

Second, the bound quantifies how errors can accumulate. In practice, our one-step update will not be perfect. Let $\epsilon_k = \mathbb{E}[\|\widehat{\psi}_{t,k} - \psi_{t,k}^*\|^2]$ be the total MSE at step $k$, and assume a minimal unavoidable error of $\delta_k$ at each step (due to finite data or model misspecification). The error recurrence is:

$$\epsilon_k \approx \delta_k + C\sum_{j=k+1}^{\tau}\epsilon_j$$

From this, we can derive a recursive relationship between adjacent steps: $\epsilon_{k-1} \approx (1 + C)\epsilon_k + (\delta_{k-1} - \delta_k)$. Assuming the base error $\delta_k$ is roughly constant ($\delta_{k-1} \approx \delta_k$), this simplifies to $\epsilon_{k-1} \approx (1 + C)\epsilon_k$. Unrolling this recursion from the final step backwards gives:

$$\epsilon_k \approx (1 + C)^{\tau-k}\epsilon_\tau \approx (1 + C)^{\tau-k}\delta_\tau \tag{60}$$

This bound shows that errors can grow exponentially with the distance from the final step, $\tau - k$. This underscores the importance of having strong data overlap (to keep $C$ small) and an accurate model (to keep the base error $\delta_\tau$ small), especially for applications involving long horizons.

# E. Additional Results

### E.1. Ablation studies

We report the ablation studies on **(1)** the neural backbone of the sequential encoder and **(2)** the $L^2$-moment loss with double optimization trick. To this, we further introduce the following baselines:

1. **DeepBlip-LSTM:** We replace the transformer architecture in the sequential encoders $\mathcal{E}_{\theta_N}^N, \mathcal{E}_{\theta_B}^B$ of nuisance network and blip prediction network by the LSTM network. From the experimental results below, we find that DeepBlip-LSTM is still highly effective: it outperforms other baselines, which proves the effectiveness of our proposed DeepBlip framework regardless of the instantiations. From the examples of DeepBlip instantiated by transformer and LSTM, we demonstrate that our DeepBlip framework can seamlessly integrate popular sequential networks.

2. **DeepBlip-WDO:** We remove the double optimization trick (WDO stands for **W**ithout **D**ouble **O**ptimization) and only make a single forward pass on $\widehat{\psi}_t$ to construct the $L^2$-moment loss. In this scenario, we discard the mandate on the order of solving the blip coefficient estimators, which is required for a correct estimation in SNMMs. This way, we are able to identify the contribution of applying the double optimization trick. Our results show that DeepBlip-WDO suffers from high estimation error and thus, demonstrate the necessity of the double optimization trick.

We report the RMSE on both synthetic dataset and semi-synthetic dataset as in the experiment section. We adopt the similar test setting in experiment section, where we test the performance against growing time-varying confounding $\gamma_{\mathrm{conf}}$ in the synthetic data experiment and test the performance against increasing prediction horizons $\tau$ in the semi-synthetic dataset. The corresponding ablation results are shown in Figures 8 and 9.

### E.2. Granular analysis of the blip coefficients prediction

So far, we only demonstrate the overall RMSE on the CATE over time estimation. Our DeepBlip works by predicting the blip coefficients $\widehat{\psi}_t(H_t) = (\widehat{\psi}_{t,0}(H_t), \ldots, \widehat{\psi}_{t,\tau}(H_t)) \in \mathbb{R}^{(\tau+1)d_a}$. In this subsection, we report a granular analysis on the prediction of the blip coefficients, bringing more transparency to our DeepBlip.

In fact, we could directly derive the ground truth personalized blip coefficients $\psi_t = (\psi_{t,0}(H_t), \ldots, \psi_{t,\tau}(H_t))$ for both the synthetic dataset in F.1 and semi-synthetic dataset in F.2. This enables us to accurately evaluate the performances of all the blip coefficient predictors. Specifically, we visualize the distributions of all the components of the difference between the true blip coefficients and the prediction $\psi_t - \widehat{\psi}_t \in \mathbb{R}^{(\tau+1)d_a}$. For synthetic dataset, we set $\gamma = 5$ and $\tau = 2$. For semi-synthetic dataset, we set $\gamma = 1$ and $\tau = 4$. In both settings, we use DeepBlip models instantiated by transformer that achieve the lowest RMSE and DeepBlip models trained without the double optimization trick to further demonstrate its necessity. The results from Figure 10 and Figure 11 show that our DeepBlip model is indeed capable of predicting each blip coefficient with high accuracy while with low variance. This supports our claim that **DeepBlip achieves robust estimation of CATE by decomposing the total effect into incremental blip effects with controlled error**. Further, by comparing the blip prediction of DeepBlip with and without double optimization, we see that the double optimization trick is essential to generating unbiased blip coefficient estimates.

### E.3. Offline evaluation efficiency

We additionally evaluate the wall-clock time required for offline treatment planning. Results are shown in Figure 12. In this setting, methods that predict a conditional average potential outcome must evaluate many candidate treatment sequences, while DeepBlip predicts reusable blip coefficients and identifies the best sequence with a single forward pass. This experiment therefore measures both CATE evaluation between fixed treatment sequences and optimal treatment search as the horizon $\tau$ increases.

### E.4. Sensitivity to nuisance misspecification

To assess robustness to nuisance misspecification, we perturb the estimated outcome nuisance function $p_{t,k}$ and treatment nuisance function $q_{t,j,k}$ separately before training the second-stage blip predictor. The resulting RMSE curves in Figure 13 quantify whether small errors in Stage 1 substantially affect the final CATE estimates. Consistent with the Neyman-orthogonality property, performance should degrade gradually rather than abruptly under mild perturbations.

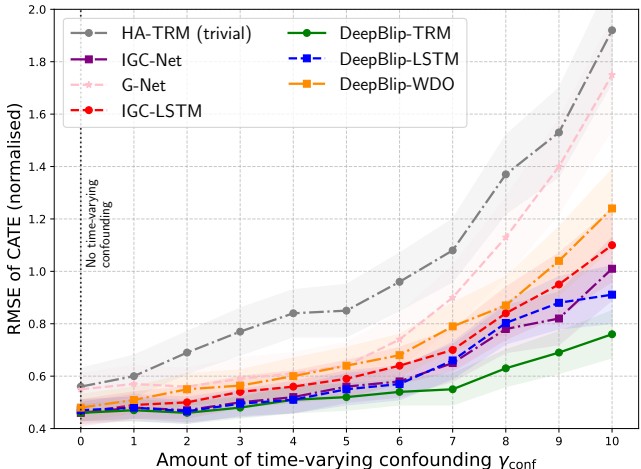

*Figure 8.* Ablation study on the tumor growth synthetic dataset. We explore two ablations: (1) We replace the transformer sequential encoder of DeepBlip and IGC-Net by LSTMs to create **DeepBlip-LSTM** and **IGC-LSTM**. (2) We remove the double optimization trick to create **DeepBlip-WDO**. As in the previous experiment, we report the average RMSE against increasing confounding $\gamma_{conf}$. We note that DeepBlip-LSTM is still more competitive than other baselines, which indicates the *effectiveness of the model-agnostic framework of DeepBlip*. DeepBlip-TRM, as our originally proposed model, remains the strongest.

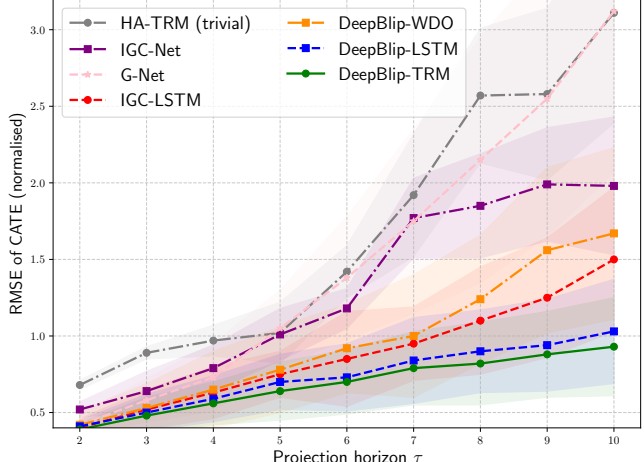

*Figure 9.* Ablation study on the MIMIC semi-synthetic dataset. Same ablations are investigated using the average RMSE against increasing projection horizon $\tau$. Even though DeepBlip-LSTM slightly underperforms the original model DeepBlip-TRM, it is still superior over other baselines. Hence, our proposed DeepBlip framework is *robust* over longer prediction horizon regardless of the backbone. We also observe that DeepBlip-WDO fails to predict CATE as accurately as the proposed DeepBlip-TRM. This shows that the adoption of a stable $L^2$-moment loss, together with the double optimization trick, is crucial for learning the blip coefficients unbiasedly.

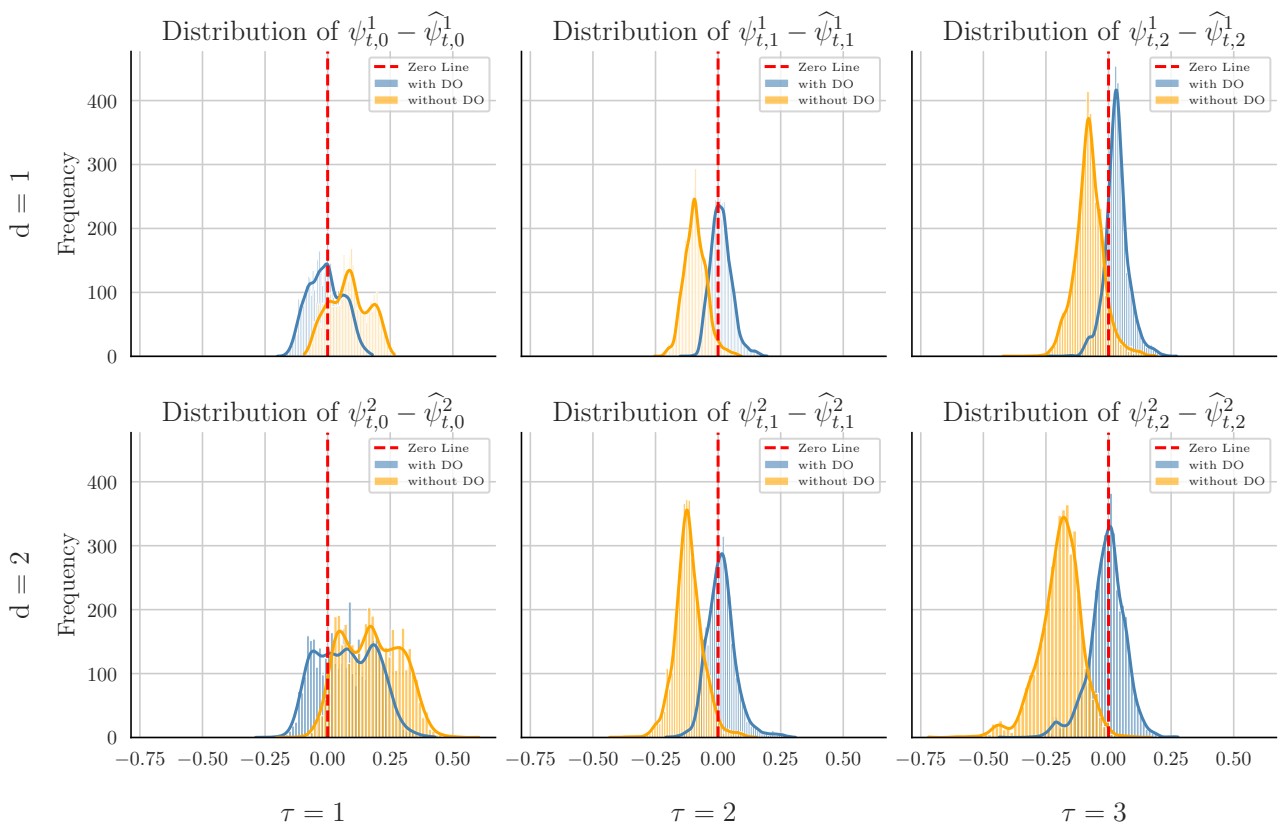

*Figure 10.* Tumor synthetic dataset ($\gamma_{\text{conf}} = 5$, $\tau = 2$): We visualize the distribution of the difference between the ground truth blip coefficients and the predictions made by our blip prediction network. Observing the histogram marked in blue represented by our DeepBlip, we find that the blip prediction network unbiasedly predict the blip coefficients, which offers proper adjustment for the confounding.

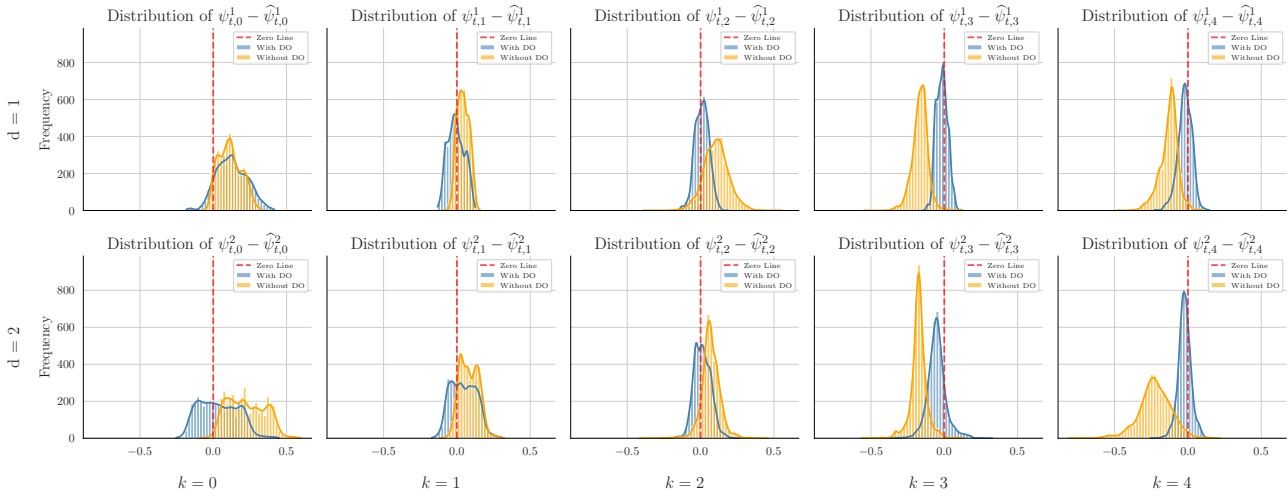

*Figure 11.* MIMIC semi-synthetic dataset ($\gamma_{\text{conf}} = 1$, $\tau = 4$): We conduct the same visualization. When $k$ grows, the prediction gets more accurate. We expect this to happen, since $\widehat{\psi}_{t,\tau}$ is the first to optimize by the double optimization trick and then $\widehat{\psi}_{t,\tau-1}$ and so on. In contrast, the blip prediction network trained without double optimization exhibits significant bias and generally higher variance. This implies that the double optimization trick is crucial for the training DeepBlip.

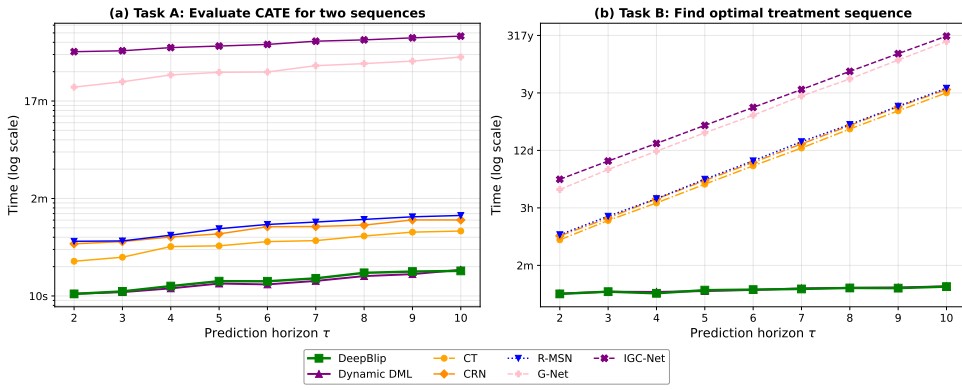

*Figure 12.* Offline evaluation efficiency over increasing prediction horizons. DeepBlip remains nearly constant because it reuses the predicted blip coefficients, whereas baseline methods require exhaustive evaluation over candidate treatment sequences.

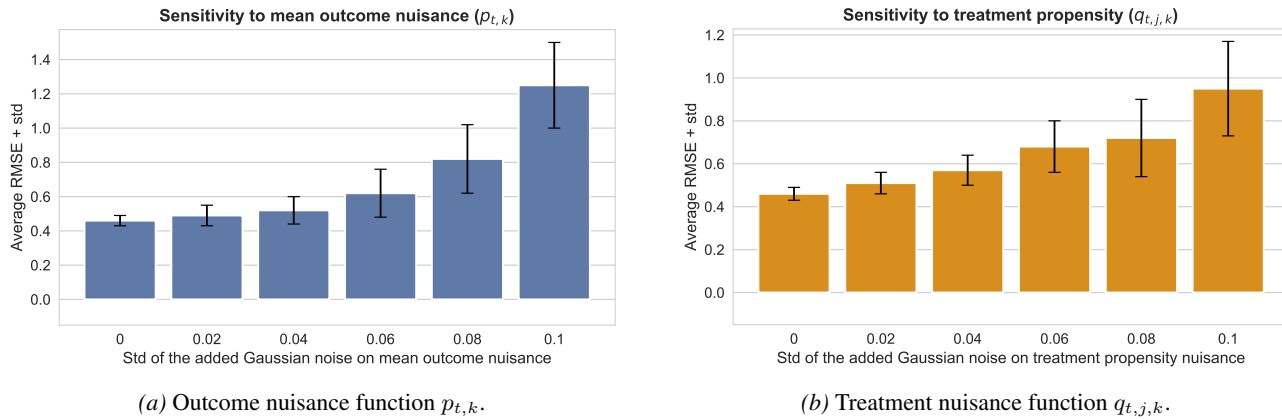

*(a)* Outcome nuisance function $p_{t,k}$.       *(b)* Treatment nuisance function $q_{t,j,k}$.

*Figure 13.* Sensitivity analysis for nuisance misspecification in the outcome and treatment nuisance functions.

## F. Dataset

In this section we will give a more detailed elaboration of the data generating process as well as the key simulation of the counterfactual outcome (namely the potential outcome). The treatment effect is then achieved by subtracting two counterfactual outcomes.

### F.1. Tumor growth dataset

We specify the dynamic of the tumor growth model to be:

$$Y_{t+1} = Y_t + \underbrace{\left(\rho \log(\frac{K}{m(\overline{Y}_{t-l})}) + \epsilon_{t+1}\right)m(\overline{Y}_{t-l})}_{\text{Tumor Growth}} - \underbrace{m(\overline{Y}_{t-l})\alpha_c c_{t+1}}_{\text{Chemotherapy}} + \underbrace{m(\overline{Y}_{t-l})(\alpha_r d_{t+1} + \beta_r d_{t+1}^2)}_{\text{Radiotherapy}} \qquad (61)$$

$\alpha_c, \alpha_r, \beta_r$ are the coefficients on the treatment effect of chemotherapy and radiotherapy, $\rho, K$ together control the growth model. $m(\overline{Y}_{t-l})$ represents the mean of tumor volumes $Y_1, .., Y_{t-l}$, where $l$ is a lag parameter. The averaging of the volumes avoids abrupt changes for the tumor volume. The chemotherapy drug dosage $c_t$ and the radiation dosage $d_t$ are applied with probability:

$$A_t^c, A_t^d \sim \text{Ber}\left(\sigma \cdot \left(\frac{\gamma_{conf}}{D_{max}}(\overline{D}_{15}(\overline{Y}_t) - \frac{D_{max}}{2}))\right)\right) \qquad (62)$$

Since the probability distribution is determined on the mean of previous 15 steps' tumor volumes $\overline{D}_{15}(\overline{Y}_t)$, time-varying confounding exists through future outcomes with its influence controlled by the strength parameter $\gamma_{conf}$.

To generate the observational dataset, we simulate the trajectories of $N = 1000$ patients with maximal sequence length

$T = 30$ under confounding strength $\gamma_{conf} \in \{0, 1, \ldots, 10\}$, resulting in 11 datasets $\mathcal{D}_{N,T,\gamma_{conf}}$. Then we split the simulated dataset into training, validation and test set by a split of $(70\%, 15\%, 15\%)$.

**Simulating counterfactuals:** For the test dataset, we compute the ground truth CATE by simulating the counterfactual outcomes $Y_{t+\tau}^{a^*}$ and $Y_{t+\tau}^{b^*}$ via a sliding window treatment along the sequence. To simulate the counterfactual outcome $Y_{t+\tau}^{(\mathbf{a}^*)}$ under an intervention $\mathbf{a}^* \in \mathbb{R}^{(\tau+1) \times 2}$ during $[t, t+\tau]$ for the tumor growth model, we iteratively compute the counterfactual trajectory starting from the observed history $H_t$. The DGP of outcome is constituted by the lagged dependency on tumor volume averages and the cumulative effects of treatments. Here we provide a strict formulation of the counterfactual outcome generation process, including the mathematical formulation and the steps for simulation.

Given the observed history $H_t = (\overline{X}_t, \overline{A}_{t-1}, \overline{Y}_{t-1})$, the counterfactual trajectory $\{Y_s^{(\mathbf{a}^*)}\}_{s=t}^{t+\tau}$ under intervention $\mathbf{a}^* = [a_t^*, a_{t+1}^*, \ldots, a_{t+\tau}^*]$ (where $a_s^* = (c_s^*, d_s^*)$) is generated with the following steps:

1. **Initialization**: Get observed history: $Y_s$ for $s < t$. Note $Y_{t-1}^{(\mathbf{a}^*)} = Y_{t-1}$.

2. **Iterative simulation**: For each time $s \in [t, t+\tau-1]$, we simulate $Y_{s+1}^{(\mathbf{a}^*)}$ based on previous simulated outcomes. Notice that since $l > \tau$, $s - l < t$, then $m(\overline{Y}_{s-l})$ is unaffected by the interventions and therefore could be treated as constant when conditioning on $H_t$. Therefore we have:

$$Y_{s+1}^{(\mathbf{a}^*)} = Y_s^{(\mathbf{a}^*)} + \underbrace{\rho \log\left(\frac{K}{m(\overline{Y}_{s-l})}\right) m(\overline{Y}_{s-l}) + \epsilon_{s+1}}_{\text{Growth Term}}$$
$$- \underbrace{m(\overline{Y}_{s-l}) \alpha_c c_{s+1}^*}_{\text{Chemotherapy Effect}} - \underbrace{m(\overline{Y}_{s-l})(\alpha_r d_{s+1}^* + \beta_r (d_{s+1}^*)^2)}_{\text{Radiotherapy Effect}}, \quad (63)$$

    where $\epsilon_{s+1}$ is the **original** noise term during the generation of the observational data. $c_{s+1}^*, d_{s+1}^*$ are the intervened chemotherapy and radiotherapy dosages at $s + 1$, drawn from $\mathbf{a}^*$

Under this iterative scheme, we could compute the final counterfactual outcome $Y_{t+\tau}^{(\mathbf{a}^*)}$.

**Treatment effect:** We then proceed to compute the individual conditioned treatment effect for a patient under treatment $\mathbf{a}^*$ and the null treatment $\mathbf{0}$. Observing the process in (63), we find that the growth term **cancels off** each other during the subtraction, leaving only the terms for chemotherapy effect and radiotherapy effect:

$$Y_{t+\tau}^{(\mathbf{a}^*)} - Y_{t+\tau}^{(\mathbf{0})} = \sum_{s=t}^{t+\tau} \left\{ m(\overline{Y}_{s-l}) \alpha_c c_{s+1}^* + m(\overline{Y}_{s-l})(\alpha_r d_{s+1}^* + \beta_r (d_{s+1}^*)^2) \right\} \quad (64)$$

The individual treatment effect is calculated for all the units on the test dataset $D_{test}$ in a sliding treatment window rolling over $t \in \{1, .., T-\tau\}$, producing $N_{test} \times (T-\tau)$ targets of individual treatment effect. We use these individual treatment effects as realizations of the CATE target $\mathbb{E}[Y_{t+\tau}^{(\mathbf{a}^*)} - Y_{t+\tau}^{(\mathbf{0})} \mid H_t]$.

### F.2. MIMIC-III semi-synthetic dataset

**MIMIC Data and Semi-Synthetic Benchmark** We use the **MIMIC-Extract** pipeline (Wang et al., 2020) to preprocess the **MIMIC-III** ICU dataset (Johnson et al., 2016), aggregating hourly clinical measurements. Missing values are addressed via forward and backward filling, and continuous time-varying covariates are normalised. Our semi-synthetic dataset extends Schulam & Saria (2017) to achieve ground-truth treatment effects.

**Cohort & Features:** A cohort of 1,000 patients is selected, restricted to ICU stays of 50–100 hours ($T^{(i)} \in [50, 100]$). The feature space includes 18 time-varying vital signs and 3 static features (gender, ethnicity, age), where categorical variables are one-hot encoded. These variables, dynamic and static, are concatenated together to form a covariate vector $X \in \mathcal{X}$ with $d_x = \dim(\mathcal{X}) = 18 + 2 + 3 + 1 = 25$.

**Untreated Outcomes:** For each patient $i$, the untreated trajectory $\mathbf{Z}_t^{(i)}$ are simulated as:

$$Z_t^{(i)} = \underbrace{\alpha_s \text{B-spline}(t) + \alpha_g g^{(i)}(t)}_{\text{Endogenous}} + \underbrace{\alpha_f f_Z(X_t^{(i)})}_{\text{Exogenous}} + \underbrace{\varepsilon_t}_{\text{Noise}}, \quad \varepsilon_t \sim \mathcal{N}(0, 0.005^2), \quad (65)$$

where $\alpha_S^j, \alpha_g^j, \alpha_f^j \in [0,1]$ control component weights. The **endogenous** term combines: (1) a global trend $\mathbf{B}$-spline$(t)$ sampled from three cubic splines; (2) patient-specific local variations $g^{j,(i)}(t)$ from a Gaussian process with Matérn kernel (length-scale $\ell = 2.0$). The **exogenous** term is modeled by the effect $f_Z(X_t^{(i)})$ via random Fourier features (RFF), which approximates Gaussian process. By combining the three terms we are able to simulate a complex untreated outcome that has a intrinsic trend modeled by (1) endogenous dependencies on a different scale (B-spline for the global trend and RFF for the local perturbations), and (2) an exogenous dependency on the time-varying covariate $X_t$. The untreated outcome model presents a challenging task which requires complex deep neural networks.

**Treatment Assignment:** 2 binary treatments $\mathbf{A}_t^l, l \in \{1, 2\}$ are assigned sequentially using:

$$p_t^l = \sigma\left(\gamma_A^l m(Y_{t-w:t-1}) + \gamma_X^l f_X^l(X_t) + b^l\right), \quad A_t^l \sim \text{Bernoulli}(p_t^l), \tag{66}$$

where $(\gamma_A^1 = 2.5, \gamma_X^1 = 1.0, \gamma_A^2 = 2.0, \gamma_X^2 = 0.25)$ control confounding strength, $b = (-3.5, -1.75)$ biases treatment probability, and $f_X^l(\cdot)$ uses RFF approximations of Gaussian process similar to $f_Y$. The term $m(Y_{t-w:t-1})$ computes the average of prior outcomes over a window $[t - w, t - 1]$.

**Treatment Effects:** Treatments affect outcomes via cumulative effects that decay over time:

$$E(t) = \sum_{i=t-w}^{t} \sum_{l=1}^{2} \frac{\beta_l A_i \kappa^l(X_i)}{\sqrt{t-i+1}}, \quad \text{where } \kappa^l(X_i) = tanh((\omega^l)'X_i) + 1 \tag{67}$$

where $w = 5$ defines the effect window, and the inverse square root decay ensures maximal effect $\beta_l$ at application time $t$. Effects are summed across all treatments to model the local treatment effect at time $t$. Sparse dependencies are enforced: Outcomes are affected by at most 3 covariates. The function $\kappa^l$ further provides heterogeneity to the treatment effect by scaling the full effect $\beta$, parametrized by the coefficients $\omega^l \in \mathbb{R}^{d_x}$ (also sparse). These arrangements **increase** the challenge to estimate treatment effect compared to previous benchmarks (Hess et al., 2026a; Melnychuk et al., 2022).

**Observed outcomes:** Finally, we combine the untreated outcome and the treatment effects together to form the observed outcome trajectories:

$$Y_t^i = Z_t^i + E(t)^i \tag{68}$$

**Simulating counterfactuals:** To generate the counterfactual outcome under an intervention $\mathbf{a}^* \in \mathbb{R}^{(\tau+1)\times 2}$ during $[t, t+\tau]$, we intervene on the treatment effect $E(t)$ by replacing the observed treatments $A_t$ by the intervened treatments $\mathbf{a}^*$ while keeping the untreated trajectory $Z_t$ and covariates $X_t$ unchanged (note that the covariates $X_t$ are predefined vital signs which remain unaffected during the DGP). The counterfactual outcome $Y_{t+\tau}^{\text{CF}}$ at time $s \in [t, t + \tau]$ is:

$$Y_{t+\tau}^{(\mathbf{a}^*)} \mid H_t = Z_{t+\tau} + \underbrace{\sum_{i=t+\tau-w}^{s} \sum_{l=1}^{2} \frac{\beta_l \cdot a_i^{*,l} \cdot \kappa^l(X_i)}{\sqrt{t+\tau-i+1}}}_{E^*(t+\tau)},$$

where: $Z_{t+\tau}$ is the untreated outcome (identical to the observed scenario). The treatments are divided by intervention period and the observed period (history)

$$A_i^{*,l} = \begin{cases} a_i^{*,l} & \text{if } t \le i \le t + \tau \text{ (intervention period)}, \\ A_i^l & \text{otherwise (observed treatment)}. \end{cases}$$

And $\kappa^l(X_i) = \tanh((\omega^l)^\top X_i) + 1$ encodes effect heterogeneity via covariates. The simulation is performed for the units only on the **test** set in a sliding window treatment pattern. This means we start by conditioning on $H_1$ to simulate counterfactual outcome at time $\tau + 1$ and end at $t = T - \tau$ to simulate counterfactual outcome at time $T$. This results in $N_{test} \times (T - \tau)$ counterfactual outcomes under intervention $\mathbf{a}^*$.

The full implementation is available in our GitHub repository, including parameter configurations for reproducibility.

# G. Baseline methods

We select six baselines to demonstrate the performance of our DeepBlip for CATE estimation over time. They are (1) History adjusted plug-in learner with LSTM instantiation (**HA-TRM**) (Frauen et al., 2025), (2) recurrent marginal structural networks (**R-MSNs**) (Lim, 2018), (3) counterfactual recurrent network (**CRN**), (Bica et al., 2020) (4) **G-Net** (Li et al., 2021), (5) **IGC-Net** (Hess et al., 2026a), and (6) causal transformer (**CT**) (Melnychuk et al., 2022). We will provide details for each baseline and briefly introduce the theory framework of MSM and G-computation. The hyperparameter tuning is detailed in Appendix I.5 and Appendix I.6.

### G.1. History Adjusted with transformer (HA-TRM)

A naïve approach for estimating CATE over time is to create a regressor with the treatment as the condition:

$$\hat{\delta}_\theta(a_{t:t+\tau}, h_t) \approx \mathbb{E}[Y_{t+\tau} \mid A_{t:t+\tau} = a_{t:t+\tau}, H_t = h_t]$$

where $\hat{\delta}_\theta$ is a non-parametric model (like neural networks). Then we estimate the CATE of $\boldsymbol{a}^*$ and $\boldsymbol{b}^*$ as:

$$\hat{\delta}_\theta(\boldsymbol{a}^*, h_t) - \hat{\delta}_\theta(\boldsymbol{b}^*, h_t) \approx \mathbb{E}[Y_{t+\tau} \mid A_{t:t+\tau} = \boldsymbol{a}^*, H_t = h_t] - \mathbb{E}[Y_{t+\tau} \mid A_{t:t+\tau} = \boldsymbol{b}^*, H_t = h_t]$$

This method is given the name **PI-HA learner** (plugged-in history adjusted meta-learner) from the work of Frauen et al. (2025). We instantiate this method with the transformer architecture to encode the history into a latent representation (see Appendix I), which is then fed into a fully connected linear network to predict the outcome.

Since this approach does not adjust for time-varying confounding, it is biased (Frauen et al., 2025). While an advantage of the HA-TRM is low-variance, it is subject to the level of confounding within the dataset as we observed in the experiment.

### G.2. Marginal Structural Models (MSM) and its neural Extension

**MSM Framework:** Marginal Structural Models (MSMs) (Hernán et al., 2001; Robins et al., 2000) address time-varying confounding via inverse probability of treatment weighting (**IPTW**). IPTW re-weights the targets to create a pseudo population that approximates the randomized controlled trial. With projection horizon $\tau$, the stabilized weight (**SW**) at time $t$ for each sample is defined as:

$$SW(t, \tau) = \prod_{n=t}^{t+\tau} \frac{f(\mathbf{A}_n | \bar{\mathbf{A}}_{n-1})}{f(\mathbf{A}_n | \bar{\mathbf{H}}_n)}, \tag{69}$$

where $f(\mathbf{A}_n | \bar{\mathbf{A}}_{n-1})$ and $f(\mathbf{A}_n | \bar{\mathbf{H}}_n)$ represent treatment probabilities conditioned on past treatments or full history. These treatments are assumed discrete, often binary. The probabilities are estimated via logistic regressions. In practice the weights are truncated at 1st and 99th percentiles to avoid numeric overflow and then renormalised (Lim, 2018). However, this division of probabilities still creates significant instability due to the sequential multiplication of the propensities in Eq. (69).

**Recurrent Marginal Structural networks (R-MSNs):** R-MSNs (Lim, 2018) extend MSMs via LSTM networks. Four components constitute the whole R-MSNs: propensity treatment network, propensity history network, encoder, decoder. Each sub-network uses LSTM as the sequential modeling architecture and a fully connected network to produce the desired prediction. Here is a detailed description of each model:

- *Propensity Network (conditioned on past treatment)* estimates numerator of $SW(t, \tau)$.

- *Propensity Network (conditioned on history)* estimates denominator of $SW(t, \tau)$.

- *Encoder* maps history $\bar{\mathbf{H}}_t$ to latent representation

- *Decoder* predicts $\tau$-step outcomes using the representation generated by the encoder from last step.

In our experiment, we replace the LSTM backbone of the encoder and decoder by transformers to ensure fairness. Training occurs in 3 phases: (1) First the two propensity networks are trained to predict the binary treatment probabilities using binary cross entropy losses. Then the stabilized weights can be computed for subsequent training of encoder and decoder. (2) Encoder is trained to predict the 1-step ahead outcome with the computed $SW(\cdot, 1)$-weighted MSE. (3) The decoder takes the latent representation from the encoder and then transform the vector into the latent representation fit for the decoder

via a memory adapter (normally a fully connected linear layer). The decoder then predict the $\tau$-step outcome with the $SW(\cdot, \tau)$-weighted MSE.

In each phase, the training loss is aggregated through averaging the local loss at each time step $0 \leq t \leq T - \tau$ (identical to our DeepBlip). At each local time step $t$, a representation is built with the transformer for the tasks. For more details please refer to Lim (2018).

### G.3. Causal estimation via balanced representations

**Counterfactual recurrent network (CRN):** CRN (Bica et al., 2020) employs adversarial learning to create treatment-invariant representations. CRN heuristically addresses the time-varying confounding through creating temporal representations that are non-predictive of the treatment assignment. For this an LSTM encoder-decoder architecture with gradient reversal layers is built to enforce balanced representation:

- *Encoder*: LSTM produces hidden states $\mathbf{h}_t$

- *Balanced Representation*: $\boldsymbol{\Phi}_t = \text{FC}(\mathbf{h}_t)$

- *Adversarial Loss*:

$$\mathcal{L} = \|\mathbf{Y}_{t+1} - G_Y(\boldsymbol{\Phi}_t, \mathbf{A}_t)\|^2 - \lambda \sum_{j=1}^{d_a} \mathbf{1}_{[\mathbf{A}_t = a_j]} \log G_A(\boldsymbol{\Phi}_t) \tag{70}$$

Here, $G_Y$ predicts outcomes while $G_A$ attempts (but fails due to gradient reversal) to predict treatments from $\boldsymbol{\Phi}_t$. We fix $\lambda = 1$ as in original implementations.

**Causal transformer (CT):** CT (Melnychuk et al., 2022) is the first transformer-based approach that estimates potential outcomes over time. CT employs a multi-input transformer architecture specifically designed to create history representations $\boldsymbol{\Phi}_t$ under time-varying treatment effect estimation setting. On top of $\boldsymbol{\Phi}_t$, two additional networks are built:

- $G_Y$: The outcome prediction network, which predicts the 1-step-ahead outcome $\mathbf{Y}_{t+1}$ using $\boldsymbol{\Phi}_t$ and the current treatment $\mathbf{A}_t$.

- $G_A$: The treatment classifier network, which attempts to predict the current treatment $\mathbf{A}_t$ from $\boldsymbol{\Phi}_t$.

Similar to CRN, CT attempts to create representations that are **predictive of outcomes** but **non-predictive of treatment assignments**, thereby mitigating confounding bias. To this, CT proposes counterfactual domain loss (CDC). The loss consists of: (1) Factual Outcome Loss: $\mathcal{L}_{G_Y} = \|\mathbf{Y}_{t+1} - G_Y(\boldsymbol{\Phi}_t, \mathbf{A}_t)\|^2$ This mean squared error ensures that $\boldsymbol{\Phi}_t$ is useful for predicting $\mathbf{Y}_{t+1}$. (2): Confusion Loss: $\mathcal{L}_{\text{conf}} = -\sum_{j=1}^{d_a} \frac{1}{d_a} \log G_A(\boldsymbol{\Phi}_t)$ This cross-entropy loss encourages $G_A$ to output a uniform distribution over treatments, making $\boldsymbol{\Phi}_t$ treatment-invariant.

The training of CT involves optimizing two adversarial objectives:

- Optimize representation parameters ($\theta_R$) and $G_Y$ parameters ($\theta_Y$): $(\hat{\theta}_Y, \hat{\theta}_R) = \underset{\theta_Y, \theta_R}{\arg\min} \mathcal{L}_{G_Y}(\theta_Y, \theta_R) + \alpha \mathcal{L}_{\text{conf}}(\hat{\theta}_A, \theta_R)$

- Optimize the $G_A$ parameters ($\theta_A$): $\hat{\theta}_A = \underset{\theta_A}{\arg\min} \alpha \mathcal{L}_{G_A}(\theta_A, \hat{\theta}_R)$

During inference, CT autoregressively predicts outcomes $\widehat{Y}_{t+k}$ using the balanced representations $\boldsymbol{\Phi}_{t+k}$ based on the observed history $H_t$.

### G.4. G-Computation-based network (G-Net, IGC-Net)

Both methods are based on the G-computation formula (Robins, 1986; Robins et al., 1999):

$$\mathbb{E}\left[Y_{t+\tau}^{(a_{t:t+\tau})} \mid H_t = h_t\right] = \mathbb{E}\left\{\mathbb{E}\left[\cdots\mathbb{E}\left\{\mathbb{E}\left[Y_{t+\tau} \mid H_{t+\tau}, A_{t:t+\tau} = a_{t:t+\tau}\right]\right.\right.\right.$$

$$\left.\left|H_{t+\tau-1}, A_{t:t+\tau-1} = a_{t:t+\tau-2}\right\}\right. \tag{71}$$

$$\left.\cdots\left|H_{t+1}, A_{t:t+1} = a_{t:t+1}\right]\right|H_t = \bar{h}_t, A_t = a_t\right\}$$

Due to the nested structure of G-computation in 71, the model misspecification error get easily propagated through the iterative steps, leading to potential bias and high variance.

**G-Net (Simulation-based)** G-Net(Li et al., 2021) is the first neural network model that uses G-computation to esimate CAPO over time. It make the Monte-Carlo simulation based on the integration:

$$\int_{\mathbb{R}^{d_x \times (\tau)} \times \mathbb{R}^{d_y \times (\tau)}} \mathbb{E}\left[Y_{t+\tau} \mid H_{t+\tau} = h_{t+\tau-1}, A_{t:t+\tau} = a_{t:t+\tau}\right]$$

$$\times \prod_{\delta=1}^{\tau} p\left(x_{t+\delta}, y_{t+\delta} \mid H_t = h_t, x_{t+1:t+\delta-1}, y_{t:t+\delta-1}, a_{t:t+\delta-1}\right) \tag{72}$$

$$d(x_{t+1:t+\tau-1}, y_{t:t+\tau-1})$$

G-Net predicts the potential outcome in two steps:

1. Estimate the conditional distribution $p\left(x_{t+\delta}, y_{t+\delta} \mid H_t = h_t, x_{t+1:t+\delta-1}, y_{t:t+\delta-1}, a_{t:t+\delta-1}\right)$

2. Compute the empirical sum of M (M=50) trajectories via Monte-Carlo simulation to estimate the integration in 72.

G-Net originally uses LSTM to encode the history into a hidden state, with an extra linear transformation layer mapping the hidden state to the latent representation. We replace the LSTM architecture by a transformer in our experiment to ensure fairness. The representation is fed into output computing head (Fully connected networks with respective activation output) to predict the conditional distribution. For fairness, we replace the LSTM encoder with the encoder of the transformer.

**IGC-Net (regression-based)** Unlike G-Net, which uses Monte-Carlo simulation, IGC-Net (Hess et al., 2026a) is built via an iterative regression on the pseudo outcomes defined as:

1. $G_{t+\tau}^a = Y_{t+\tau}$
2. $G_{t+\delta}^a = \mathbb{E}[G_{t+\delta+1}^a \mid \bar{H}_{t+\delta}^t, A_{t:t+\delta} = a_{t:t+\delta}]$ for $\delta = 0, .., \tau - 1$

IGC-Net predicts the outcome in a masked transformer encoder. It uses transformer to encode the history at $t + \delta$ into a representation $Z_{t+\delta} = z_\phi(h_{t+\delta})$, and then predicts the pseudo outcome at step $\delta$ as:

$$g_\phi^\delta(Z_{t+\delta}^a, A_{t+\delta}) \approx G_{t+\delta}^a$$

And then the CAPO is predicted as $g_\phi^0(Z_t, a_t)$. The encoder $z_\phi(\cdot)$ is instantiated by a transformer architecture similar to CT (For more details see Hess et al. (2026a)).

### G.5. Dynamic DML

The Dynamic DML method (heterogeneous version) is implemented as originally designed (See link to the code (Supplemental material)), using sequential g-estimation to solve the blip predictors. Since the original method was developed for static covariates, we adapt it to our time-series setting by concatenating all variables in the patient history $H_t$ into a single, high-dimensional feature vector. This vector then serves as the conditioning variable for the Dynamic DML model. To ensure fairness, the nuisance residuals $(\widetilde{Y}, \widetilde{A})$ are taken directly from our trained Stage 1 nuisance network, providing both methods with inputs of the same quality. However, in its second stage, Dynamic DML is limited because it can only accept a history variable from a fixed time step. This limitation prohibits the use of a complex neural network in scenarios where one needs to take dynamic patient history as input. Hence, we follow the original implementation and use the LASSO regression to model the blip coefficient predictor.

# H. Algorithm

---

**Algorithm 1** DeepBlip training algorithm

---

1: **Input:** Dataset $\mathcal{D}$, horizon $\tau$, feature map $\phi$, number of folds $W$, learning rates $\eta_N, \eta_B$
2: **Output:** Blip prediction network $(\mathcal{E}^B_{\theta_B}, \mathrm{gb}^k_{\theta_B})\widehat{\psi}_{\theta_B}$
3: ───────────────────────────────────────────────────────
4: **First Stage: Train Nuisance Network** $(\mathcal{E}^N_{\theta_N}, \mathbf{gp}^k_{\theta_N}, \mathbf{gq}^{j,k}_{\theta_N})$
5: Partition $\mathcal{D}$ into $W$ folds: $S_1, S_2, \ldots, S_W$
6: **for** each fold $w = 1$ to $W$ **do**
7:     Train nuisance network on $\mathcal{D} \setminus S_w$
8:     **for** each $t = 1$ to $T - \tau$, $k = 0$ to $\tau$ **do**
9:         $z^N_{t+k} \leftarrow \mathcal{E}^N_\theta(h_{t+k})$
10:         $\hat{p}_k(z^N_{t+k}) \leftarrow \mathrm{gp}^k_{\theta_N}(z^N_{t+k})$
11:         $\mathcal{L}^p_{t,k} = (Y_{t+\tau} - \hat{p}_k(z^N_{t+k}))^2$
12:         **for** j = k to $\tau$ **do**
13:             $\hat{q}_{j,k}(z^N_{t+k}) \leftarrow \mathrm{gq}^{j,k}_{\theta_N}(z^N_{t+k})$
14:             $Q_{t,j} = \phi(X_{t+1:t+j}, A_{t:t+j}) - \phi(X_{t+1:t+j}, (A_{t:t+j-1}, 0))$
15:             $\mathcal{L}^q_{t,j,k} = BCE(\hat{q}_{j,k}(z^N_{t+k}), Q_{t,j})$ (if binary treatment)
16:         **end for**
17:     **end for**
18:     $\mathcal{L}_N \leftarrow \frac{1}{T-\tau} \sum_{t=1}^{T-\tau} \left( \frac{1}{\tau+1} \sum_{k=0}^{\tau} \mathcal{L}^p_{t,k} + \frac{2}{(\tau+1)(\tau+2)} \sum_{\tau \geq j \geq k \geq 0} \mathcal{L}^q_{t,j,k} \right)$
19:     {Compute the gradient update the nuisance network parameter $\theta_N$}
20:     $\theta_N \leftarrow \theta_N - \eta_N \nabla_{\theta_N} \mathcal{L}_N$
21:     **for** each sample in $S_w$ **do**
22:         Compute held-out residuals for cross-fitting:
23:         $\widetilde{Y}_{t,k} = Y_{t+\tau} - \mathrm{gp}^k_{\theta_N}(z^N_{t+k})$
24:         $\widetilde{A}_{t,j,k} = A_{t+j} - \mathrm{gq}^{j,k}_{\theta_N}(z^N_{t+k})$
25:     **end for**
26: **end for**
27: ───────────────────────────────────────────────────────
28: **Second Stage: Train Blip Prediction Network** $(\mathcal{E}^B_{\theta_B}, \mathrm{gb}^k_{\theta_B})$
29: **for** each $t = 1$ to $T - \tau$ **do**
30:     Compute the hidden state of the encoder: $z^B_t \leftarrow \mathcal{E}^B_{\theta_B}(h_t)$
31:     **for** $k = 0$ to $\tau$ **do**
32:         $\hat{\psi}^1_k(z^B_t) \leftarrow \mathrm{gb}^k_{\theta_B}(z^B_t)$
33:         $\hat{\psi}^2_k(z^B_t) \hookleftarrow \mathrm{gb}^k_{\theta_B}(z^B_t)$
34:         $\mathcal{L}_{\mathrm{blip},t,k} = \left( \widetilde{Y}_{t,k} - \sum_{j=k+1}^{\tau} \hat{\psi}^2_j(z^B_t)' \widetilde{A}_{t,j,k} - \hat{\psi}^1_k(z^B_t)' \widetilde{A}_{t,k,k} \right)^2$
35:     **end for**
36: **end for**
37: $\mathcal{L}_{\mathrm{blip}} \leftarrow \frac{1}{T-\tau} \sum_{t=1}^{T-\tau} \left( \frac{1}{\tau+1} \sum_{k=0}^{\tau} \mathcal{L}_{\mathrm{blip},t,k} \right)$
38: Compute the gradient update the blip prediction network parameter $\theta_B$
39: $\theta_B \leftarrow \theta_B - \eta_B \nabla_{\theta_B} \mathcal{L}_{\mathrm{blip}}$
40: ───────────────────────────────────────────────────────

*Note:* We use $\leftarrow$ operation for value assignment in the computation graph while $\hookleftarrow$ operations are detached from the computation graph.

---

**Algorithm 2** DeepBlip inference algorithm

---

1: **Input:** Patient history $H_t = h_t$, treatment sequence $a^*$, baseline treatment sequence $b^*$
2: **Output:** CATE estimate $\mathbb{E}[Y_{t+\tau}^{(a^*)} - Y_{t+\tau}^{(b^*)} \mid H_t = h_t]$
3: **Step 1: Compute the hidden state of the encoder**
4: $z_t^B \leftarrow \mathcal{E}_{\theta_B}^B(h_t)$
5: **Step 2: Predict the blip coefficients**
6: $\hat{\psi}_k(z_t^B) \leftarrow \text{gb}_{\theta_B}^k(z_t^B)$
7: **Step 3: Estimate CATE**
8: (when $\phi$ only depends the last treatment: $\phi(x_{t+1:t+k}, a_{t:t+k}) = \phi(a_{t+k})$)
9: CATE $= \sum_{k=0}^{\tau} \hat{\psi}_k(z_t^B)'(\phi(a_k^*) - \phi(b_k^*))$

---

# I. Implementation details

In subsection I.2, we introduce the runtime of our system. We provide a runtime comparison of the different methods in subsection I.2. Then in subsection I.3 and I.4, we introduce how to instantiate our DeepBlip with transformers and LSTM respectively. In subsection I.5 and subsection I.6, we report the details of hyperparameter tuning. Tables 4 and 5 summarize the search ranges used for the tumor growth and MIMIC-III experiments, respectively.

## I.1. Software & system

Our DeepBlip is implemented using **PyTorch** (Paszke et al., 2019), combined with **PyTorch Lightning** (Falcon & The PyTorch Lightning team, 2019) for streamlined code organization and model training. Our code is available at https://github.com/mHaoruikk/DeepBlip. Training and inference are performed on a single GeForce RTX 4080 with 18GB memory.

## I.2. Runtime comparison

In this section we report: (1) The runtime for training a single epoch on the training set, (2) The runtime for inference on the test set, and (3) The theoretical runtime required to identify the optimal treatment offline. The last metric is crucial for efficient offline treatment planning in practical medical scenarios. During the test, we choose transformer as the main backbone for all the selected methods. For two stage methods like RMSNs and DeepBlip, the first stage uses a simpler LSTM instantiation instead for higher efficiency. The results are reported in Table 3.

Baselines like HA-TRM, RMSNs, CRN, CT estimate one conditional average potential outcome (CAPO) over time within a single inference. Therefore it takes $N(\tau)$ inferences to identify the treatment sequence with best outcome, where $N(\tau)$ denotes the number of possible treatment combinations within the time window $[t, t + \tau]$. G-Net computes the CAPO over time by performing Monte Carlo simulations, generating $M$ trajectories for each possible treatment sequence. IGC-Net only predicts the CAPO under a fixed intervention sequence, which means it requires additional $N(\tau) - 1$ re-trainings to calculate all the possible outcomes. Our DeepBlip, however, estimates CATE over time via predicting blip coefficients to identify the treatment combination with the optimal effect. By Eq. 4, only one forward pass to predict the blip coefficients suffices to identify the optimal treatment. Hence, although our DeepBlip is not the most efficient method during training or inference, **it is significantly faster in offline treatment planning**, which is critical for personalized medicine.

## I.3. Transformer instantiation for DeepBlip

Our DeepBlip uses the transformer architecture to encode history into a latent representation. The transformer is built upon the multi-input transformer from causal transformer (Melnychuk et al., 2022). The transformer is specially designed for medical application where the inputs include the outcomes, covariates and treatments. Each type of variable has a corresponding sub-transformer. The three sub-transformers perform not only the classic multi-headed self-attention within themselves, but also the cross-attention in-between. This ensures that information is shared across these sub-transformers.

Let the three sub-transformers be $z_\theta^k(\cdot), k = 1, 2, 3$. Suppose for $k = 1$, the input sequence is the $\boldsymbol{U}^k = (X_1, \ldots, X_t)$. Then the sub-transformer $z_\phi^k(\cdot)$ generate the representation in the following steps.

| Methods | Training ($t_T$) | Inference ($t_I$) | Offline optimal treatment identification |
|---|---|---|---|
| HA-TRM (Frauen et al., 2025) | $2.5 \pm 0.4$ | $0.16 \pm 0.04$ | $t_I \cdot N(\tau)$ |
| R-MSNs (Lim, 2018) | $7.2 \pm 0.9$ | $0.30 \pm 0.06$ | $t_I \cdot N(\tau)$ |
| CRN (Bica et al., 2020) | $5.4 \pm 0.6$ | $0.28 \pm 0.06$ | $t_I \cdot N(\tau)$ |
| CT (Melnychuk et al., 2022) | $3.4 \pm 0.5$ | $0.20 \pm 0.04$ | $t_I \cdot N(\tau)$ |
| G-Net (Li et al., 2021) | $2.8 \pm 0.4$ | $0.24 \pm 0.03$ | $t_I \cdot N(\tau) \cdot M$ [†] |
| IGC-Net (Hess et al., 2026a) | $2.6 \pm 0.4$ | $0.20 \pm 0.04$ | $(N(\tau) - 1) \cdot t_T \cdot n_e + t_I \cdot N(\tau)$ [††] |
| DeepBlip-WDO | $3.1 \pm 0.4$ | $0.17 \pm 0.06$ | $t_I$ |
| **DeepBlip (ours)** | $3.6 \pm 0.5$ | $0.17 \pm 0.06$ | $t_I$ |

†: G-Net performs $M$ simulations to infer the potential outcome.
††: IGC-Net requires re-training additionally $N(\tau) - 1$ times.

*Table 3.* Runtime report for the methods instantiated by transformers. $t_T$ (in min) is the training time on the training dataset per epoch. $t_I$ (in min) is the inference time on the test dataset. In the last column we report the time needed to identify the optimal treatment. $N(\tau)$ is the number of all possible combination of treatments during the future horizons $[t : t + \tau]$.

**Input transformation:** The raw input first goes through a linear transformation:

$$H_0^k = \text{Linear}_k(\boldsymbol{U}) \in \mathbb{R}^{t \cdot d_{\text{trm}}} \tag{73}$$

, where $H_0^k$ is the input of the first transformer blocks.

**Transformer blocks:** The multi-input sub-transformer has $B$ transformer blocks. Within each transformer block, the multi-headed self-attention and cross-attention are performed. Then the output goes through a feed forward network. Here are the details of a transformer block at layer $b$.

(1) Multi-headed self/cross-attentions First the keys, queries and values, namely $K, Q, V$ for $n_h$ attention heads, are computed from the output from last transformer block:

$$
\begin{aligned}
Q_b^{k,i} &= H_b^{k,i} W_Q^{k,i} + \mathbf{b}_Q^{k,i} \in \mathbb{R}^{t \cdot d_{qkv}}, \\
K_b^{k,i} &= H_b^{k,i} W_K^{k,i} + \mathbf{b}_W^{k,i} \in \mathbb{R}^{d_{t \cdot qkv}}, \\
V_b^{k,i} &= H_b^{k,i} W_V^{k,i} + \mathbf{b}_V^{k,i} \in \mathbb{R}^{d_{t \cdot qkv}},
\end{aligned}
\tag{74}
$$

where $i \in [1, n_h]$ is the index of a single attention head. Then the $i$-th attention is computed via a softmax over the scaled dot-product:

$$\text{Attn}_b^{k,i} = \text{Softmax}\left( \frac{Q_b^{k,i}(K_b^{k,i})^T}{\sqrt{d_{qkv}}} \right) V_b^{k,i} \tag{75}$$

The multi-head self-attention is then calculated by concatenating the attention heads:

$$\text{MHA}(Q_b^k, K_b^k, V_b^k) = \text{Concat}(\text{Attn}_b^{k,1}, \ldots, \text{Attn}_b^{k,n_h}) \in \mathbb{R}^{t \cdot d_{\text{trm}}} \tag{76}$$

Since $\text{MHA}(Q_b^k, K_b^k, V_b^k)$ has the same dimension as $H_b^k$, then we could generate a new set of queries for the cross-attentions with residual connection:

$$\widetilde{Q}_b^k = H_b^k + \text{MHA}(Q_b^k, K_b^k, V_b^k) \tag{77}$$

The cross-attentions are put on top of the self-attention layers and uses the queries and keys from the other two sub-transformers. For example, the multi-head cross-attention of sub-transformer $k$ with sub-transformer $m$ is:

$$\text{MHA}(\widetilde{Q}_b^k, K_b^m, V_b^m) = \text{Concat}(\text{CrossAttn}_b^{k,m,1}, \ldots, \text{CrossAttn}_b^{k,m,n_h}), \quad (m \neq k), \tag{78}$$

where the CrossAttn is defined as:

$$\text{CrossAttn}_b^{k,m,i} = \text{Softmax}\left( \frac{\widetilde{Q}_b^k(K_b^{m,i})^T}{\sqrt{d_{qkv}}} \right) V_b^{m,i}. \tag{79}$$

Finally, the output is achieved by adding self-attention's output and cross-attention's output together:

$$Z_b^k = \widetilde{Q}_b^k + \sum_{m \neq k} \text{MHA}(\widetilde{Q}_b^k, K_b^m, V_b^m) + S. \tag{80}$$

The static covariates $S$ are added when pooling different cross-attention outputs (see section 4.1 from Melnychuk et al. (2022)). The operations ensures that the information is shared across sub-transformers. Layer normalizations are added for all the self- and cross attentions (Vaswani et al., 2017).

(2) Feed-forward networks The output of the attention mechanism is processed through a position-wise feed-forward network (FFN) applied to each time step:

$$H_b^{k,\text{ff}} = \text{Linear}(\text{ReLU}(\text{Linear}(Z_b^k)), \tag{81}$$

where linear layers are followed by a dropout (Melnychuk et al., 2022). The output $H_b^{k,\text{ff}}$ serves as the input $(H_{b+1}^k)$ to the next transformer block. After processing through all $B$ blocks, the final representation for sub-transformer $k$ becomes $H_B^k$.

These final representations are then aggregated via an element-wise average pooling followed by a linear transformation and a exponential linear unit (ELU):

$$H_B = \text{ELU}(\text{Linear}(\text{Pool}(H_B^1, H_B^2, H_B^3))) \in \mathbb{R}^{t \cdot d_{\text{hr}}}. \tag{82}$$

Hence, the transformer-based sequential encoder $\mathcal{E}_\theta$ outputs the latent vector: $\mathcal{E}_\theta(H_t) = \mathbf{hr}_t = H_{B,t} \in \mathbb{R}^{d_{\text{trm}}}$ at time step $t$. We omit the positional encoding for better clarity. For more relevant details see section 4.2 from Melnychuk et al. (2022).

Our DeepBlip has two stages: (1) Nuisance network, and (2) Blip prediction network. Each stage uses a separate sequential encoder: $\mathcal{E}_{\theta_N}^N$ for (1) and $\mathcal{E}_{\theta_B}^B$ for (2). To accelerate the training, we only instantiate $\mathcal{E}_{\theta_B}^B$ by the transformer, and use LSTM to instantiate (1). The underlying reason is that the demand for accuracy in Stage (1) is not as high as Stage (2), as the $L^2$-moment loss is Neyman orthogonal over the nuisance functions.

**Nuisance Network:** At each time step $t$, the nuisance network estimates the nuisance functions:

$$p_{t,k}(h_{t+k}) := \mathbb{E}\big[Y_{t+\tau} \mid H_{t+k} = h_{t+k}\big], \quad 1 \le t \le T - \tau, 0 \le k \le \tau \tag{83}$$

$$q_{t,j,k}(h_{t+k}) := \mathbb{E}\big[Q_{t,j} \mid H_{t+k} = h_{t+k}\big], 1 \le t \le T - \tau, 0 \le k \le j \le \tau \tag{84}$$

Therefore a **multi-head output layer** is added on top of the encoder $\mathcal{E}_\theta^N$. The heads receive the representation $\text{hr}_t^N$ from the encoder and transform the input with multi-layer perceptron networks:

1. $\hat{p}_{t,k}(h_{t+k}) = \text{MLP}_p^{(k)}(\mathbf{hr}_{t+k}^N) \approx p_{t,k}(h_{t+k}), \quad k = 0, 1, .., \tau$

2. $\hat{q}_{t,j,k}(h_{t+k}) = \text{MLP}_q^{(j,k)}(\mathbf{hr}_{t+k}^N) \approx q_{t,j,k}(h_{t+k}), \quad 0 \le k \le j \le \tau$

where $\text{MLP}_p^{(k)}$ and $\text{MLP}_q^{(j,k)}$ are fully connected networks with ReLU activations. Each MLP has input size of $d_{\text{hr}}$, a single hidden layer with $d_{\text{hidden}}$ perceptrons and a single output transformation that maps the hidden layer to a 1-dimensional output. For binary output in some $q_{t,j,k}$, an additional sigmoid activation is added to contract the range into $[0, 1]$.

**Blip prediction network's output** Likewise a multi-head output layer is added on top of the sequential encoder $\mathcal{E}_{\theta_B}^B$ to predict the blip coefficients. For each horizon $k \in \{0, 1, \ldots, \tau\}$:

$$\widehat{\psi}_k(h_t) = \text{MLP}_{\theta_B}^{(k)}(\mathbf{hr}_t^B) \approx \psi_{t,k}(h_t)$$

where $\text{MLP}_{\theta_B}^{(k)}$ maps $\mathbf{hr}_t^B$ to blip coefficients. $\text{MLP}_{\theta_B}^{(k)}$ has the same structure as the MLPs in stage (1).

### I.4. LSTM instantiation for DeepBlip

DeepBlip-LSTM uses LSTM as the sequential encoders in two stages: 1. Stage (1) (Nuisance Network): Estimates residuals for blip function estimation. 2. Stage (2) (Blip Prediction Network): Predicts blip parameters $\psi_{t,k}(h_t)$.

**Sequential Encoding via LSTM** Each stage shares an individual LSTM to encode patient history. Let $\mathbf{hz}_t$ be the hidden state at time $t$. At each time step $t$, the input vector $\text{v}_t \in \mathbb{R}^{d_{\text{input}}}$ concatenates:

1. Static features $X_s \in \mathbb{R}^{d_{\text{static}}}$ (remain constant across time)

2. Current time-varying covariates $X_d \in \mathbb{R}^{d_{\text{dynamic}}}$ ($d_{\text{dynamic}} + d_{\text{static}} = d_x$)

3. Previous treatments $a_{t-1} \in \mathbb{R}^{d_a}$

4. Previous outcomes $Y_{t-1} \in \mathbb{R}$

to form the input vector:

$$V_t = \text{Concat}\big(X_s, X_d, A_{t-1}, Y_{t-1}\big) \in \mathbb{R}^{d_{\text{input}}}, \quad d_{\text{input}} = d_x + d_a + 1.$$

Then the LSTM processes $\{V_t\}_{t=1}^T$ into hidden state $\text{hs}_t$ and cell state $c_t$:

$$(\text{hs}_t, c_t) = \text{LSTM}\big(v_t, (\text{hs}_{t-1}, c_{t-1})\big),$$

where $\mathbf{h}_t \in \mathbb{R}^{d_{\text{hidden}}}$. Next we take the hidden state and derive the temporal representation of the patient state at time $t$:

$$\mathbf{hr}_t = \text{ELU}\big(\mathbf{W}_{\text{hr}} \cdot \text{dropout}(\text{hs}_t) + b_{\text{hr}}\big),$$

where $\mathbf{W}_{\text{hr}} \in \mathbb{R}^{d_{\text{hr}} \times d_{\text{hidden}}}$ is a learnable projection matrix.

Above is the whole process of generating the temporal representation at time $t$. Since the DeepBlip has two stages, therefore we need to train **two separate** sequential encoders: $\mathcal{E}_{\theta_N}^N(h_t) = \mathbf{hr}_t^N$ for the nuisance network and $\mathcal{E}_{\theta_B}^B(h_t) = \mathbf{hr}_t^B$ for the blip prediction network. The output layer of both stages remain the same as the DeepBlip-TRM.

**I.5. Hyperparameter tuning for tumor growth dataset**

**I.5. Hyperparameter tuning for tumor growth dataset**

**I.6. Hyperparameter tuning for MIMIC-III dataset**

*Table 4.* Hyperparameter Tuning for Methods on Tumor Growth Data: We perform a random grid search with 20 iterations and choose the best hyperparameters for each task. $C = d_{\text{input}}$ is the dimension of the input. For the causal transformer, the CDC coefficient is $\alpha = 0.01$ (Melnychuk et al., 2022). The number of MC samples of G-Net is 50 (Li et al., 2021). Models are either instantiated by LSTMs or transformers. Here we display the search ranges of the *original* instantiations for each method. In the experiments, we adopt a universal instantiation type for all the baselines for fairness. Hence, the performance comparison in experiment section is fair.

| Method | Component | Hyperparameter | Tuning Range |
|---|---|---|---|
| HA-TRM (Frauen et al., 2025) | (end-to-end) | Transformer blocks ($B$)
Learning rate ($\eta$)
Batch size
Attention heads ($n_h$)
Transformer units ($d_{\text{trm}}$)
hidden representation ($d_{\text{hr}}$)
FC hidden units ($d_{\text{fc}}$)
Number of epochs ($n_e$) | [1, 2]
[0.01, 0.001, 0.0001]
[64, 128]
2
[0.5C, 1C]
[0.5C, 1C]
[0.5$d_{\text{hr}}$, 1$d_{\text{hr}}$]
10 |
| CRN (Bica et al., 2020) | Encoder | LSTM layers ($l$)
Learning rate ($\eta$)
LSTM hidden units ($d_h$)
LSTM dropout rate ($p$)
Balanced representation size ($d_r$)
FC hidden units ($d_{\text{fc}}$)
Number of epochs ($n_e$) | [1, 2, 3]
[0.1, 0.01, 0.001]
[32, 64, 128, 256]
[0.1, 0.2, 0.3]
[0.5$d_h$, 1$d_h$, 2$d_h$]
[0.5$d_h$, 1$d_h$, 2$d_h$]
10 |
| | Decoder | LSTM layers ($l$)
Learning rate ($\eta$)
LSTM hidden units ($d_h$)
Balanced representation size ($d_r$)
FC hidden units ($d_{\text{fc}}$)
LSTM dropout rate ($p$)
Number of epochs ($n_e$) | [1, 2, 3]
[0.1, 0.01, 0.001]
[256, 512, 1024]
[0.5$d_h$, 1$d_h$, 2$d_h$]
[0.5$d_h$, 1$d_h$, 2$d_h$]
[0.1, 0.2, 0.3]
20 |
| CT (Melnychuk et al., 2022) | (end-to-end) | Transformer blocks ($B$)
Learning rate ($\eta$)
Batch size
Attention heads ($n_h$)
Transformer units ($d_{\text{trm}}$)
hidden representation ($d_{\text{hr}}$)
FC hidden units ($d_{\text{fc}}$)
Number of epochs ($n_e$) | [1, 2]
[0.01, 0.001, 0.0001]
[64, 128]
2
[0.5C, 1C]
[0.5C, 1C]
[0.5$d_{\text{hr}}$, 1$d_{\text{hr}}$]
20 |
| R-MSNs (Lim, 2018) | Treatment/History Propensity Network | LSTM layers ($l$)
Learning rate ($\eta$)
LSTM hidden units ($d_h$)
FC hidden units ($d_{\text{fc}}$)
LSTM dropout rate ($p$)
Max gradient norm
Number of epochs ($n_e$) | 1
[0.1, 0.01, 0.001]
[64, 128, 256]
[0.5$d_{\text{hr}}$, 1$d_{\text{hr}}$]
[0.1, 0.2]
[0.5, 1.0, 2.0]
10 |
| | Encoder / Decoder | LSTM layers ($l$)
Learning rate ($\eta$)
LSTM hidden units ($d_h$)
FC hidden units ($d_{\text{fc}}$)
LSTM dropout rate ($p$)
Max gradient norm
Number of epochs ($n_e$) | 1
[0.01, 0.001, 0.0001]
[256, 512, 1024]
[0.5$d_{\text{hr}}$, 1$d_{\text{hr}}$]
[0.1, 0.2, 0.4]
[0.5, 1.0, 2.0]
20 |
| G-Net (Li et al., 2021) | (end-to-end) | LSTM layers ($l$)
Learning rate ($\eta$)
LSTM hidden units ($d_h$)
LSTM output size ($d_o$)
Feed-forward hidden units ($d_{\text{ff}}$)
LSTM dropout rate ($p$)
Number of epochs ($n_e$) | [1, 2, 3]
[0.1, 0.01, 0.001]
[64, 128, 256]
[0.5$d_h$, 1$d_h$, 2$d_h$]
[0.5$d_h$, 1$d_h$, 2$d_h$]
[0.1, 0.2]
20 |
| IGC-Net (Hess et al., 2026a) | (end-to-end) | Transformer blocks ($B$)
Learning rate ($\eta$)
Batch size
Attention heads ($n_h$)
Transformer units ($d_{\text{trm}}$)
hidden representation ($d_{\text{hr}}$)
FC hidden units ($d_{\text{fc}}$)
Number of epochs ($n_e$) | [1, 2]
[0.01, 0.001, 0.0001]
[64, 128]
2
[0.5C, 1C]
[0.5C, 1C]
[0.5$d_{\text{hr}}$, 1$d_{\text{hr}}$]
10 |
| DeepBlip (ours) | Nuisance Network | LSTM layers ($l$)
Learning rate ($\eta$)
LSTM hidden units ($d_h$)
LSTM dropout rate ($p$)
hidden representation size ($d_r$)
FC hidden units ($d_{\text{fc}}$)
Number of epochs ($n_e$) | [1, 2, 3]
[0.1, 0.01, 0.001]
[32, 64, 128, 256]
[0.1, 0.2]
[0.5$d_h$, 1$d_h$, 2$d_h$]
[0.5$d_h$, 1$d_h$, 2$d_h$]
10 |
| | Blip Prediction Network | Transformer blocks ($B$)
Learning rate ($\eta$)
Batch size
Attention heads ($n_h$)
Transformer units ($d_{\text{trm}}$)
hidden representation ($d_{\text{hr}}$)
FC hidden units ($d_{\text{fc}}$)
Number of epochs ($n_e$) | [1, 2]
[0.01, 0.001, 0.0001]
[64, 128]
2
[0.5C, 1C]
[0.5C, 1C]
[0.5$d_{\text{hr}}$, 1$d_{\text{hr}}$]
20 |

*Table 5.* Hyperparameter Tuning for Methods on MIMIC-III semi-synthetic Data: We perform a random grid search with 20 iterations and choose the best hyperparameters for each task. $C = d_{\text{input}}$ is the dimension of the input. For the causal transformer, the CDC coefficient is $\alpha = 0.01$ (Melnychuk et al., 2022). The number of MC samples of G-Net is 50 (Li et al., 2021). Models are either instantiated by LSTMs or transformers. Here we display the search ranges of the *original* instantiations for each method. In the experiments, we adopt a universal instantiation type for all the baselines for fairness. Hence, the performance comparison in experiment section is fair.

| Method | Component | Hyperparameter | Tuning Range |
|---|---|---|---|
| HA-TRM (Frauen et al., 2025) | (end-to-end) | Transformer blocks ($B$)
Learning rate ($\eta$)
Batch size
Attention heads ($n_h$)
Transformer units ($d_{\text{trm}}$)
hidden representation ($d_{\text{hr}}$)
FC hidden units ($d_{\text{fc}}$)
Number of epochs ($n_e$) | [1, 2]
[0.01, 0.001, 0.0001]
[64, 128]
2
[0.5C, 1C]
[0.5C, 1C]
[0.5$d_{\text{hr}}$, 1$d_{\text{hr}}$]
10 |
| CRN (Bica et al., 2020) | Encoder | LSTM layers ($l$)
Learning rate ($\eta$)
LSTM hidden units ($d_h$)
LSTM dropout rate ($p$)
Balanced representation size ($d_r$)
FC hidden units ($d_{\text{fc}}$)
Number of epochs ($n_e$) | [1, 2, 3]
[0.1, 0.01, 0.001]
[32, 64, 128, 256]
[0.1, 0.2, 0.3]
[0.5$d_h$, 1$d_h$, 2$d_h$]
[0.5$d_h$, 1$d_h$, 2$d_h$]
10 |
| | Decoder | LSTM layers ($l$)
Learning rate ($\eta$)
LSTM hidden units ($d_h$)
Balanced representation size ($d_r$)
FC hidden units ($d_{\text{fc}}$)
LSTM dropout rate ($p$)
Number of epochs ($n_e$) | [1, 2, 3]
[0.1, 0.01, 0.001]
[256, 512, 1024]
[0.5$d_h$, 1$d_h$, 2$d_h$]
[0.5$d_h$, 1$d_h$, 2$d_h$]
[0.1, 0.2, 0.3]
20 |
| CT (Melnychuk et al., 2022) | (end-to-end) | Transformer blocks ($B$)
Learning rate ($\eta$)
Batch size
Attention heads ($n_h$)
Transformer units ($d_{\text{trm}}$)
hidden representation ($d_{\text{hr}}$)
FC hidden units ($d_{\text{fc}}$)
Number of epochs ($n_e$) | [1, 2]
[0.01, 0.001, 0.0001]
[64, 128]
2
[0.5C, 1C]
[0.5C, 1C]
[0.5$d_{\text{hr}}$, 1$d_{\text{hr}}$]
20 |
| R-MSNs (Lim, 2018) | Treatment/History Propensity Network | LSTM layers ($l$)
Learning rate ($\eta$)
LSTM hidden units ($d_h$)
FC hidden units ($d_{\text{fc}}$)
LSTM dropout rate ($p$)
Max gradient norm
Number of epochs ($n_e$) | 1
[0.1, 0.01, 0.001]
[64, 128, 256]
[0.5$d_{\text{hr}}$, 1$d_{\text{hr}}$]
[0.1, 0.2]
[0.5, 1.0, 2.0]
10 |
| | Encoder / Decoder | LSTM layers ($l$)
Learning rate ($\eta$)
LSTM hidden units ($d_h$)
FC hidden units ($d_{\text{fc}}$)
LSTM dropout rate ($p$)
Max gradient norm
Number of epochs ($n_e$) | 1
[0.01, 0.001, 0.0001]
[256, 512, 1024]
[0.5$d_{\text{hr}}$, 1$d_{\text{hr}}$]
[0.1, 0.2, 0.4]
[0.5, 1.0, 2.0]
20 |
| G-Net (Li et al., 2021) | (end-to-end) | LSTM layers ($l$)
Learning rate ($\eta$)
LSTM hidden units ($d_h$)
LSTM output size ($d_o$)
Feed-forward hidden units ($d_{\text{ff}}$)
LSTM dropout rate ($p$)
Number of epochs ($n_e$) | [1, 2, 3]
[0.1, 0.01, 0.001]
[64, 128, 256]
[0.5$d_h$, 1$d_h$, 2$d_h$]
[0.5$d_h$, 1$d_h$, 2$d_h$]
[0.1, 0.2]
20 |
| IGC-Net (Hess et al., 2026a) | (end-to-end) | Transformer blocks ($B$)
Learning rate ($\eta$)
Batch size
Attention heads ($n_h$)
Transformer units ($d_{\text{trm}}$)
hidden representation ($d_{\text{hr}}$)
FC hidden units ($d_{\text{fc}}$)
Number of epochs ($n_e$) | [1, 2]
[0.01, 0.001, 0.0001]
[64, 128]
2
[0.5C, 1C]
[0.5C, 1C]
[0.5$d_{\text{hr}}$, 1$d_{\text{hr}}$]
10 |
| DeepBlip (ours) | Nuisance Network | LSTM layers ($l$)
Learning rate ($\eta$)
LSTM hidden units ($d_h$)
LSTM dropout rate ($p$)
hidden representation size ($d_r$)
FC hidden units ($d_{\text{fc}}$)
Number of epochs ($n_e$) | [1, 2, 3]
[0.1, 0.01, 0.001]
[32, 64, 128, 256]
[0.1, 0.2]
[0.5$d_h$, 1$d_h$, 2$d_h$]
[0.5$d_h$, 1$d_h$, 2$d_h$]
10 |
| | Blip Prediction Network | Transformer blocks ($B$)
Learning rate ($\eta$)
Batch size
Attention heads ($n_h$)
Transformer units ($d_{\text{trm}}$)
hidden representation ($d_{\text{hr}}$)
FC hidden units ($d_{\text{fc}}$)
Number of epochs ($n_e$) | [1, 2]
[0.01, 0.001, 0.0001]
[64, 128]
2
[0.5C, 1C]
[0.5C, 1C]
[0.5$d_{\text{hr}}$, 1$d_{\text{hr}}$]
20 |

