# OpenReview forum: "DeepBlip: Estimating Conditional Average Treatment Effects Over Time"
_ICML.cc/2026/Conference — ICML 2026 regular_

### Official Review · Reviewer_XYQ2 · 2026-02-23

**Soundness:** 4
**Presentation:** 4
**Significance:** 3
**Originality:** 3
**Overall Recommendation:** 5
**Confidence:** 1

**Summary:**

The paper proposes a novel nerual network model for estimating Blip function, which is used to estimate CATE over the time. The empirical valuation shows that the tailored net structure and loss function benefits the estimation.

**Compliance With Llm Reviewing Policy:**

Affirmed.

**Final Justification:**

I acknowledged the rebuttal and remain in the same score

**Key Questions For Authors:**

There is no more question from me.

**Limitations:**

Yes

**Strengths And Weaknesses:**

**Soundness**: The empirical estimation successfully supports the proposed loss function and network strucure.

**Presentation**: The paper is well-presented

**Significance**: The paper tackles the important questions.

**Originality**: Although the components of network structures are imported from existing research (e.g. LSTM, Transformers) the use of them in the scenario is new.

---

> ### Author Rebuttal · Authors · 2026-03-30
>
> We thank the reviewer for the positive assessment and for recognizing the contribution of our work.
>
> We further clarify: our contribution is **not** simply applying existing neural networks to a sequential setting. Rather, the key advance is a new **neural SNMM framework** for CATE estimation over time, which includes (i) a **double-optimization trick** that overcomes the sequential dependence of standard g-estimation and enables simultaneous gradient-based learning of all blip functions, (ii) a **Neyman-orthogonal L2-moment objective** for robustness to nuisance estimation error, and (iii) **efficient offline evaluation** through predicted blip coefficients. These are precisely the ingredients that distinguish DeepBlip from prior neural approaches based on G-computation/MSMs as well as from Dynamic DML.
>
> We will further revise the paper to make this contribution more explicit. In case you have any further questions, we would be happy to explain and clarify.

---

> > ### Author Rebuttal · Reviewer_XYQ2 · 2026-04-02
> >
> > There were no more concerns from me.

---

### Official Review · Reviewer_3kz8 · 2026-03-09

**Soundness:** 3
**Presentation:** 3
**Significance:** 3
**Originality:** 3
**Overall Recommendation:** 4
**Confidence:** 4

**Summary:**

This paper presents a new method for estimating conditional average treatment effects over time when time-varying confounding is present. The authors propose DeepBlip first neural network framework for learning treatment effects over time using SNMMs. Particularly, the authors circumvent a problem with implementing sequential g-estimation with neural networks with their novel double optimization trick that allows for end-to-end gradient training and simultaneous learning of all blip prediction networks.

The method uses a two-stage architecture - the nuisance network (stage 1) estimates conditional expectations of outcomes and treatments given patient history to compute residualized variables necessary for moment conditions. The blip prediction network (stage 2) learns neural network predictors of blip coefficients. They provide experimental validation on simulated tumor growth data and the MIMIC-III clinical benchmark, showing their method is more robust to time-varying confounding and has improved performance and scalability compared to baselines.

**Compliance With Llm Reviewing Policy:**

Affirmed.

**Final Justification:**

The rebuttal addressed my comments - I believe my prior assessment still holds.

**Key Questions For Authors:**

1) How sensitive is DeepBlip to misspecification of the nuisance models?
2) Can the authors provide experiments to show the footnote claim that the linearity assumption on the treatment effect can be relaxed?
3) Are there fully non-synthetic data examples this could be evaluated on?

Addressing these questions as well as the weaknesses listed above could strengthen the paper and my evaluation.

**Limitations:**

yes

**Strengths And Weaknesses:**

The strengths of this paper include its well-grounded causal framework, novelty as the first neural implementation of SNMMs, well-motivated framework with the double optimization trick, sound theoretical justification, and strong performance in experiments compared to included baselines. Additionally, the problem addressed has significance - handling time-varying treatment effects with more flexibility is crucial for applications such as precision medicine and sequential decision making.

The authors rely upon results in an extensive appendix to justify claims made in the main body of the paper without always stating the most important takeaways or intuition in the main text. While the appendix should be commended for the coverage of different aspects of evaluating DeepBlip, the readability of the main text would be improved if it were not assumed or required that each reader has to thoroughly examine the appendix for full understanding of the main text.
A second weakness is that the authors mention that the nonlinear extension to the linear treatment effect assumption made in the paper is not experimentally validated.

---

> ### Author Rebuttal · Authors · 2026-03-30
>
> We sincerely thank the reviewer for recognizing the contribution and novelty of our work. We are pleased to address your concerns below.
>
> ## W1: Main text readability relies on appendix
>
> We appreciate your feedback on readability. We agree that the main text should be more self-contained. Specifically:
>
> \- **Double optimization trick (Sec. 4.2):** We will add the intuitive gradient-based explanation of why the trick works: Currently, this argument is only implicit in Appendix D.
> \- **Neyman-orthogonality (Remark 4.1):** We will add a paragraph explaining why Neyman-orthogonality provides crucial benefits on the robustness of the second-stage learner.
> \- **MSE rate (Remark 4.2):** We will add the key intuition that under standard regularity conditions, the blip coefficient estimation error vanishes at a rate $O(\\log\\log(n)/n)$, which is near-parametric.
>
> **Action:** We will revise the main text to include these explanations so that the readers need not to refer to the appendix.
>
> ## W2: The nonlinear treatment effect extension is not experimentally validated
>
>
> We apologize for the lack of clarity. In fact, we already use the nonlinear extension in our tumor growth experiments. The radiotherapy effect in the tumor growth DGP (Eq. (61), Appendix F.1) includes a quadratic term $\\alpha\_r d\_{t+1} \+ \\beta\_r d\_{t+1}^2$. To capture this, our implementation uses the **nonlinear** feature map $\\phi(a\_{t+k}) \= (c\_{t+k},\\, d\_{t+k},\\, d\_{t+k}^2)$, then the blip function becomes $\\gamma\_{t,k}(h\_t) \= \\psi\_{t,k}(h\_t)^\\prime \\phi(a\_{t+k})$, exactly as described in Footnote 5\. Therefore, our numerical experiments already validate our method in a nonlinear setting..
>
> **Action:** We will explicitly state in the main text (Sec. 4.1) that the tumor growth experiments use $\\phi(a) \= (c, d, d^2)$ to show that the nonlinear extension from Footnote 5 is already validated. We will also add respective details in the appendix of implementation.
>
> ## Q1: Sensitivity of our framework
>
> Our framework provides  **Neyman-orthogonality (Remark 4.1, Appendix C.2):** This provides theoretical robustness against nuisance misspecification: even if Stage 1 estimates are moderately inaccurate, Stage 2 estimates remain approximately correct (Chernozhukov et.al. 2018).
>
> We further conduct experiments to assess the sensitivity of DeepBlip to misspecification in the nuisance functions $\\hat p\_{t,k}$ (mean outcome model) and $\\hat q\_{t,j,k}$ (treatment propensity model). To isolate the effects of $\\hat p\_{t,k}$ and $\\hat q\_{t,j,k}$, we perturb them \\textbf{separately}; the corresponding results are shown in Figure 5-6 from [Linked PDF](https://anonymous.4open.science/r/ICML2026-DeepBlip-rebuttal-pdf/DeepBlip_rebuttal.pdf).
>
> For both nuisance components, the RMSE increases \\emph{slowly} as the perturbation magnitude grows, without abrupt changes. In particular, when the noise level is low to moderate ($\\sigma \\leq 0.02$ or $0.06$), the performance degradation remains limited. This indicates that DeepBlip, with the Neyman orthogonality property, is not sensitive to mild nuisance misspecification.
>
> **Action:** We will add a dedicated sensitivity analysis in the revised supplement where we gradually degrade the nuisance network by adding artificial random noise and check its effect on the second stage’s RMSE. This will directly quantify the robustness provided by Neyman-orthogonality in practice.
>
> ## Q2: Empirical evidence on relaxation of linear assumption
>
> Please see our clarification in W2.
>
> ## Q3: Validation on real-world dataset
>
> We are happy to clarify the absence of the real-world experiment. The fundamental challenge with fully real-world evaluation of CATE methods is that **ground-truth treatment effects are never observed**. We only observe the outcome under the treatment that was actually administered, but not the counterfactual. For this reason, the entire literature on CATE estimation over time (Bica et al., 2020; Melnychuk et al., 2022) relies on synthetic or semi-synthetic benchmarks to rigorously assess the performance.
>
> Nevertheless, we showcase the effectiveness of our DeepBlip by analyzing the treatment effect of vasopressors on blood pressure in the MIMIC-III, following the setup of Melny et. al. 2022\. Then we visualize the average blip coefficients as well as the cumulative CATE over time. The figures are included in the [Linked PDF](https://anonymous.4open.science/r/ICML2026-DeepBlip-rebuttal-pdf/DeepBlip_rebuttal.pdf) (Figure 7,8).  The estimated results align with known clinical knowledge.
>
> **Action:** We will add a clinical consistency check on the MIMIC-III data as our real-world experiment, and discuss the inherent limitations of real-world CATE evaluation more explicitly in the revised paper.

---

> > ### Author Rebuttal · Reviewer_3kz8 · 2026-03-31
> >
> > The points I brought up have been well justified and the proposed actions for revision seem effective

---

### Official Review · Reviewer_V1Rr · 2026-03-13

**Soundness:** 3
**Presentation:** 3
**Significance:** 3
**Originality:** 3
**Overall Recommendation:** 5
**Confidence:** 3

**Summary:**

This paper presents DeepBlip, a framework for learning conditional average treatment effects (CATEs) via a neural-based structural nested mean model (SNMM). The method is unbiased and has an MSE guarantee. The authors demonstrate the utility of DeepBlip on two medical datasets.

**Compliance With Llm Reviewing Policy:**

Affirmed.

**Final Justification:**

The proposed method is novel, has useful applications, and appears to have strong performance across a range of settings. The rebuttal maintained my positive assessment.

**Key Questions For Authors:**

See minor issues raised in Strengths and Weaknesses.

While the quantitative improvements are substantial (especially at larger values of $\tau$), would you point to any qualitative improvements relative to baselines?

**Limitations:**

Yes

**Strengths And Weaknesses:**

- _Soundness:_ The complement of theoretical and empirical evidence is strong. I have not independently verified the correctness of the technical claim in Remark 4.2, but it is a strong result if sound. The empirical evidence is also fairly strong, with significant advantages for the proposed DeepBlip method over state-of-the-art methods at large values of $\tau$. More explanation on the mediocre performance at low values of $\tau$ (relative to the improved performance at larger values of $\tau$) would be helpful.
- _Presentation:_ The submission is well-structured and presented, with a few areas for improvement. Figure 2 has several typos ("sequential"), and Figures 3 and 4 should include confidence intervals.
- _Significance:_ The submission deals with a well-known problem area (CATE) and provides what seems to be a useful tool. The authors overcame several challenges (a two-stage neural architecture, and a double optimization trick) that make the solution non-obvious. It is a bit disappointing that one of the main motivations of the work (and SNMMs more broadly) is that the architecture "promotes interpretability through the incremental effects and enables the efficient offline evaluation of optimal treatment policies without re-computation", but the submission stops short of showing those abilities. As a result, the submission may be more interesting to an audience that is already familiar with SNMMs rather than a broader audience.
- _Originality:_ The authors introduce a new method with non-obvious architectural requirements, perform theoretical analysis, and demonstrate empirical success.

---

> ### Author Rebuttal · Authors · 2026-03-30
>
> We sincerely thank the reviewer for the positive assessment and for recognizing the non-trivial challenges our neural SNMM framework addresses. We will address your concerns below.
>
> ## W1: Performance at low prediction horizon $\\tau$
>
> We are happy to clarify the mediocre performance at shorter horizons $\\tau$. When $\\tau$ is small (e.g. $tau \= 3), all methods that properly adjust for confounding perform (see Table 2). With fewer time steps, the problems from inverse propensity weighting and nested expectation (G-computation) is not severe. In this circumstance, all the methods that adjusts for time-varying confounding (e.g. GT, G-Net, R-MSN) have sufficient accuracy and DeepBlip only offers a limited benefit. It still outperforms the baselines, but the relative advantage is smaller given the problem is easier for everyone.
>
> The advantage of our method only emerges at longer horizons. Our DeepBlip decomposes the total effect into $\\tau \+ 1$ localized blip effects, each estimated from its own loss function $\\mathcal{L}\_{k}$. The decomposition prevents error accumulation, which is prevalent in MSM-based models and G-computation-based models.
>
> **Action:** We can add this discussion to the experiment section if you consider it important to be clarified.
>
> ## W2: Typos in Figure 2 and confidence interval
>
> Thank you for catching the error. We will correct this  to "Sequential." We already report the confidence interval for semi-synthetic experiments (see Table 2). We update the training time comparison. We further update the results of the tumor growth experiment as well as the ablation studies to include confidence intervals. The new figures are included in a pdf file: see Figure 1,2,3,4 from [Linked PDF](https://anonymous.4open.science/r/ICML2026-DeepBlip-rebuttal-pdf/DeepBlip_rebuttal.pdf)).
>
> **Action**: Both issues will be fixed in the revised manuscript.
>
> ## W3: Lack of demonstration on interpretability and offline evaluation
>
> Thanks for the important feedback. We are happy to strengthen the paper's appeal to a broader audience. We would like to clarify the following:
>
> **Regarding offline evaluation efficiency:** We do provide a quantitative comparison of offline evaluation cost in Table 3 (Appendix I.2). DeepBlip identifies the optimal treatment sequence in a single forward pass ($t\_I$​), whereas other baselines either require $N(\\tau)$ forward passes ($N(\\tau)= 2^{\\tau+1}$ when treatment is binary) or even re-trainings (GT). This is a **qualitative** difference in computational complexity.
>
> **Regarding interpretability**: The granular blip coefficient analysis in Appendix E.2 (Figures 9–10) demonstrates interpretability: clinicians can inspect each $\\hat{\\psi}\_{t,k}$ to understand the effect of treatment at step $t+k$ on the final outcome at $t+\\tau$. For example, in the tumor growth setting, one can read off how the effect of chemotherapy at step $k$ differs from that of radiotherapy, and if $\\hat{\\psi}\_{t,k} \> 0$, then chemotherapy is the better treatment option. This is not possible with other methods like the Causal Transformer. We further conduct experiments on real-world dataset to use our DeepBlip to analyze the effect of vasopressor on blood pressure (see Figure 7,8 from the \[[Linked PDF](https://anonymous.4open.science/r/ICML2026-DeepBlip-rebuttal-pdf/DeepBlip_rebuttal.pdf)), which demonstrates the high interpretability of our DeepBlip.
>
> **Action:** In the revised text, we will move the runtime comparison (Table 3\) and a summary of the interpretability analysis into the main text.
>
> ## Q1: Qualitative improvements relative to baselines
>
> Thanks for asking this question. Yes, beyond the quantitative RMSE gains, DeepBlip offers several qualitative advantages that baselines lack:
>
> 1\. **Temporal stability.** Baselines exhibit rapidly growing RMSE and variance as $\\tau$ increases (Table 2). For example, R-MSNs: std. dev. grows from 0.17 ($\\tau=3$) to 1.50 ($\\tau=10$), and G-Net:s RMSE from 0.58 to 3.12. DeepBlip: RMSE growth is substantially slower (0.50 to 0.98), and its variance remains stable (0.12 to 0.32). This stability is a qualitative property — it means DeepBlip's predictions remain significantly robust compared to other baselines.
>
> 2\. **Efficient inference/policy search.** Please see the discussion in W3.
>
> 3\. **Interpretability.**  Please see the discussion in W3.

---

> > ### Author Rebuttal · Reviewer_V1Rr · 2026-04-04
> >
> > Thank you for the response. I maintain my positive score.

---

### Official Review · Reviewer_JDPn · 2026-03-16

**Soundness:** 3
**Presentation:** 4
**Significance:** 3
**Originality:** 2
**Overall Recommendation:** 4
**Confidence:** 4

**Summary:**

This paper proposes a deep neural framework for temporal causal inference based on Structural Nested Mean Models (SNMMs). The proposed method, named DeepBlip, parameterizes the blip functions using LSTM or transformers to capture the time-dependent local effect of each treatment at each time, avoiding directly modeling the full counterfactual trajectory over time. The proposed framework is reasonable, and the experimental results also show it effectiveness.

**Compliance With Llm Reviewing Policy:**

Affirmed.

**Final Justification:**

My concerns are well addressed. I am happy to increase my score, conditional on the changes being incorporated into the revised version.

**Key Questions For Authors:**

see above

**Limitations:**

see above

**Strengths And Weaknesses:**

### Strengths

1. This paper is well written and easy to follow.

2. The proposed double optimization trick aims to improve runtime efficiency and shows improved performance.

3. The experiments are comprehensive and show the effectiveness of the proposed framework.


### Weaknesses / Questions

1. The novelty may be somewhat incremental. LSTMs and transformers have been widely used in estimating CATE over time, and the main contribution is implementing SNMMs with such architectures.
1. It could be better to provide a runtime comparison between DeepBlip and DeepBlip-WDO since the double optimization trick is designed to improve computational efficiency.
1. It is unclear why the double optimization trick leads to improved performance as the ablation study shows. Additional explanation or analysis would help clarify this effect.
1. It is unclear whether the framework requires cross-fitting, as this only appears in Figure 2. Typically the Neyman-orthogonality requires that.
1. In the experimental results, GT achieve the second-best performance, better and more stable than Dynamic DML that also uses SNMM-based decomposition. Could the authors clarify this point?
1. Regarding the datasets, the data generation process seems different from the standard setups used in [1,2]. How does the proposed method perform compared with baselines in the original datasets?

I am happy to increase my score if the authors can address the concerns well.


[1] ESTIMATING COUNTERFACTUAL TREATMENT OUTCOMES OVER TIME THROUGH ADVERSARIALLY BALANCED REPRESENTATIONS. ICLR 2020

[2] IGC-NET FOR CONDITIONAL AVERAGE POTENTIAL OUTCOME ESTIMATION OVER TIME. ICLR 2026.

---

> ### Author Rebuttal · Authors · 2026-03-30
>
> We thank the reviewer for the constructive feedback and for acknowledging the effectiveness of the double optimization trick and the experiments. We will address your concern below.
>
> ## W1: Novelty maybe incremental
>
> We respectfully disagree that the contribution is incremental. While LSTMs and transformers have been used for CATE estimation, the core of our work is **not** simply plugging in these architectures from existing frameworks. Our novelty lies in making SNMMs compatible with end-to-end neural training, which is nontrivial because classic g-estimation must be done in a sequential way. To address this, we make the following contributions:
>
>
> - **The double optimization trick**  (Sec. 4.2): It breaks the sequential dependence and enables simultaneous training of the blip predictors. We provide theoretical justification for convergence and prove that the resulting loss retains Neyman-orthogonality (Remark 4.1 \+ Appendix C.2).
> - **The $L^2$-moment loss formulation**
> - **Efficient offline evaluation**
>
> ## W2: Runtime comparison between DeepBlip and DeepBlip-WDO
>
> Thank you for your suggestion. The primary goal of double optimization is to enable scalable training instead of accelerating batch iteration. Nevertheless, we provide the training time comparison between DeepBlip and DeepBlip-WDO to validate this:
>
> DeepBlip (with double optimiztion): 3.6 min / epoch (See Table 3\)
> DeepBlip-WDO : 3.1 min / epoch
>
> **Action**: We will add the runtime of DeepBlip-WDO to the runtime report in Table 3\.
>
> ## W3: Why the double optimization trick improves performance
>
> We appreciate this question as it involves explaining one of the main novelties of our paper. Let’s look at the original objective in Eq. (5). We aim to minimize a group of $\\tau \+ 1$ objectives $\\mathcal{L}\_{k}$, i.e. solve all the blip predictors $\\psi_{t, k}$ ($0 \\leq k \\leq \\tau$). To achieve unbiased estimates, previously we can only solve them in a backward order ($k \= \\tau$ down to $k \= 0$). We provide an intuitive explanation using $\\tau \= 2$ as an example.
>
> **The problem without DO (DeepBlip-WDO).** When all $\\psi\_k$ share the same network and all losses are optimized simultaneously, the gradient w.r.t. $\\psi\_2$ is:
>
> $$\\nabla\_{\\psi\_2} \\mathcal{L} \= \\underbrace{\\nabla\_{\\psi\_2} \\mathcal{L}\_2}\_{\\text{correct gradient}} \+ \\underbrace{\\nabla\_{\\psi\_2} \\mathcal{L}\_1 \+ \\nabla\_{\\psi\_2} \\mathcal{L}\_0}\_{\\text{spurious gradient}}$$
>
> The spurious terms $\\nabla\_{\\psi\_2} \\mathcal{L}\_1$ and $\\nabla\_{\\psi\_2} \\mathcal{L}\_0$ pull $\\psi\_2$ in directions that minimize $\\mathcal{L}\_1$ and $\\mathcal{L}\_0$ given the \*\*current\*\* (incorrect) estimates of $\\psi\_0$ and $\\psi\_1$. This leads to slow convergence.
>
> **How DO resolves this.** The DO trick detaches the copy $\\hat{\\psi}^2$ that appears in $\\mathcal{L}\_1$ and $\\mathcal{L}\_0$. As a result, the gradient becomes:
>
> $\\nabla\_{\\psi\_2} \\mathcal{L}\_{\\text{DO}} \= \\nabla\_{\\psi\_2} \\mathcal{L}\_2 \+ 0 \+ 0$
>
> Each $\\psi\_k$ receives gradients only from its own loss $L\_k$, preventing the spurious gradient described above. We also show theoretically in Appendix D that DO converges: $\\psi\_\\tau$ converges first, then $\\psi\_{\\tau-1}$, and so on.
>
> **Empirical evidence:** We validate this empirically in Figures 9-10 in Appendix E.2: with DO, the predicted blip coefficients are centered around zero error; without DO, systematic bias is clearly visible in all steps.
>
> **Action**: We will add this gradient-level explanation to the main text to make the DO trick more easy to understand.
>
> ## W4: Whether cross-fitting is required
>
> Yes, cross-fitting is employed in our framework, as required by any orthogonal two-stage learner. This procedure is already described in Algorithm 1 in Appendix H.
>
> **Action:** We will add details about cross-fitting in Section 4.4 and supply details in Appendix I (implementation details).
>
> ## W5: GT outperforms DML
>
> A plausible explanation for this phenomenon is history encoding: GT uses a transformer to encode the full patient history $H\_t$ into a rich representation. Dynamic DML does not address such dynamic encoding, with concatenated history into a fixed-size vector. This shows Dynamic DML’s limitation in a dynamic setting, despite its principled SNMM adjustments.
>
> ## W6 Dataset difference
>
> We thank the reviewer for this question. We clarify the following: we extend the tumor dataset to provide a more thorough assessment of robustness to time-varying confounding. For the MIMIC-III experiment, we use a consistent structure with existing benchmarks while adding a more heterogeneous treatment effect, instead of a single minimal effect. Hence, our DGPs are more challenging by design. If the reviewer considers it essential, we are also willing to add results on the exact DGP in the supplement as an additional validation.

---

> > ### Author Rebuttal · Reviewer_JDPn · 2026-04-02
> >
> > Thanks for the responses. My concerns are well addressed. I am happy to increase my score, conditional on the changes being incorporated into the revised version.
> >
> > A few minor comments:
> > 1. The claimed contribution 'Efficient offline evaluation' could be further validated in experiments.
> > 2. While cross fitting is theoretically required, neural-network-based implementations in practice are often trained on the full dataset, e.g, [1], with appropriate regularization to avoid overfitting.
> > 3. It would also be nice to see results on the original datasets.
> >
> > [1] Nonparametric estimation of heterogeneous treatment effects: From theory to learning algorithms. In International Conference on Artificial Intelligence and Statistics

---

> > > ### Author Response · Authors · 2026-04-07
> > >
> > > We thank the reviewer for the constructive follow-up comments.
> > >
> > > **Point 1 (Offline evaluation):** We have **added a dedicated experiment** validating the offline evaluation efficiency. We compare the wall-clock time for (a) evaluating the CATE between two given treatment sequences and (b) finding the optimal treatment sequence (with exhaustive search), across all methods and increasing $\\tau$. The results confirm that DeepBlip's time remains nearly constant (\~10–18s) while baselines grow exponentially — e.g., at $\\tau=10$ this becomes simply infeasible. The full figure and description are included in the \[[Linked PDF](https://anonymous.4open.science/r/ICML2026-DeepBlip-rebuttal-pdf/DeepBlip_rebuttal-2.pdf)\] (Figure 1).
> > >
> > > **Point 2 (Cross-fitting):** We appreciate this reference. We note that the practical justification in \[1\] applies to settings where the nuisance model directly predicts outcomes — there, mild overfitting is absorbed by the loss. In our two-stage framework, the situation is different: Stage 2 trains on the residuals $\\tilde{Y}\_{t,k}$ and $\\tilde{A}\_{t,j,k}$ computed from Stage 1's outputs. If Stage 1 overfits, these residuals become biased, directly corrupting the $L^2$-moment loss. Standard regularization mitigates but does not eliminate this issue. Hence, it’s safer to always use cross-fitting to obtain the residuals.
> > >
> > > **Point 3 (Original datasets):** We have **added new experiments** on the original DGPs from \[1\] and \[2\]. The results are included in the \[[Linked PDF](https://anonymous.4open.science/r/ICML2026-DeepBlip-rebuttal-pdf/DeepBlip_rebuttal-2.pdf)\] (Figure 2,3). We find that our method outperforms the second best baseline by 17.9\% ($\gamma = 10$) on tumor dataset and 21.2\% ($\\tau = 10$) on MIMIC dataset respectively, which demonstrates the ability to adjust for time-varying confounding and stabilizing over long horizons.
> > >
> > > **Action**: We will incorporate these changes into the revised paper.

---

### Decision · Program_Chairs · 2026-04-30

**Decision:**

Accept (regular)

**Comment:**

Reviewers agreed that this paper presents a novel and theoretically sound neural framework for Structural Nested Mean Models. While some initial concerns were raised regarding computational efficiency and interpretability, the authors' rebuttal successfully addressed these points, showing the framework's robustness to nuisance model misspecification and the practical value in downstream tasks such as healthcare applications.

All reviewers have converged on a positive recommendation (2x Accept, 2x Weak Accept). Therefore, the paper is recommended for acceptance to the conference.